# Tolerant Algorithms for Learning with Arbitrary Covariate Shift

**Surbhi Goel**
Department of Computer Science
University of Pennsylvania
surbhig@seas.upenn.edu

**Abhishek Shetty**[*]
Department of EECS
UC Berkeley
shetty@berkeley.edu

**Konstantinos Stavropoulos**[†]
Department of Computer Science
UT Austin
kstavrop@utexas.edu

**Arsen Vasilyan**[‡]
Department of EECS
UC Berkeley
arsenvasilyan@gmail.com

## Abstract

We study the problem of learning under arbitrary distribution shift, where the learner is trained on a labeled set from one distribution but evaluated on a different, potentially adversarially generated test distribution. We focus on two frameworks: *PQ learning* [GKKM20], allowing abstention on adversarially generated parts of the test distribution, and *TDS learning* [KSV24b], permitting abstention on the entire test distribution if distribution shift is detected. All prior known algorithms either rely on learning primitives that are computationally hard even for simple function classes, or end up abstaining entirely even in the presence of a tiny amount of distribution shift.

We address both these challenges for natural function classes, including intersections of halfspaces and decision trees, and standard training distributions, including Gaussians. For PQ learning, we give efficient learning algorithms, while for TDS learning, our algorithms can tolerate moderate amounts of distribution shift. At the core of our approach is an improved analysis of spectral outlier-removal techniques from learning with nasty noise. Our analysis can (1) handle arbitrarily large fraction of outliers, which is crucial for handling arbitrary distribution shifts, and (2) obtain stronger bounds on polynomial moments of the distribution after outlier removal, yielding new insights into polynomial regression under distribution shifts. Lastly, our techniques lead to novel results for tolerant *testable learning* [RV23], and learning with nasty noise.

## 1 Introduction

Despite the tremendous progress of machine learning, real-world deployment and use of machine learning models has proven challenging. A major reason for this is *distribution shift*, which occurs when the model is trained on one distribution $\mathcal{D}^{\text{train}}$ over $\mathcal{X} \times \{\pm 1\}$, while the data during deployment comes from a different distribution $\mathcal{D}^{\text{test}}$. In such scenarios, a model can unexpectedly make incorrect

---

[*]Supported by an Apple AI/ML PhD Fellowship

[†]Supported by the NSF AI Institute for Foundations of Machine Learning (IFML) and by scholarships from Bodossaki Foundation and Leventis Foundation.

[‡]Supported by NSF awards CCF-2006664, DMS-2022448, CCF-1565235, CCF-1955217, Big George Fellowship and Fintech@CSAIL. Part of this work was conducted while the author was visiting the Simons Institute for the Theory of Computing.

38th Conference on Neural Information Processing Systems (NeurIPS 2024).

predictions, leading to loss of reliability, as well as erosion of trust in the machine learning system itself. Among many other critical applications, distribution shift continues to be a major challenge in healthcare applications [ZBL+18, SS20, WOD+21, TCK+22].

Handling distribution shift when $\mathcal{D}^{\text{train}}$ and $\mathcal{D}^{\text{test}}$ are allowed to be arbitrary is known to be impossible [DLLP10]. To circumvent this impossibility, recent works [GKKM20, KK21, KSV24b, GHMS24, KSV24a] allow the machine learning model to additionally *abstain* (not make a prediction) on some or all of the inputs. These frameworks generalize standard PAC learning, requiring the algorithm to abstain from making predictions rather than giving incorrect predictions. In this work, we focus on two such frameworks for binary classification:

**PQ learning** [GKKM20, KK21], requiring the learning algorithm to output a *selective* classifier $\widehat{f}$, which is allowed to abstain on some inputs and simultaneously satisfy: (i) $\epsilon$-*accuracy:* the probability that $\widehat{f}$ does not abstain and incorrectly classifies an input $\mathbf{x}$ from the test distribution $\mathcal{D}^{\text{test}}$ is at most $\epsilon$, and (ii) $\epsilon$-*rejection rate:* the probability that $\widehat{f}$ abstains on an input $\mathbf{x}$ from the original distribution $\mathcal{D}^{\text{train}}$ is at most $\epsilon$. In particular, this implies that $\widehat{f}$ abstains on $\mathcal{D}^{\text{test}}$ with probability at most $\mathrm{d}_{\text{TV}}(\mathcal{D}^{\text{train}}, \mathcal{D}^{\text{test}}) + \epsilon$, i.e. the probability of abstention deteriorates only in proportion to the amount of distribution shift.

**Testable Distribution Shift (TDS)** [KSV24b], allowing the classifier to abstain on the entire distribution $\mathcal{D}^{\text{test}}$ if any distribution shift is detected. If there is no distribution shift, then the classifier is $\epsilon$-accurate on $\mathcal{D}^{\text{test}}$.

Prior known algorithms for both these settings have strong inherent limitations, making them impractical for real-world scenarios. For PQ learning, all known algorithms require access to oracles that are computationally inefficient even for the most basic concept classes and training distributions. For example, even for the most basic class of halfspaces (linear separators) over $\mathbb{R}^d$ under the Gaussian training distribution, no PQ learning algorithm has run-time better than $2^{d^{\Omega(1)}}$. On the other hand, TDS learning algorithms, while being computationally efficient, reject entire test sets even when the test set has a tiny amount of distribution shift. For example, the algorithms of [KSV24b], use the *low-degree moment-matching approach*, which can reject distributions $\mathcal{D}^{\text{test}} \neq \mathcal{D}^{\text{train}}$ even when $\mathrm{d}_{\text{TV}}(\mathcal{D}^{\text{train}}, \mathcal{D}^{\text{test}}) = o(\epsilon)$.

## 1.1 Our results

In this work, we overcome both these limitations using a unified approach: spectral outlier removal [DKS18] in tandem with strong polynomial approximation results in terms of $\mathcal{L}_2$-sandwiching [KSV24b]. For PQ learning, we give the first dimension-efficient learning algorithms. For TDS learning, we give the first tolerant TDS learners that accept test sets with moderate amount of distribution shift in TV distance, $\mathrm{d}_{\text{TV}}(\mathcal{D}^{\text{train}}, \mathcal{D}^{\text{test}}) = O(\epsilon)$. We summarize our results in Table 1.

| Concept class $\mathcal{F}$ | $\mathcal{D}_{\mathcal{X}}^{\text{train}}$ | PQ runtime | TDS runtime |
|---|---|---|---|
| Halfspaces (*realizable*) | $\mathcal{N}(0, I)$ | $d^{O(\log 1/\epsilon)}$ | $d^{O(\log 1/\epsilon)}$ |
| Halfspaces | $\mathcal{N}(0, I), \mathcal{U}\left(\{\pm 1\}^d\right)$ | $d^{\tilde{O}(1/\epsilon^4)}$ | $d^{\tilde{O}(1/\epsilon^2)}$ |
| Intersections of $\ell$ halfspaces | $\mathcal{N}(0, I), \mathcal{U}\left(\{\pm 1\}^d\right)$ | $d^{\tilde{O}(\ell^6/\epsilon^4)}$ | $d^{\tilde{O}(\ell^6/\epsilon^2)}$ |
| Size-$s$ decision trees | $\mathcal{U}\left(\{\pm 1\}^d\right)$ | $d^{O(\log(s/\epsilon))}$ | $d^{O(\log(s/\epsilon))}$ |
| Size-$s$ depth-$\ell$ formulas | $\mathcal{U}\left(\{\pm 1\}^d\right)$ | $d^{O(\sqrt{s}(\log(s/\epsilon))^{5\ell/2})}$ | $d^{O(\sqrt{s}(\log(s/\epsilon))^{5\ell/2})}$ |

Table 1: Summary of our results for PQ learning and tolerant TDS learning. Except for the first row, all results are for the agnostic noise model.

**Application: Testable Agnostic Learning.** Our techniques give new learning algorithms in the *testable agnostic learning* framework of [RV23]. Testable learning does not address distribution shift, as it assumes that the training and testing distributions are the same. Similarly to the TDS learning algorithms of [KSV24b], all known testable agnostic learning algorithms are based either entirely [RV23, GKK23] or partially [GKSV23, DKK+23, GKSV24] on low-degree moment matching[4], and

---

[4]In fact, [GKSV24], is partially based on a slightly more general hypercontractivity tester by [KS17]. However, this tester will reject some distributions $\mathcal{D}^{\text{test}} \neq \mathcal{D}^{\text{train}}$ for which $\mathrm{d}_{\text{TV}}(\mathcal{D}^{\text{train}}, \mathcal{D}^{\text{test}}) = o(\epsilon)$.

are subsequently not tolerant to small amounts of violations of the testing assumption in TV distance. We give the first tolerant testable learning algorithms for a number of function classes, including Halfspaces and low-depth formulas (see Table 2 in Appendix D for details).

**Application: Learning with Nasty Noise.** As a corollary of our tolerant agnostic learning algorithms we obtain algorithms that withstand an $\Omega(\epsilon)$ amount of nasty noise corruption, and produce classifiers with an error at most $\epsilon$. (In this setting $\Omega(\epsilon)$ fraction of both labels and examples given to the algorithm are corrupted). The error bound of $\epsilon$ compares favorably with the bound of [KKM18] under $\Omega(\epsilon)$ nasty noise, which is $\sqrt{\epsilon}$. Compared with the results of [DKS18] in the nasty noise setting, our results are incomparable (see relevant discussion in Section 3, Appendix D.2 for more information).

## 1.2 Our Techniques

To explain our technical approach, we focus of the PQ learning setting (in TDS learning setting and testable agnostic learning setting, our approach is analogous). If the TV distance between $\mathcal{D}^{\text{test}}$ and the training distribution $\mathcal{D}^{\text{train}}$ is at most $\mathrm{d}_{\text{TV}}(\mathcal{D}^{\text{train}}, \mathcal{D}^{\text{test}})$, then we think of the dataset as consisting of $1 - \mathrm{d}_{\text{TV}}(\mathcal{D}^{\text{train}}, \mathcal{D}^{\text{test}})$ fraction of *inliers* and an $\mathrm{d}_{\text{TV}}(\mathcal{D}^{\text{train}}, \mathcal{D}^{\text{test}})$ fraction of *outliers*. In order to accomplish PQ learning, we aim to remove a portion of the test set while (i) ensuring that a learning algorithm based on low-degree polynomial regression [KOS08] works on the remaining data (ii) not removing more than $\epsilon$ fraction of the inliers[5].

It was known from [KSV24b] that the degree-$k$ polynomial regression performs correctly if the dataset satisfies the degree-$k$ moment-matching test. Despite its power, the low-degree moment test can reject distributions even $\epsilon/d^{O(k)}$-close to the reference distribution. However (i) it is not clear how to efficiently prune the dataset, so the remaining datapoints satisfy the moment-matching condition (ii) even if one could do this efficiently, this can require one remove a constant fraction of inliers. To overcome this issue, we introduce the notion of *low-degree spectral boundedness*, which requires that for every degree-$k$ polynomial $p$ the expectation $\mathbb{E}_{\mathbf{x} \sim \mathcal{D}}[p(\mathbf{x})^2]$ does not exceed the analogous expectation with respect to the reference distribution by more than a desired factor. Our first key insight is that by using the notion of $\mathcal{L}_2$-sandwiching polynomials [KSV24b], for many settings the low-degree moment matching test can be replaced by this low-degree spectral boundedness test.

If our dataset does not satisfy low-degree spectral boundedness, our second key insight is to make it do so by removing outliers. As in many other algorithms based on outlier removal[6] (see e.g. [DKK+19, LRV16, HLZ20, Ste18, DK19, DV04] and references therein), our outlier-removal algorithm repeatedly finds regions in $\mathbb{R}^d$, such that at least $1 - \epsilon$ fraction of points in them are outliers. This way, as we remove all the points in such outlier-rich regions, we will not remove too many inliers. Finally, we find such outlier-rich regions efficiently using a spectral approach. Specifically, if the dataset $S$ does not satisfy the low-degree spectral boundedness, then there is some polynomial $p$ for which $\mathbb{E}_{x \sim S}[p(\mathbf{x})^2]$ is much greater than the corresponding expectation over the reference distribution. We infer that, for an appropriate value of $\tau$, at least $1 - \epsilon$ fraction of points in the region $\{\mathbf{x} : p(\mathbf{x})^2 > \tau\}$ are outliers.

## 1.3 Related work

**Domain Adaptation.** During the last two decades, there has been a long line of works in domain adaptation literature (see, e.g., [BDBCP06, BDBC+10, MMR09, BCK+07, DLLP10, RMH+20, KZZ24] and references therein), aiming to provide generalization bounds for the error on the test distribution, after training using only labeled examples from the training distribution. However, the generalization bounds provided involve distances between the training and test marginals that typically involve enumerations over the whole concept class and no efficient algorithms for estimating or even testing such distances directy are available.

**PQ Learning.** The PQ learning framework was defined by [GKKM20], which showed that a PQ learner can be efficiently implemented using an oracle to a distribution-free agnostic learner. In follow-up work by [KK21], it was shown that distribution-free PQ learning is actually equivalent to distribution-free agnostic reliable learning, which is a learning primitive known to be hard even for

---

[5]Precisely, we aim to avoid removing more than $\epsilon N$ outliers, where $N$ is the size of the test dataset.

[6]Our notion of outlier removal is connected to the notion of sampling correctors from [CGR16]. We note that over $\mathbb{R}^d$ the algorithms of [CGR16] run in time $2^{\Omega(d)}$, while ours are dimension-efficient.

the fundamental class of halfspaces $(\exp(\Omega(\sqrt{d}))$ time is believed to be necessary). Here, we show how to take advantage of standard assumptions on the training marginal (e.g., Gaussianity) in order to obtain the first dimension-efficient results for PQ learning of several fundamental concept classes.

**TDS Learning.** Testable learning with distribution shift was defined recently by [KSV24b], where dimension-efficient algorithms for several concept classes including halfspaces, halfspace intersections, decision trees and boolean formulas were provided. In this work, we give similar results for each of these classes in the tolerant TDS learning framework. Further work by [KSV24a] provided improved guarantees for TDS learning halfspace intersections in the realizable case. We believe that our techniques can likely be used to provide similar improvements for tolerant TDS learning, but, for ease of exposition, we do not include such results in this work.

**Tolerant Distribution Testing:** The notion of tolerance in property testing was introduced in [PRR06] and has been the focus of many works including [FF05, VV11, BCE+19, RV20, CJKL22, CFG+22, BH18, CP23]. However, over $\mathbb{R}^d$ all existing tolerant distribution testing algorithms (such as [VV11]) have run-times and sample complexities of $2^{\Omega(d)}$, which greatly exceeds our run-times.

## 2 Preliminaries

**Notation.** For details on the notation, see Appendix A. We denote with $\mathbf{x}^{\otimes k}$ the vector of monomials of degree $k$ of $\mathbf{x} \in \mathbb{R}^d$, i.e., $\mathbf{x}^{\otimes k}$ is a vector of length $d^k$ with elements of the form $\mathbf{x}^r = \prod_{i=1}^d \mathbf{x}^{r_i}$, where $\sum_{i\in[d]} r_i \leq k$, $r_i \in \mathbb{N}$, $r = (r_1, \dots, r_d)$ and $k$ is the degree of $\mathbf{x}^r$. A polynomial $p$ over $\mathbb{R}^d$ is a function $p(\mathbf{x}) = \sum_{r\in\mathbb{N}^d} p_r \mathbf{x}^r = p^\top \mathbf{x}^{\otimes d}$, where we abuse the notation to denote with $p$ the vector of coefficients of the corresponding polynomial. A polynomial $p$ over $\{\pm 1\}^d$ is defined similarly, but all of the coefficients corresponding to monomials $\mathbf{x}^r$ where $r_i > 1$ for some $i$ are zero.

**Learning Setting.** We consider distribution $\mathcal{D}$ over $\mathcal{X}$ and $\mathcal{D}^{\text{train}}, \mathcal{D}^{\text{test}}$ distributions over $\mathcal{X} \times \{\pm 1\}$ such that the marginal on $\mathcal{X}$ of $\mathcal{D}^{\text{train}}$ is $\mathcal{D}$ and the marginal of $\mathcal{D}^{\text{test}}$ is $\mathcal{D}^{\text{test}}_{\mathcal{X}}$. We also consider some concept class $\mathcal{F} \subseteq \{\mathcal{X} \to \{\pm 1\}\}$. The learner is given access to labeled examples from $\mathcal{D}^{\text{train}}$ as well as unlabeled examples from $\mathcal{D}^{\text{test}}_{\mathcal{X}}$ and the goal is to produce some hypothesis with low error on $\mathcal{D}^{\text{test}}$, but is also allowed to abstain from predicting either on specific points (for PQ learning, Def. 4.1) or even the entire distribution (for TDS learning, Def. 5.1) if distribution shift is detected.

In the **realizable** setting, the labels of both the training distribution and the test distribution are generated according to some concept $f^* \in \mathcal{F}$ and the training examples are of the form $(\mathbf{x}, f^*(\mathbf{x}))$, where $\mathbf{x} \sim \mathcal{D}$. The target test error is $\epsilon$ for some arbitrarily chosen $\epsilon \in (0, 1)$. In the **agnostic** setting, the distributions $\mathcal{D}^{\text{train}}$ and $\mathcal{D}^{\text{test}}$ can be arbitrary, except from the assumption that the marginal of $\mathcal{D}^{\text{train}}$ is $\mathcal{D}^{\text{train}}_{\mathcal{X}} = \mathcal{D}$. To quantify the target error, we use parameter $\lambda = \lambda(\mathcal{F}; \mathcal{D}^{\text{train}}, \mathcal{D}^{\text{test}}) = \min_{f\in\mathcal{F}}(\text{err}(f; \mathcal{D}^{\text{train}}) + \text{err}(f; \mathcal{D}^{\text{test}}))$, where $\text{err}(f; \mathcal{D}^{\text{train}}) = \mathbb{P}_{(\mathbf{x},y)\sim\mathcal{D}^{\text{train}}}[y \neq f(\mathbf{x})]$ (and similarly for $\text{err}(f; \mathcal{D}^{\text{test}})$). The error guarantee we can hope for is some function of $\lambda$, because $\lambda$ encodes the (unknown) relationship between the training and test distributions, in that $\lambda$ is small when there is a concept in the class $\mathcal{F}$ that has low error on both training and test distributions. Error bounds in terms of $\lambda$ are standard (and necessary) in the domain adaptation literature (see, e.g., [BDBCP06, BDBC+10]) as well as TDS learning (see [KSV24b]).

**Properties of Distributions.** We make standard assumptions about training marginal $\mathcal{D}$. We denote with $\mathcal{N}_d$ the standard Gaussian distribution over $\mathbb{R}^d$ and with $\text{Unif}(\{\pm 1\}^d)$ the uniform distribution over the hypercube $\{\pm 1\}^d$. A distribution $\mathcal{D}$ over $\mathcal{X}$ is $k$-**tame** if for every degree-$k$ polynomial $p$ over $\mathcal{X}$ with $\mathbb{E}_{\mathbf{x}\sim\mathcal{D}}[(p(\mathbf{x}))^2] \leq 1$ and every $B$ greater than $e^k$ we have $\mathbb{P}_{\mathbf{x}\sim\mathcal{D}}\left[(p(\mathbf{x}))^2 > B\right] \leq e^{-\Omega(B^{1/(2k)})}$. And note that the Gaussian distribution, all isotropic log-concave distributions over $\mathbb{R}^d$, as well as the uniform distribution over $\{\pm 1\}^d$ are $k$-tame for all $k \in \mathbb{N}$ (see Appendix A.4).

For a concept class $\mathcal{F}$, a distribution $\mathcal{D}$ over $\mathcal{X}$, $\epsilon \in (0, 1)$, we say that $\mathcal{F}$ has $\epsilon$-$\mathcal{L}_2$ **sandwiching degree** $k$ with respect to $\mathcal{D}$ if for any $f \in \mathcal{F}$, there exist polynomials $p_{\text{up}}, p_{\text{low}}$ over $\mathcal{X}$ with degree at most $k$ such that (1) $p_{\text{low}}(\mathbf{x}) \leq f(\mathbf{x}) \leq p_{\text{up}}(\mathbf{x})$ for all $\mathbf{x} \in \mathcal{X}$ and (2) $\mathbb{E}_{\mathbf{x}\sim\mathcal{D}}[(p_{\text{up}}(\mathbf{x}) - p_{\text{low}}(\mathbf{x}))^2] \leq \epsilon$. If the coefficients of $p_{\text{up}}, p_{\text{low}}$ are all absolutely bounded by $B$, we say that $\mathcal{F}$ has $\epsilon$-$\mathcal{L}_2$ sandwiching coefficient bound $B$.

# 3 Outlier Removal Procedure

The key ingredient of our approach is an outlier removal procedure which is closely related to the corresponding procedure proposed by [DKS18] in the context of learning with nasty noise, but ours enjoys stronger error guarantees and works even when the fraction of outliers is arbitrarily large. The last property is important because we aim to handle arbitrary covariate shifts. Our outlier removal procedure outputs a selector $g : \mathcal{X} \to \{0, 1\}$ that satisfies two main guarantees, provided examples drawn independently from some arbitrary, unknown distribution $\mathcal{D}'$: (1) for any low-degree polynomial $p$, the part of the expectation of $p^2(\mathbf{x})$ under $\mathcal{D}'$ within the selected subset of $\mathcal{X}$ (i.e., $\mathbb{E}_{\mathbf{x} \sim \mathcal{D}'}[p^2(\mathbf{x})g(\mathbf{x})]$), is a bounded multiple of the expectation of $p^2(\mathbf{x})$ under the reference distribution $\mathcal{D}$ and (2) the probability of rejecting a fresh sample drawn from $\mathcal{D}$ (i.e., $\mathbb{P}_{\mathbf{x} \sim \mathcal{D}}[g(\mathbf{x}) = 0]$) is bounded by a multiple of the statistical distance between $\mathcal{D}$ and $\mathcal{D}'$. Formally, we prove the following theorem.

**Theorem 3.1** (Outlier Removal, see Appendix E). *There exists an algorithm (Algorithm 1) that, given sample access to an arbitrary distribution $\mathcal{D}'$ over $\mathcal{X} \subseteq \mathbb{R}^d$, sample access to a $k$-tame probability distribution $\mathcal{D}$ over $\mathcal{X}$, parameters $\epsilon, \alpha, \delta \in (0, 1)$ and $k \in \mathbb{N}$, runs in time $\mathrm{poly}(\frac{1}{\epsilon}(kd)^k \log \frac{1}{\delta})$ and outputs a succinct $\mathrm{poly}(\frac{1}{\epsilon}(kd)^k \log \frac{1}{\delta})$-time-computable description of a function $g : \mathcal{X} \to \{0, 1\}$ that satisfies the following properties with probability at least $1 - \delta$.*

(a) $\mathbb{E}_{\mathbf{x} \sim \mathcal{D}'}\left[(p(\mathbf{x}))^2 g(\mathbf{x})\right] \leq \frac{200}{\alpha} \mathbb{E}_{\mathbf{x} \sim \mathcal{D}}[(p(\mathbf{x}))^2]$, *for any polynomial $p$ with $\deg(p) \leq k$.*

(b) $\mathbb{P}_{\mathbf{x} \sim \mathcal{D}}[g(\mathbf{x}) = 0] \leq \alpha \, \mathrm{d}_{\mathrm{TV}}(\mathcal{D}, \mathcal{D}') + \frac{\epsilon}{2}$.

*Remark* 3.2. In Theorem 3.1, Condition (b) also implies some bound on the rejection rate over the distribution $\mathcal{D}'$ and, in particular, $\mathbb{P}_{\mathbf{x} \sim \mathcal{D}'}[g(\mathbf{x}) = 0] \leq (1 + \alpha)\mathrm{d}_{\mathrm{TV}}(\mathcal{D}, \mathcal{D}') + \epsilon/2$.

*Remark* 3.3. Our algorithm further satisfies a strengthened form of Condition (b) (with probability at least $1 - \delta$). For $\sigma > \alpha/2$ and any distribution $\mathcal{D}''$ that is $1/\sigma$-smooth w.r.t. $\mathcal{D}$, (i.e. for any measurable set $T \subset \mathbb{R}^d$ we have $\mathbb{P}_{\mathbf{x} \sim \mathcal{D}''}[\mathbf{x} \in T] \leq \frac{1}{\sigma} \mathbb{P}_{\mathbf{x} \sim \mathcal{D}}[\mathbf{x} \in T]$) it is the case that

$$\mathbb{P}_{\mathbf{x} \sim \mathcal{D}}[g(\mathbf{x}) = 0] \leq \frac{\alpha}{\sigma} \, \mathrm{d}_{\mathrm{TV}}(\mathcal{D}'', \mathcal{D}') + \frac{\epsilon}{2},$$

which in particular implies that $\mathbb{P}_{\mathbf{x} \sim \mathcal{D}}[g(\mathbf{x}) = 0] \leq \epsilon/2$ if $\mathcal{D}'$ itself is $2/\alpha$-smooth w.r.t. $\mathcal{D}$.

---

**Algorithm 1:** Outlier Removal Procedure

---

**Input:** Sets $S_{\mathcal{D}}, S_{\mathcal{D}'}$, each of size $N$, containing points in $\mathcal{X} \subseteq \mathbb{R}^d$ and parameters $k, \epsilon, \delta, \alpha$
**Output:** A succinct description of a selector function $g : \mathcal{X} \to \{0, 1\}$
Let $t = \binom{d+k-1}{k}$, $B = \frac{4}{\epsilon}d^{3k}$ and $\Delta = 200Bd^k(\frac{\log N}{N}\log(1/\delta))^{1/2}$
Compute monomial correlations estimate $\widehat{M}$ by running Algorithm 2 on inputs $S_{\mathcal{D}}$, $k$ and $\delta/10$.
$S^0 \leftarrow S_{\mathcal{D}'} \setminus \{\mathbf{x} \in S_{\mathcal{D}'} : \text{there is } p \in \mathbb{R}^t \text{ with } (p^\top \mathbf{x}^{\otimes k})^2 > B \text{ and } p^\top \widehat{M} p \leq 1\}$
**for** $i = 1, 2, \ldots, N$ **do**

    Let $p_i \in \mathbb{R}^t$ be the solution and $\mu_i$ and the value of the following quadratic program.

$$\max_{p \in \mathbb{R}^t} \frac{1}{N} \sum_{\mathbf{x} \in S^{i-1}} (p^\top \mathbf{x}^{\otimes k})^2 \quad \text{s.t.: } p^\top \widehat{M} p \leq 1$$

    **if** $\mu_i \leq \frac{50}{\alpha}(1 + \Delta)$ **then** set $i_{\max} = i - 1$ and exit the loop;
    **else** let $\tau_i$ be the minimum non-negative real number such that the following is true

$$\frac{1}{N}\left|\{\mathbf{x} \in S^{i-1} : (p_i^\top \mathbf{x}^{\otimes k})^2 > \tau_i\}\right| \geq \frac{10}{\alpha}\left(\mathbb{P}_{\mathbf{x} \sim S_{\mathcal{D}}}[B \geq (p_i^\top \mathbf{x}^{\otimes k})^2 > \tau_i] + \Delta\right)$$

    Set $S^i \leftarrow S^{i-1} \setminus \{\mathbf{x} \in S^{i-1} : (p_i^\top \mathbf{x}^{\otimes k})^2 > \tau_i\}$ ;

Set $g(\mathbf{x})$ to be 0 if and only if either there is $p \in \mathbb{R}^t$ with $(p^\top \mathbf{x}^{\otimes k})^2 > B$ and $p^\top \widehat{M} p \leq 1$, or $(p_i^\top \mathbf{x}^{\otimes k})^2 > \tau_i$ for some $i \in [i_{\max}]$. Otherwise, set $g(\mathbf{x}) = 1$.

---

The outlier removal procedure of Theorem 3.1 iteratively solves a quadratic program with quadratic constraints (which can be solved efficiently, see Appendix E.1.1) and increases the rejection region by setting $g(\mathbf{x}) = 0$ on each point $\mathbf{x}$ where the corresponding (maximum second moment) polynomial

takes large values. The procedure halts when the solution of the quadratic program has value bounded by $O(1/\alpha)$ (which implies condition (a)).

**Proof overview.** The main idea for the analysis is that whenever the stopping criterion does not hold, then there is a polynomial with unreasonably large second moment over the remaining part of $\mathcal{D}'$ (after the rejections). When such a polynomial $p$ exists, there must be a threshold $\tau$ for the squared values of $p$ such that $\mathcal{D}'$ assigns $\Omega(1/\alpha)$ times more mass on non-rejected points $\mathbf{x}$ with $p^2(\mathbf{x}) > \tau$ compared to the reference distribution $\mathcal{D}$. Such points can be safely rejected, because, in that case, the mass of points under $\mathcal{D}$ rejected is multiplicatively smaller (by a factor of $O(\alpha)$) than the corresponding mass under $\mathcal{D}'$ (which implies condition (b)). Note that the procedure will have to end eventually, because in each iteration, at least one example is removed.

In order to account for errors incurred by sampling (from $\mathcal{D}$ and $\mathcal{D}'$), it is important to provide a bound on the number of iterations that is independent from the number of examples drawn, because the complexity of the selector $g$ depends on the number of iterations and we need the desired properties of $g$ to generalize to the actual distributions $\mathcal{D}$ and $\mathcal{D}'$. To this end, we consider the trace of the matrix $M_i = \frac{1}{N} \sum_{\mathbf{x} \in S^i} (\mathbf{x}^{\otimes k})(\mathbf{x}^{\otimes k})^\top$ as a potential function and we show that it reduces by a multiplicative factor in each iteration (see Claim 6 in the Appendix).

**Comparison with [DKS18].** Among all outlier removal algorithms, ours is most related to the algorithm of [DKS18], which also removes elements in regions of the form $\{\mathbf{x} : (p(\mathbf{x}))^2 > \tau\}$. However, there are two differences. First, [DKS18] assume that the fraction of outliers is bounded, while ours provides meaningful guarantees even in the presence of arbitrary fraction of outliers. In particular, we can maintain low rejection rates even in the presence of large fractions of outliers by relaxing the bound on the polynomial moments after outlier removal. This is important for PQ learning, because we need low error guarantees even when the amount of distribution shift is arbitrarily large. Second, even when the fraction of outliers is small, our bound on the second moments of polynomials does not depend on the degree and the degree dependence only appears in the runtime of the outlier removal process. This gives new insights on polynomial regression in the presence of outliers (due to distribution shift or noise). In contrast, the moment bound of [DKS18] scales with the degree of the corresponding polynomial and when the degree bound scales with the target learning error, their results become vacuous. This enables us to combine the outlier removal process with $\mathcal{L}_2$ sandwiching results from TDS learning to obtain, for example, the first dimension-efficient robust learners with nasty noise of rate $\Omega(\epsilon)$ that achieve error $\epsilon$ for the class of intersections of halfspaces. While [DKS18] also provide robust learners for this class, they only achieve error guarantees that scale as $\tilde{O}(k^{1/12}\epsilon^{1/11})$, for intersections of $k$ halfspaces. The key difference between our analysis and that of [DKS18] is that, to bound the number of iterations, we use an appropriate potential function, while [DKS18] ensure that the number of iterations is bounded by making sure to remove at least some fraction of points in each step. As a result, their stopping criterion scales with the target polynomial degree.

# 4 Selective Classification with Arbitrary Covariate Shift

In order to provide provable learning guarantees in the presence of distribution shift, when no test labels are available, one reasonable approach is to enable the model to abstain on certain regions for which the training samples do not provide sufficient information. The model should not be able to abstain frequently on samples from the training distribution, since, otherwise the provided guarantees would be vacuous (e.g., when the model abstains always). A formal definition of this framework was given by [GKKM20] and, in this section, we provide the first end-to-end, dimension-efficient algorithms for learning various fundamental classes (e.g., halfspaces) in this setting.

**PQ Setting.** We first consider the case where the test samples are independently drawn from some (potentially adversarial) distribution $\mathcal{D}^{\text{test}}$ and the goal of the learner is to achieve low error under $\mathcal{D}^{\text{test}}$ (on points where the learner does not abstain), without abstaining frequently on fresh training samples, as described formally in the following definition of agnostic PQ learning.

**Definition 4.1** (PQ Learning [GKKM20]). Let $\mathcal{F}$ be a concept class over $\mathcal{X} \subseteq \mathbb{R}^d$ and $\mathcal{D}$ a distribution over $\mathcal{X}$. The algorithm $\mathcal{A}$ is a PQ-learner for $\mathcal{F}$ with respect to $\mathcal{D}$ up to error $\gamma$, rejection rate $\eta$ and probability of failure $\delta$ if, upon receiving $m_{\text{train}}$ labeled samples from a training distribution $\mathcal{D}^{\text{train}}$ with $\mathcal{X}$-marginal $\mathcal{D}$ and $m_{\text{test}}$ unlabeled samples from a test distribution $\mathcal{D}^{\text{test}}$, algorithm $\mathcal{A}$ outputs, w.p. at least $1 - \delta$, a hypothesis $h : \mathcal{X} \to \{\pm 1\}$ and a selector $g : \mathcal{X} \to \{0, 1\}$ such that:

(a) (*accuracy*) The test error is bounded as $\mathbb{P}_{(\mathbf{x},y)\sim\mathcal{D}^{\text{test}}}[y \neq h(\mathbf{x}) \text{ and } g(\mathbf{x}) = 1] \leq \gamma$.

(b) (*rejection rate*) The probability of rejection is bounded as $\mathbb{P}_{\mathbf{x}\sim\mathcal{D}}[g(\mathbf{x}) = 0] \leq \eta$.

The error $\gamma$ and the rejection rate $\eta$ are, in general, functions of the parameter $\lambda = \lambda(\mathcal{F}; \mathcal{D}^{\text{train}}, \mathcal{D}^{\text{test}})$.

**Adversarial Setting.** Another reasonable scenario from [GKKM20] corresponds to the case where the test examples are not independent, but are chosen adversarially as follows. The adversary receives $N$ independent samples $S_{\text{iid}}$ from $\mathcal{D}$ and substitutes any number of them adversarially, forming a new unlabeled dataset $S_{\text{test}}$ which is given to the learner along with a fresh set of independent samples from $\mathcal{D}$, labeled according to some hypothesis $f^* \in \mathcal{F}$ (realizable setting). The goal is to learn a hypothesis $h : \mathcal{X} \to \{\pm 1\}$ and a set $S_g \subseteq S_{\text{test}}$ such that $\mathbb{P}_{\mathbf{x}\in S_{\text{test}}}[h(\mathbf{x}) \neq f^*(\mathbf{x}) \text{ and } \mathbf{x} \in S_g] \leq \gamma$ and $|S_{\text{iid}} \cap (S_{\text{test}} \setminus S_g)| \leq \eta N$ (only a small fraction of i.i.d. points can be rejected). Note that the adversarial setting is primarily interesting in the realizable case, since there is no underlying test distribution and for any meaningful notion learning to be possible, there needs to be some relationship between the training and test labels. In the rest of this section, we focus on positive results for PQ learning, but, as we argue in Appendix C.2, all of our positive results on (realizable) PQ learning also work analogously in the adversarial setting. This is because our outlier removal process (Theorem 3.1) also works when the examples from the distribution $\mathcal{D}'$ are in fact generated adversarially (see Theorem E.1).

## 4.1 PQ Learning of Halfspaces

We now give the first dimension-efficient PQ learning algorithms for the fundamental concept class of halfspaces, in the realizable setting and with respect to the Gaussian distribution, i.e., when both the training and the test labels are generated by some unknown halfspace and the training marginal $\mathcal{D}$ is the standard Gaussian distribution $\mathcal{N}_d$.

**Warm-Up: Homogeneous Halfspaces.** We first focus on the class $\mathcal{F}$ of homogeneous halfspaces, i.e., functions $f : \mathbb{R}^d \to \{\pm 1\}$ with $f(\mathbf{x}) = \text{sign}(\mathbf{w} \cdot \mathbf{x})$ for $\mathbf{w} \in \mathbb{S}^{d-1}$. Recent work by [KSV24b] showed that there is a simple fully polynomial-time TDS learner for this problem. In fact, their approach readily implies a PQ learner as well.

**Proposition 4.2** (Implicit in [KSV24b]). *For any $\epsilon, \delta \in (0, 1)$, there is an algorithm that PQ learns the class of homogeneous halfspaces with respect to $\mathcal{N}_d$ in the realizable setting, up to error and rejection rate $\epsilon$ and probability of failure $\delta$ that runs in time $\text{poly}(d, \frac{1}{\epsilon}) \log(1/\delta)$.*

The algorithm of [KSV24b, Proposition 5.1] rejects when the probability that a randomly chosen example $\mathbf{x}$ from the test marginal falls in some particular region $\mathbf{D}$ in $\mathbb{R}^d$ (for which there is an efficient membership oracle) is greater than $\Omega(\epsilon)$. Since the training marginal is Gaussian, the ERM, run on sufficiently many labeled training examples, outputs a hypothesis $h(\mathbf{x}) = \text{sign}(\widehat{\mathbf{w}} \cdot \mathbf{x})$ such that $\|\widehat{\mathbf{w}} - \mathbf{w}^*\|_2 \leq \epsilon'$, where $\mathbf{w}^*$ is the ground truth. Region $\mathbf{D}$ consists precisely of the points $\mathbf{x}$ for which the ERM hypothesis $h$ is not confident: there are two (potential ground truth) unit vectors $\mathbf{v}_1$ and $\mathbf{v}_2$ that are both $\epsilon'$-close to $\widehat{\mathbf{w}}$ and $\text{sign}(\mathbf{v}_1 \cdot \mathbf{x}) \neq \text{sign}(\mathbf{v}_2 \cdot \mathbf{x})$. Crucially, the Gaussian mass of $\mathbf{D}$ is known to be $\text{poly}(\epsilon') \cdot \sqrt{d}$ (see, e.g., [Han14]). Therefore, for PQ learning, we may return the classifier $h$ along with the selector $g(\mathbf{x}) = \mathbb{1}\{\mathbf{x} \notin \mathbf{D}\}$ and note that access to unlabeled test examples is not neeeded to form $h$ and $g$.

**General Halfspaces.** For the class of general halfspaces (i.e., functions of the form $f(\mathbf{x}) = \text{sign}(\mathbf{w} \cdot \mathbf{x} + \tau)$ where $\mathbf{w} \in \mathbb{S}^{d-1}$ and $\tau \in \mathbb{R}$), the labeled training samples do not always provide sufficient information to recover the unknown parameters. This is because the bias $\tau^*$ of the ground truth could take arbitrarily large positive or negative values, in which case all of the training examples will likely have the same label and (almost) no information about the ground truth $\mathbf{w}^*$ is revealed. The concern in that case is that the test marginal $\mathcal{D}'$ assigns a lot of mass far from the origin in the direction of $\mathbf{w}^*$. By appropriately applying Theorem 3.1 to select a part of the test marginal $\mathcal{D}'$ that is sufficiently concentrated in every direction (hence even in the direction of $\mathbf{w}^*$), we obtain the following PQ learning result.

**Theorem 4.3** (PQ Learning of Halfspaces). *For any $\epsilon, \delta \in (0, 1)$, there is an algorithm that PQ learns the class of general halfspaces with respect to $\mathcal{N}_d$ in the realizable setting, up to error and rejection rate $\epsilon$ and probability of failure $\delta$ that runs in time $\text{poly}(d^{\log(\frac{1}{\epsilon})}, \log(1/\delta))$.*

The first ingredient for Theorem 4.3 is a result from [KSV24b] regarding recovering the parameters of an unknown general halfspace provided labeled examples from the Gaussian distribution, which was previously used for TDS learning.

**Proposition 4.4** (Halfspace Parameter Recovery, Proposition 5.5 in [KSV24b]). *For $\epsilon, \delta \in (0,1)$ and $\tau \in \mathbb{R}$, suppose that $S$ consists of at least $m = \text{poly}(d, 1/\epsilon)e^{O(\tau^2)}\log(1/\delta)$ i.i.d. samples from $\mathcal{N}_d$, labeled by some halfspace of the form $f^*(\mathbf{x}) = \text{sign}(\mathbf{w}^* \cdot \mathbf{x} + \tau^*)$, for some $\mathbf{w}^* \in \mathbb{S}^{d-1}$. Then, with probability at least $1 - \delta$, for $\widehat{\mathbf{w}} = \sum_{(\mathbf{x},y) \in S} \mathbf{x}y / \|\sum_{(\mathbf{x},y) \in S} \mathbf{x}y\|_2$ and $\widehat{\tau} = \widehat{\mathbf{w}} \cdot \mathbf{x}$ for some $\mathbf{x}$ from $S$ such that $\mathbb{P}_{(\mathbf{x},y) \in S}[y \neq \text{sign}(\widehat{\mathbf{w}} \cdot \mathbf{x} + \widehat{\tau})]$ is minimized, we have $\|\widehat{\mathbf{w}} - \mathbf{w}^*\|_2 \leq \epsilon$ and $|\widehat{\tau} - \tau^*| \leq \epsilon$.*

Therefore, in the case when the bias $\tau^*$ of the unknown ground truth halfspace is not too large in absolute value, the selector can reject all points $\mathbf{x}$ for which there exist two halfspaces with parameters close to $\widehat{\mathbf{w}}$ and $\widehat{\tau}$ accordingly that disagree on $\mathbf{x}$ (similarly to the case of homogeneous halfspaces). Once more, such a selector can be implemented efficiently via a convex program.

When the bias is large, in TDS learning, checking whether the first $O(\log(1/\epsilon))$ moments of the test marginal $\mathcal{D}'$ match the corresponding Gaussian moments is sufficient to ensure that the distribution is concentrated in every direction and, therefore, even in the unknown direction of $\mathbf{w}^*$. In order to obtain a selective classifier for this case, we instead use Theorem 3.1 with $k = O(\log(1/\epsilon))$ and ensure that the selected part of the test marginal is indeed sufficiently concentrated in every direction as required. For more details, see Appendix C.1.1.

## 4.2 PQ for Classes with Low Sandwiching Degree

The outlier removal process of Theorem 3.1 enables one to fully control the ratios between the second moment of any low-degree polynomial under the selected part of the test marginal $\mathcal{D}'$ and its second moment under the reference distribution $\mathcal{D}$, since the provided bound (see condition (a)) does not depend on the degree of the polynomial, but only on the target rejection rate. Combining our outlier removal process with ideas from TDS learning, we provide a general result on PQ learning classes with low $\mathcal{L}_2$ sandwiching degree. In particular, we require the following properties for the hypothesis class $\mathcal{F}$ and the training marginal $\mathcal{D}$.

**Definition 4.5** (Reasonable Pairs of Classes and Distributions). We say that the pair $(\mathcal{D}, \mathcal{F})$, where $\mathcal{D}$ is a distribution over $\mathcal{X} \subseteq \mathbb{R}^d$ and $\mathcal{F} \subseteq \{\mathcal{X} \to \{\pm 1\}\}$ is $(\epsilon, \delta, k, m)$-reasonable if the following properties hold: (1) the $\epsilon$-$\mathcal{L}_2$ sandwiching degree of $\mathcal{F}$ under $\mathcal{D}$ is at most $k$ with coefficient bound $B$, (2) the distribution $\mathcal{D}$ is $k$-tame and (3) if $S$ consists of $m'$ i.i.d. samples from some distribution $\mathcal{D}$ over $\mathcal{X} \times \{\pm 1\}$ with marginal $\mathcal{D}$ and $m' \geq m$ then, with probability at least $1 - \delta$ we have that for any degree-$k$ polynomial $p$ with coefficient bound $B$ it holds $|\mathbb{E}_{\mathcal{D}_{\mathcal{X}\mathcal{Y}}}[(y - p(\mathbf{x}))^2] - \mathbb{E}_S[(y - p(\mathbf{x}))^2]| \leq \epsilon$.

Property (1) corresponds to the existence of sandwiching approximators, which is known to be important for learning in the presence of distribution shift by prior work on TDS learning [KSV24b]. Property (2) is the tameness condition Definition A.8, which is important for the outlier removal procedure and was used in the work of [DKS18] for similar purposes. Finally, property (3) ensures generalization for polynomial regression.

We obtain the following theorem which gives the first dimension-efficient results on PQ learning several fundamental concept classes with respect to standard training marginals, including intersections of halfspaces, decision trees and boolean formulas. The results work even in the agnostic setting. The algorithm runs the outlier removal process once to form the selector and runs polynomial regression on the training distribution to form the output hypothesis.

**Theorem 4.6** (PQ Learning via Sandwiching). *For $\epsilon, \eta, \delta \in (0,1)$, let $\mathcal{X} \subseteq \mathbb{R}^d$ and $(\mathcal{D}, \mathcal{F})$ be an $(\frac{\epsilon\eta}{C}, \frac{\delta}{C}, k, m)$-reasonable pair (Definition 4.5) for some sufficiently large universal constant $C > 0$. Then, there is an algorithm that PQ learns $\mathcal{F}$ with respect to $\mathcal{D}$ up to error $O(\frac{\lambda}{\eta}) + \epsilon$, rejection rate $\eta$ and probability of failure $\delta$ with sample complexity $m + \text{poly}(\frac{1}{\eta}(kd)^k \log(1/\delta))$ and time complexity $\text{poly}(\frac{m}{\eta}(kd)^k \log(1/\delta))$.*

*Proof of Theorem 4.6.* The algorithm forms the selector $g$ by applying the outlier removal process of Theorem 3.1 with parameters $\alpha, \epsilon \leftarrow \frac{\eta}{2}$, $\delta \leftarrow \delta/C$ and $k \leftarrow k$. Then, we run the following box-constrained least squares problem, using at least $m$ labeled examples $S_{\text{train}}$ from $\mathcal{D}^{\text{train}}$, where

$t = d^k$ and $B$ is the value specified in Definition 4.5.

$$\min_p \mathbb{E}_{(\mathbf{x},y)\sim S_{\text{train}}}[(y - p(\mathbf{x}))^2] \text{ s.t. } p \text{ has degree at most } k \text{ and coefficient bound } B$$

Let $\widehat{p}$ be the solution. The algorithm returns the selector $g$ and the classifier $h(\mathbf{x}) = \text{sign}(\widehat{p}(\mathbf{x}))$. The rejection rate is bounded due to condition (b) in Theorem 3.1 while the accuracy can be shown by applying condition (a) as follows, where $f^* \in \mathcal{F}$ is the concept achieving $\lambda = \min_{f\in\mathcal{F}}(\text{err}(f; \mathcal{D}^{\text{train}}) + \text{err}(f; \mathcal{D}^{\text{test}}))$ and $p_{\text{up}}, p_{\text{low}}$ are the corresponding sandwiching polynomials for $f^*$ (as per Definition 4.5). We show that $\mathbb{P}_{(\mathbf{x},y)\sim\mathcal{D}^{\text{test}}}[y \neq h(\mathbf{x}), g(\mathbf{x}) = 1] \leq O(\frac{\lambda}{\eta}) + \epsilon$.

$$\mathbb{P}_{(\mathbf{x},y)\sim\mathcal{D}^{\text{test}}}[y \neq h(\mathbf{x}), g(\mathbf{x}) = 1] \leq \mathbb{P}_{(\mathbf{x},y)\sim\mathcal{D}^{\text{test}}}[y \neq f^*(\mathbf{x})] + \mathbb{P}_{\mathbf{x}\sim\mathcal{D}'}[f^*(\mathbf{x}) \neq \text{sign}(\widehat{p}(\mathbf{x})), g(\mathbf{x}) = 1]$$

$$\leq \lambda + \mathbb{E}_{\mathbf{x}\sim\mathcal{D}'}[(f^*(\mathbf{x}) - \widehat{p}(\mathbf{x}))^2 g(x)]$$

The term $\mathbb{E}_{\mathbf{x}\sim\mathcal{D}'}[(f^*(\mathbf{x}) - \widehat{p}(\mathbf{x}))^2 g(x)]$ can be bounded as $\mathbb{E}_{\mathbf{x}\sim\mathcal{D}'}[(f^*(\mathbf{x}) - \widehat{p}(\mathbf{x}))^2 g(x)] \leq 2\,\mathbb{E}_{\mathbf{x}\sim\mathcal{D}'}[(f^*(\mathbf{x}) - p_{\text{low}}(\mathbf{x}))^2 g(x)] + 2\,\mathbb{E}_{\mathbf{x}\sim\mathcal{D}'}[(p_{\text{low}}(\mathbf{x}) - \widehat{p}(\mathbf{x}))^2 g(x)]$ and since $p_{\text{up}}, p_{\text{low}}$ sandwich $f^*$, we bound $\mathbb{E}_{\mathbf{x}\sim\mathcal{D}'}[(f^*(\mathbf{x}) - p_{\text{low}}(\mathbf{x}))^2 g(x)]$ by $\mathbb{E}_{\mathbf{x}\sim\mathcal{D}'}[(p_{\text{up}}(\mathbf{x}) - p_{\text{low}}(\mathbf{x}))^2 g(x)]$.

By applying condition (a) from Theorem 3.1, since $(p_{\text{low}}(\mathbf{x}) - \widehat{p}(\mathbf{x}))^2$ and $(p_{\text{up}}(\mathbf{x}) - p_{\text{low}}(\mathbf{x}))^2$ are squares of polynomials of degree $k$, we have the following for some sufficiently large constant $C'$.

$$\mathbb{P}_{\mathcal{D}^{\text{test}}}[y \neq h(\mathbf{x}), g(\mathbf{x}) = 1] \leq \lambda + \frac{C'}{\eta}(\mathbb{E}_{\mathbf{x}\sim\mathcal{D}}[(p_{\text{low}}(\mathbf{x}) - \widehat{p}(\mathbf{x}))^2] + \mathbb{E}_{\mathbf{x}\sim\mathcal{D}}[(p_{\text{up}}(\mathbf{x}) - p_{\text{low}}(\mathbf{x}))^2])$$

The term $\frac{C'}{\eta}\,\mathbb{E}_{\mathbf{x}\sim\mathcal{D}}[(p_{\text{up}}(\mathbf{x}) - p_{\text{low}}(\mathbf{x}))^2]$ is at most $\frac{\epsilon}{3}$ and $\frac{C'}{\eta}\,\mathbb{E}_{\mathbf{x}\sim\mathcal{D}}[(p_{\text{low}}(\mathbf{x}) - \widehat{p}(\mathbf{x}))^2]$ is bounded by $O(\frac{\lambda}{\eta}) + 2\epsilon/3$, as $\mathbb{E}_{\mathbf{x}\sim\mathcal{D}}[(p_{\text{low}}(\mathbf{x}) - \widehat{p}(\mathbf{x}))^2] \leq 2\,\mathbb{E}_{\mathcal{D}^{\text{train}}}[(p_{\text{low}}(\mathbf{x}) - y)^2] + 2\,\mathbb{E}_{\mathcal{D}^{\text{train}}}[(y - \widehat{p}(\mathbf{x}))^2]$. We have that $\mathbb{E}_{\mathcal{D}^{\text{train}}}[(p_{\text{low}}(\mathbf{x}) - y)^2] \leq 2\,\mathbb{E}_{\mathcal{D}^{\text{train}}}[(p_{\text{low}}(\mathbf{x}) - f^*(\mathbf{x}))^2] + 2\,\mathbb{E}_{\mathcal{D}^{\text{train}}}[(y - f^*(\mathbf{x}))^2] \leq \frac{\epsilon\eta}{C} + O(\lambda)$, due to sandwiching and the definition of $\lambda$. The term $\mathbb{E}_{\mathcal{D}^{\text{train}}}[(y - \widehat{p}(\mathbf{x}))^2]$ is $\frac{\epsilon\eta}{C}$-close to $\mathbb{E}_{S_{\text{train}}}[(y - \widehat{p}(\mathbf{x}))^2]$ (due to Definition 4.5) and, since $\widehat{p}$ is the solution of the least squares program, $\mathbb{E}_{S_{\text{train}}}[(y - \widehat{p}(\mathbf{x}))^2] \leq \mathbb{E}_{S_{\text{train}}}[(y - p_{\text{low}}(\mathbf{x}))^2] \leq \mathbb{E}_{\mathcal{D}^{\text{train}}}[(y - p_{\text{low}}(\mathbf{x}))^2] + \frac{\epsilon\eta}{C} = O(\frac{\epsilon\eta}{C}) + O(\lambda)$. $\quad\square$

*Remark* 4.7. In a semi-agnostic setting where $\lambda$ is known, then $\eta$ can be chosen to balance the error and rejection rates in Theorem 4.6, obtaining bounds of $O(\sqrt{\lambda})$, which is known to be best-possible in the PQ setting, even for contrived concept classes (see [GKKM20]).

By combining Theorem 4.6 with bounds on the $\mathcal{L}_2$ sandwiching degree of fundamental concept classes by [KSV24b] (see Appendix A.5), we obtain the results of Table 2 for PQ learning.

## 5 Tolerant TDS Learning

Another approach to provide provable learning guarantees with distribution shift is to enable rejecting the whole test distribution. The formal definition of this setting was given by [KSV24b], but the proposed algorithms were allowed to reject even if a miniscule amount of distribution shift was detected. We provide the first TDS learners that are guaranteed to accept whenever the test marginal $\mathcal{D}'$ is close to the training marginal $\mathcal{D}$ in total variation distance (see Definition A.7).

**Definition 5.1** (Tolerant TDS Learning, extension of [KSV24b]). Let $\mathcal{F}$ be a concept class over $\mathcal{X} \subseteq \mathbb{R}^d$ and $\mathcal{D}$ a distribution over $\mathcal{X}$. The algorithm $\mathcal{A}$ is a TDS-learner for $\mathcal{F}$ with respect to $\mathcal{D}$ up to error $\gamma$ (which is, in general a function of parameter $\lambda$), tolerance $\theta$ and probability of failure $\delta$ if, upon receiving $m_{\text{train}}$ labeled samples from a training distribution $\mathcal{D}^{\text{train}}$ with $\mathcal{X}$-marginal $\mathcal{D}$ and $m_{\text{test}}$ unlabeled samples from a test distribution $\mathcal{D}^{\text{test}}$, algorithm $\mathcal{A}$ either rejects or accepts and outputs a hypothesis $h : \mathcal{X} \to \{\pm 1\}$ such that, with probability at least $1 - \delta$, the following hold:

(a) *(soundness)* Upon acceptance, the test error is bounded as $\mathbb{P}_{(\mathbf{x},y)\sim\mathcal{D}^{\text{test}}}[y \neq h(\mathbf{x})] \leq \gamma$.

(b) *(completeness)* If $\text{d}_{\text{TV}}(\mathcal{D}, \mathcal{D}_{\mathcal{X}}^{\text{test}}) \leq \theta$, then the algorithm accepts.

We first observe that tolerant TDS learning is implied by PQ learning.

**Proposition 5.2** (PQ implies Tolerant TDS Learning, modification of Proposition 56 in [KSV24b]). *Suppose that there is an algorithm $\mathcal{A}$ that PQ learns the class $\mathcal{F}$ with respect to $\mathcal{D}$ up to error $\gamma$, rejection rate $\eta$ and failure probability $\delta$. Then, for any $\epsilon, \theta \in (0, 1)$ there is an algorithm that TDS learns $\mathcal{F}$ with respect to $\mathcal{D}$ up to error $\gamma + \eta + \theta + \epsilon$, with tolerance $\theta$ and failure probability $\delta$ that calls $\mathcal{A}$ once and uses $O(\frac{1}{\epsilon^2} \log(1/\delta))$ additional samples and evaluations of the selector given by $\mathcal{A}$.*

The above result readily follows by two simple observations about the selector $g$ and the hypothesis $h$ in the output of a PQ learner. In particular, we have $\mathbb{P}_{\mathbf{x}\sim\mathcal{D}_{\mathcal{X}}^{\text{test}}}[g(\mathbf{x})=0] \leq \mathbb{P}_{\mathbf{x}\sim\mathcal{D}}[g(\mathbf{x})=0] + \mathrm{d}_{\text{TV}}(\mathcal{D},\mathcal{D}_{\mathcal{X}}^{\text{test}}) \leq \eta + \mathrm{d}_{\text{TV}}(\mathcal{D},\mathcal{D}_{\mathcal{X}}^{\text{test}})$ and $\mathrm{err}(h;\mathcal{D}^{\text{test}}) \leq \mathbb{P}_{\mathbf{x}\sim\mathcal{D}_{\mathcal{X}}^{\text{test}}}[g(\mathbf{x})=0] + \mathbb{P}_{(\mathbf{x},y)\sim\mathcal{D}^{\text{test}}}[y \neq h(\mathbf{x}), g(\mathbf{x})=1]$. The TDS learner will reject if the empirical estimate of $\mathbb{P}_{\mathbf{x}\sim\mathcal{D}_{\mathcal{X}}^{\text{test}}}[g(\mathbf{x})=0]$ is larger than $\eta + \theta + \Omega(\epsilon)$; otherwise it will output $h$.

Proposition 5.2 allows us to conclude that the realizable tolerant TDS learning algorithm in Table 1 follows from the PQ learning algorithm of Theorem 4.3.

In the agnostic setting, however, the error rate of PQ learning is known to be necessarily high (i.e., $\Omega(\sqrt{\lambda})$) even for very simple classes (see Remark 4.7). Therefore, the corresponding TDS learning results implied by Proposition 5.2 do not achieve the optimum error rate for the case of TDS learning (i.e. $\Theta(\lambda)$). Nevertheless, we are able to use, once more, our outlier removal process directly and obtain the following analogue of Theorem 4.6 for tolerant TDS learning.

**Theorem 5.3** (Tolerant TDS Learning via Sandwiching). *For $\epsilon,\theta,\delta \in (0,1)$, let $\mathcal{X} \subseteq \mathbb{R}^d$ and $(\mathcal{D},\mathcal{F})$ be an $(\frac{\epsilon}{C},\frac{\delta}{C},k,m)$-reasonable pair (Definition 4.5) for some sufficiently large universal constant $C > 0$. Then, there is an TDS learner $\mathcal{F}$ with respect to $\mathcal{D}$ up to error $O(\lambda) + 2\theta + \epsilon$, tolerance $\theta$ and probability of failure $\delta$ with sample and time complexity $\mathrm{poly}(\frac{m}{\epsilon}(kd)^k \log(1/\delta))$.*

Furthermore, (via Remark 3.3) we show that our tolerant TDS learning algorithm will with high probability be guaranteed to accept a distribution $\mathcal{D}'$ that is $1/2$-smooth with respect to $\mathcal{D}$.

For the full proof, see Appendix C.3. As a corollary of Theorem 5.3, we obtain the results of Table 2 for tolerant TDS learning.

**Limitations, Broader Impacts, and Future Work.** Our current results hold only for a limited class of training marginal distributions (e.g., standard Gaussian or uniform over the hypercube). We leave it as an interesting open question to relax these distributional assumptions, as well as expand the completeness criterion for our tolerant TDS learning algorithms to accept when the test marginal is close to any distribution among the members of some wide class of well-behaved distributions (i.e., satisfy the universality condition as defined by [GKSV23]). Additionally, we would like to point out that rejecting to predict on certain distributions may lead to unfair or biased predictions if the distributions overlap significantly with the minority groups.

**Acknowledgements.** We thank the anonymous reviewers of NeurIPS 2024 for their constructive feedback. We thank Adam Klivans for insightful conversations and helpful references. K.S. thanks Aravind Gollakota for useful discussions regarding PQ learning of homogeneous halfspaces and the observation that no unlabeled test examples are needed in this case. A.V. thanks Shyam Narayanan for helpful conversations about robust mean estimation and outlier removal.

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

# A Extended Preliminaries

## A.1 Notation

We use $\mathbf{x}, \mathbf{w}, \mathbf{v}$ to denote vectors in the $d$-dimensional Euclidean space $\mathbb{R}^d$. For some distribution $\mathcal{D}$ and a function $f$, we denote with $\mathbb{E}_{\mathbf{x} \sim \mathcal{D}}[f(\mathbf{x})]$ the expectation of the random variable $f(\mathbf{x})$ when $\mathbf{x}$ is drawn from $\mathcal{D}$. For a set of points $X$, we use a similar notation for the empirical expectations over $X$, i.e., $\mathbb{E}_{\mathbf{x} \sim X}[f(\mathbf{x})] = \frac{1}{|X|} \sum_{\mathbf{x} \in X} f(\mathbf{x})$. In our paper, $\mathcal{X} \subseteq \mathbb{R}^d$ will either be $\mathbb{R}^d$ or the hypercube $\{\pm 1\}^d$. We denote with $\mathbb{N}$ the set of natural numbers $\mathbb{N} = \{0, 1, 2, \dots\}$. The expression $\mathbf{x} \cdot \mathbf{w}$ or $\mathbf{x}^\top \mathbf{w}$ denotes the inner product between two vectors, i.e., $\mathbf{x} \cdot \mathbf{w} = \sum_{i=1}^{d} x_i w_i$ (where $x_i$ is the value of the $i$-th coordinate of $\mathbf{x}$).

## A.2 Polynomials

Throughout this work, we will refer to polynomials whose degree is at most $k$ as "degree-$k$ polynomials" for brevity. We will identify every degree-$k$ polynomial $p$ with the vector of its coefficients. Furthermore, for a vector $\mathbf{x}$ in $\mathbb{R}^d$ we will denote $\mathbf{x}^{\otimes k}$ the vector whose entries correspond to the values of all monomials of degree at most $k$ evaluated on $\mathbf{x}$. Both the vector corresponding to a degree-$k$ polynomial and the vector $\mathbf{x}^{\otimes k}$ have dimension $m$, where $m$ is the number of distinct monomials on $\mathbb{R}^d$ of degree at most $k$. Note that $m \leq d^k$ and that with this notation in hand we have $p(\mathbf{x}) = p \cdot (\mathbf{x}^{\otimes k}) = p^\top \mathbf{x}^{\otimes k}$.

## A.3 Learning theory

We will usually consider function classes over $\mathbb{R}^d$ taking values in $\{\pm 1\}$ or in $\{0, 1\}$. Consider the following definitions:

**Definition A.1.** A halfspace over $\mathbb{R}^d$ is a function mapping $\mathbf{x}$ in $\mathbb{R}^d$ to $\text{sign}(\mathbf{w} \cdot \mathbf{x} + \theta)$ for some $\mathbf{w}$ in $\mathbb{S}^{d-1}$ and $\theta$ in $\mathbb{R}$.

**Definition A.2.** A degree-$k$ polynomial threshold function (PTF) over $\mathbb{R}^d$ is a function mapping $\mathbf{x}$ in $\mathbb{R}^d$ to $\text{sign}(p(\mathbf{x}))$ for some degree-$k$ polynomial $p$.

**Definition A.3.** The OR of a collection of function classes $\mathcal{F}_1 \vee \cdots \vee \mathcal{F}_m$, is defined as the collection of functions $f$ defined as $f(\mathbf{x}) = f_1(\mathbf{x}) \vee \cdots \vee f_m(\mathbf{x})$ for each $f_i$ belonging to $\mathcal{F}_i$ respectively. Analogously, the AND of the collection of function classes $\mathcal{F}_1 \wedge \cdots \wedge \mathcal{F}_m$, is defined as the collection of functions $f$ defined as $f(\mathbf{x}) = f_1(\mathbf{x}) \wedge \cdots \wedge f_m(\mathbf{x})$ for each $f_i$ belonging to $\mathcal{F}_i$ respectively.

The following facts about VC dimensions of various classes are standard:

**Fact A.4.** *The VC dimension of halfspaces over $\mathbb{R}^d$ is at most $d + 1$, and the VC dimensions of degree-$k$ polynomial threshold functions is at most $d^k + 1$.*

**Fact A.5** (e.g. [VDVW09] and references therein.)**.** *Let $\{\mathcal{F}_1, \cdots, \mathcal{F}_m\}$ be a collection of function classes each of which has a VC dimension of $V$. Then, the VC dimension of $\mathcal{F}_1 \vee \cdots \vee \mathcal{F}_m$ and $\mathcal{F}_1 \wedge \cdots \wedge \mathcal{F}_m$ are at most $O(Vm \log m)$.*

We will also need the standard uniform convergence bound for function classes of bounded VC dimension.

**Fact A.6.** *Let $\mathcal{F}$ be a function class over $\mathbb{R}^d$ taking values in $\{\pm 1\}$ with VC dimension at most $V$, let $D$ be a distribution over $\mathbb{R}^d \times \{\pm 1\}$, and let $S \subset \mathbb{R}^d \times \{\pm 1\}$ be composed of $N$ i.i.d. examples from $D$. Then, with probability at least $1 - \delta$ we have*

$$\sup_{f \in \mathcal{F}} \left[ \left| \frac{1}{N} \sum_{(\mathbf{x}, y) \in S} [|f(\mathbf{x}) - y|] - \mathbb{E}_{(\mathbf{x}, y) \sim D}[|f(\mathbf{x}) - y|] \right| \right] \leq \sqrt{\frac{V \log N}{N} \log \frac{1}{\delta}}.$$

*The same statement also true if $\mathcal{F}$ is taking values in $\{0, 1\}$ and $D$ be a distribution over $\mathbb{R}^d \times \{0, 1\}$.*

## A.4 Properties of Distributions

We denote with $\mathcal{D}, \mathcal{D}'$ distributions over $\mathcal{X} \subseteq \mathbb{R}^d$. For two distributions $\mathcal{D}, \mathcal{D}'$, the total variation distance between them is defined as follows.

**Definition A.7.** Let $\mathcal{D}, \mathcal{D}'$ be distributions over $\mathcal{X} \subseteq \mathbb{R}^d$ (for some $\sigma$-algebra $\mathcal{B} \subseteq \mathrm{Pow}(\mathbb{R}^d)$). Then, the total variation distance between $\mathcal{D}$ and $\mathcal{D}'$ is

$$d_{\mathrm{TV}}(\mathcal{D}, \mathcal{D}') = \sup_{A \in \mathcal{B}} \left| \mathbb{P}_{\mathbf{x} \sim \mathcal{D}}[\mathbf{x} \in A] - \mathbb{P}_{\mathbf{x} \sim \mathcal{D}'}[\mathbf{x} \in A] \right|$$

We define the family of tame distributions as follows.

**Definition A.8.** A distribution $\mathcal{D}$ over $\mathcal{X}$ is $k$-tame if for every degree-$k$ polynomial $p$ over $\mathcal{X}$ with $\mathbb{E}_{\mathbf{x} \sim \mathcal{D}}[(p(\mathbf{x}))^2] \leq 1$ and every $B$ greater than $e^{2k}$ we have $\mathbb{P}_{\mathbf{x} \sim \mathcal{D}}[(p(\mathbf{x}))^2 > B] \leq e^{-\Omega(B^{\frac{1}{2k}})}$.

It is known that the standard Gaussian distribution, any log-concave distribution, as well as the uniform distribution over the hypercube $\{\pm 1\}^d$ are tame (see [DKS18] and references therein).

**Fact A.9.** *Let $\mathcal{D}$ be either the standard Gaussian distribution $\mathcal{N}$ or the uniform distribution over $\{\pm 1\}^d$, then, then $\mathcal{D}$ is $k$-tame for any $k \in \mathbb{N}$.*

**Fact A.10.** *Let $\mathcal{D}$ be a log-concave distribution over $\mathbb{R}^d$, then $\mathcal{D}$ is $k$-tame for any $k \in \mathbb{N}$.*

### A.5 Sandwiching Polynomials

We provide a formal definition of the $\mathcal{L}_2$ sandwiching property which is key in order to be able to apply our outlier removal process many of for our main learning applications.

**Definition A.11** ($\mathcal{L}_2$ Sandwiching)**.** For a concept class $\mathcal{F}$, a distribution $\mathcal{D}$ over $\mathcal{X}$, $\epsilon \in (0, 1)$, we say that $\mathcal{F}$ has $\epsilon$-$\mathcal{L}_2$ **sandwiching degree** $k$ with respect to $\mathcal{D}$ if for any $f \in \mathcal{F}$, there exist polynomials $p_{\mathrm{up}}, p_{\mathrm{low}}$ over $\mathcal{X}$ with degree at most $k$ such that (1) $p_{\mathrm{low}}(\mathbf{x}) \leq f(\mathbf{x}) \leq p_{\mathrm{up}}(\mathbf{x})$ for all $\mathbf{x} \in \mathcal{X}$ and (2) $\mathbb{E}_{\mathbf{x} \sim \mathcal{D}}[(p_{\mathrm{up}}(\mathbf{x}) - p_{\mathrm{low}}(\mathbf{x}))^2] \leq \epsilon$. If the coefficients of $p_{\mathrm{up}}, p_{\mathrm{low}}$ are all absolutely bounded by $B$, we say that $\mathcal{F}$ has $\epsilon$-$\mathcal{L}_2$ sandwiching coefficient bound $B$.

In order to obtain our learning results, in addition to $\mathcal{L}_2$ sandwiching, we use some further properties of the marginal distribution (see Definition 4.5). The following proposition from [KSV24b] shows that these properties are true for the Gaussian distribution as well as the uniform distribution over the hypercube $\{\pm 1\}^d$.

**Proposition A.12** (Appendix D in [KSV24b])**.** *Let $\mathcal{D}$ be either $\mathcal{N}_d$ or $\mathrm{Unif}(\{\pm 1\}^d)$ and let $\mathcal{F}$ be some concept class with $\epsilon$-$\mathcal{L}_2$ sandwiching degree $k$ with respect to $\mathcal{D}$. Then, $\mathcal{F}$ $\epsilon$-$\mathcal{L}_2$ sandwiching coefficient bound $B = d^{O(k)}$ and $(\mathcal{D}, \mathcal{F})$ is $(\epsilon, \delta, k, m)$-reasonable, where $m = O(\frac{1}{\delta})(dk)^{O(k)}$.*

Finally, we list a number of fundamental concept classes that are known to admit low-degree $\mathcal{L}_2$ sandwiching approximators.

**Lemma A.13** (Decision Trees, Lemma 34 in [KSV24b])**.** *Let $\mathcal{D}$ be the uniform distribution over the hypercube $\mathcal{X} = \{\pm 1\}^d$. For $s \in \mathbb{N}$, let $\mathcal{F}$ be the class of Decision Trees of size $s$. Then, for any $\epsilon > 0$ the $\epsilon$-$\mathcal{L}_2$ sandwiching degree of $\mathcal{F}$ is at most $k = O(\log(s/\epsilon))$.*

**Lemma A.14** (Boolean Formulas, Theorem 6 in [OS03] and Lemma 35 in [KSV24b])**.** *Let $\mathcal{D}$ be the uniform distribution over the hypercube $\mathcal{X} = \{\pm 1\}^d$. For $s, \ell \in \mathbb{N}$, let $\mathcal{F}$ be the class of Boolean formulas of size at most $s$, depth at most $\ell$. Then, for any $\epsilon > 0$ the $\epsilon$-$\mathcal{L}_2$ sandwiching degree of $\mathcal{F}$ is at most $k = (C \log(s/\epsilon))^{5\ell/2} \sqrt{s}$, for some sufficiently large universal constant $C > 0$.*

**Lemma A.15** (Intersections and Decision Trees of Halfspaces, Lemma 37 in [KSV24b])**.** *Let $\mathcal{D}$ be either the uniform distribution over the hypercube $\mathcal{X} = \{\pm 1\}^d$ or the Gaussian $\mathcal{N}_d$ over $\mathcal{X} = \mathbb{R}^d$. For $\ell \in \mathbb{N}$, let also $\mathcal{F}$ be the class of intersections of $\ell$ halfspaces on $\mathcal{X}$. Then, for any $\epsilon > 0$ the $\epsilon$-$\mathcal{L}_2$ sandwiching degree of $\mathcal{F}$ is at most $k = \widetilde{O}(\frac{\ell^6}{\epsilon^2})$. For Decision Trees of halfspaces of size $s$ and depth $\ell$, the bound is $k = \widetilde{O}(\frac{s^2 \ell^6}{\epsilon^2})$.*

### A.6 Other Learning Settings

Our approach provides new results even for standard learning scenarios (without distribution shift). In particular, we provide the first tolerant testable learning algorithms. In the testable learning setting, there is no distribution shift, but rather, the learner receives labeled examples from some distribution $\mathcal{D}_{\mathcal{X}\mathcal{Y}}$ over $\mathcal{X} \times \{\pm 1\}$ and is asked to either reject, or accept and output a hypothesis with low error on $\mathcal{D}_{\mathcal{X}\mathcal{Y}}$. The target error is $\mathrm{opt} + \epsilon$, where $\mathrm{opt} = \min_{f \in \mathcal{F}} \mathrm{err}(f; \mathcal{D}_{\mathcal{X}\mathcal{Y}})$. In order to obtain efficient

learners, we allow for the algorithm to reject if it detects that the marginal distribution $\mathcal{D}'$ of $\mathcal{D}_{\mathcal{X}\mathcal{Y}}$ on $\mathcal{X}$ is not equal to some given target distribution $\mathcal{D}$ which is known to be well-behaved. The tester is, once more, allowed to reject even when $\mathcal{D}'$ and $\mathcal{D}$ differ by a tiny amount. We are interested in tolerant testable learning, where the tester-learner is required to accept when $\mathcal{D}'$ is moderately close to $\mathcal{D}$ and provide the first upper bounds for the problem.

**Definition A.16** (Tolerant Testable Learning, extension of [RV23])**.** Let $\mathcal{F}$ be a concept class over $\mathcal{X} \subseteq \mathbb{R}^d$ and $\mathcal{D}$ a distribution over $\mathcal{X}$. The algorithm $\mathcal{A}$ is a tester-learner for $\mathcal{F}$ with respect to $\mathcal{D}$ up to error $\gamma$, tolerance $\theta$ and probability of failure $\delta$ if, upon receiving $m$ labeled samples from a distribution $\mathcal{D}_{\mathcal{X}\mathcal{Y}}$ with $\mathcal{X}$-marginal $\mathcal{D}'$, algorithm $\mathcal{A}$ either rejects or accepts and outputs a hypothesis $h : \mathcal{X} \to \{\pm 1\}$ such that, w.p. at least $1 - \delta$, the following hold:

(a) (*soundness*) Upon acceptance, the error is bounded as $\mathbb{P}_{(\mathbf{x},y)\sim\mathcal{D}_{\mathcal{X}\mathcal{Y}}}[y \neq h(\mathbf{x})] \leq \mathsf{opt} + \gamma$.

(b) (*completeness*) If $\mathrm{d}_{\mathrm{TV}}(\mathcal{D}, \mathcal{D}') \leq \theta$, then the algorithm accepts.

The optimum error opt is defined as $\mathsf{opt} = \min_{f \in \mathcal{F}} \mathbb{P}_{(\mathbf{x},y)\sim\mathcal{D}_{\mathcal{X}\mathcal{Y}}}[y \neq f(\mathbf{x})]$.

Finally, we give a definition for the model of learning with nasty noise (proposed by [BEK02]).

**Definition A.17** (Learning with Nasty Noise [BEK02])**.** Let $\mathcal{F}$ be a concept class over $\mathcal{X} \subseteq \mathbb{R}^d$ and $\mathcal{D}$ a distribution over $\mathcal{X}$. The algorithm $\mathcal{A}$ is a learner for $\mathcal{F}$ with respect to $\mathcal{D}$, robust under nasty noise with rate $\eta \in (0, 1)$, up to error $\gamma$ and probability of failure $\delta$ if the following hold. If the algorithm $\mathcal{A}$ receives a set of $N$ labeled samples $S$ that are formed by some adversary who first draws $N$ i.i.d. labeled samples $S_{\mathrm{iid}}$ from $\mathcal{D}$, labeled by some concept $f^* \in \mathcal{F}$ and then corrupts at most $\eta N$ (arbitrarily chosen) elements of $S_{\mathrm{iid}}$ and substitutes them by $\eta N$ arbitrary points of $\mathcal{X} \times \{\pm 1\}$, then $\mathcal{A}$ outputs w.p. at least $1 - \delta$ some hypothesis $h : \mathcal{X} \to \{\pm 1\}$ such that $\mathbb{P}_{\mathbf{x}\sim\mathcal{D}}[f^*(\mathbf{x}) \neq h(\mathbf{x})] \leq \gamma$.

The error $\gamma$ is a function of the noise rate $\eta$.

# B  Additional Tools

## B.1  Miscellaneous lemmas

In this section we present two technical lemmas used for the design and analysis of our filtering algorithm. The following lemma allows one to efficiently estimate the moments of a $k$-tame distribution and is used for the algorithms of Theorem E.1 and Theorem E.2.

---

**Algorithm 2:** Monomial Correlations Matrix Estimation

**Input:** Set $S$ of size $N$, containing points in $\mathcal{X} \subseteq \mathbb{R}^d$, parameter $\delta$ and parameter $k \in \mathbb{N}$
**Output:** A matrix $\widehat{M}$
Partition $S_{\mathcal{D}}$ into $\sqrt{N}$ parts $(S_{\mathcal{D}}^{(i)})_{i \in [\sqrt{N}]}$, each of size $\sqrt{N}$.
Compute the matrix $\widehat{M}_i = \mathbb{E}_{\mathbf{x}\sim S_{\mathcal{D}}^{(i)}}[(\mathbf{x}^{\otimes k})(\mathbf{x}^{\otimes k})^\top]$ for each $i \in [\sqrt{N}]$.
Let $\widehat{M} = \widehat{M}_i$ for some $i$ such that the number of indices $j \in [\sqrt{N}]$ for which the following condition holds

$$0.99 \cdot p^\top \widehat{M}_j p \leq p^\top \widehat{M}_i p \leq 1.01 \cdot p^\top \widehat{M}_j p, \text{ for all degree } k \text{ polynomials } p$$

is at least $0.8\sqrt{N}$, if such an index $i \in [\sqrt{N}]$ exists.
Otherwise, let $\widehat{M} = \widehat{M}_1$.

---

**Lemma B.1.** *For some sufficiently large absolute constant $C$, the following holds. There is an algorithm (Algorithm 2) that takes a parameter $\delta$ in $(0, 1)$, a positive integer $k$ and $N \geq C\left((kd)^k \log 1/\delta\right)^C$ i.i.d. examples from a $k$-tame distribution $\mathcal{D}$ over $\mathbb{R}^d$. The algorithm runs in time $\mathrm{poly}(N)$ and with probability at least $1 - \delta$ the outputs an $m \times m$ symmetric positive-semidefinite matrix $\widehat{M}$ that for every degree-$k$ polynomial $p$ satisfies*

$$\frac{9}{10} \mathbb{E}_{\mathbf{x}\sim\mathcal{D}}[(p(\mathbf{x}))^2] \leq p^\top \widehat{M} p \leq \frac{11}{10} \mathbb{E}_{\mathbf{x}\sim\mathcal{D}}[(p(\mathbf{x}))^2].$$

*Proof.* The run-time bound of $\text{poly}(N)$ is immediate. We now argue that the algorithm succeeds with probability at least $1 - \delta$ when the absolute constant $C$ is large enough. The matrix $\widehat{M}$ is symmetric positive-semidefinite because each matrix $M_i = \mathbb{E}_{\mathbf{x} \sim S_i}\left[(\mathbf{x}^{\otimes k})(\mathbf{x}^{\otimes k})^\top\right]$ is symmetric positive-semidefinite by construction.

Let $M$ denote the matrix $\mathbb{E}_{\mathbf{x} \sim \mathcal{D}}\left[(\mathbf{x}^{\otimes k})(\mathbf{x}^{\otimes k})^\top\right]$. Clearly $\mathbb{E}_{S_i}[M_i] = M$, and we will use the second moment method to bound the deviation, but first we observe that with probability 1 for every polynomial $p$ in the nullspace of $M$ we also have $p^\top \mathbb{E}_{S_i}[M_i]p = 0$. Indeed, this is the case because $p^\top M p = \mathbb{E}_{\mathbf{x} \sim \mathcal{D}}(p(\mathbf{x}))^2$ means that $p(\mathbf{x})^2 = 0$ almost surely for $\mathbf{x} \sim \mathcal{D}$. Thus, with probability 1 this holds for a collection of basis elements for the nullspace of $M$, and consequently for the entire nullspace of $M$.

Now, we bound the deviation between $M$ and $M_i$ for polynomials $p$ for which $p^\top M p > 0$. Let $m'$ be such that $m - m'$ is the dimension of the nullspace of $M$. Then there is a collection $\{r_1, \cdots, r_{m'}\}$ of degree-$k$ polynomials that satisfy

$$\mathbb{E}_{\mathbf{x} \sim \mathcal{D}}[r_j(\mathbf{x})r_{j'}(\mathbf{x})] = \begin{cases} 1 & \text{if } j = j' \\ 0 & \text{otherwise.} \end{cases}$$

(Such collection necessarily exists via the Gram-Schmidt process.) Overall, for any polynomial $p$

$$p^\top M p = \mathbb{E}_{\mathbf{x} \sim \mathcal{D}}[(p(\mathbf{x}))^2] = \sum_j \left(\mathbb{E}_{\mathbf{x} \sim \mathcal{D}}[p(\mathbf{x})r_j(\mathbf{x})]\right)^2$$

Therefore, denoting $\{e_1, \cdots, e_m\}$ the $m$ basis vectors in $\mathbb{R}^m$ we have

$$M^{1/2}p = \begin{bmatrix} \mathbb{E}_{\mathbf{x} \sim \mathcal{D}}[p(\mathbf{x})r_1(\mathbf{x})] \\ \mathbb{E}_{\mathbf{x} \sim \mathcal{D}}[p(\mathbf{x})r_2(\mathbf{x})] \\ \vdots \\ \mathbb{E}_{\mathbf{x} \sim \mathcal{D}}[p(\mathbf{x})r_{m'}(\mathbf{x})] \\ 0 \\ \cdots \\ 0 \end{bmatrix} \qquad M^{-1/2}\left(\sum_{i=1}^{m'} c_i e_i\right) = \sum_{i=1}^{m'} c_i r_i \qquad \text{(B.1)}$$

(Where $M^{-1/2}p$ is defined to be the Moore-Penrose pseudo-inverse of $M^{1/2}$ if $M$ is singular). We now bound the expected Frobenius norm:

$$\mathbb{E}_{S_i}\left[\left\|I - M^{-1/2}M_i M^{-1/2}\right\|_F^2\right] = \mathbb{E}_{S_i}\left[\sum_{j,j' \in \{1,\cdots,m\}} \left(e_j^\top \left(I - M^{-1/2}M_i M^{-1/2}\right)e_{j'}\right)^2\right] =$$

$$= \mathbb{E}_{S_i}\left[\sum_{j,j' \in \{1,\cdots,m\}} \left(\mathbf{1}_{j=j'} - r_j^\top M_i r_{j'}\right)^2\right] = \mathbb{E}_{S_i}\left[\sum_{j,j' \in \{1,\cdots,m\}} \left(\mathbb{E}_{\mathbf{x} \sim \mathcal{D}}[r_j(\mathbf{x})r_{j'}(\mathbf{x})] - r_j^\top M_i r_{j'}\right)^2\right] =$$

$$= \sum_{j,j'} \mathbb{E}_{S_i}\left(\mathbb{E}_{\mathbf{x} \sim \mathcal{D}}[r_j(\mathbf{x})r_{j'}(\mathbf{x})] - \mathbb{E}_{\mathbf{x} \sim S_i}[r_j(\mathbf{x})r_{j'}(\mathbf{x})]\right)^2 = \frac{1}{\sqrt{N}}\sum_{j_1,j_2 \in \{1,\cdots,m'\}} \text{Var}_{\mathbf{x} \sim \mathcal{D}}(r_j(\mathbf{x})r_{j'}(\mathbf{x}))$$

$$\text{(B.2)}$$

From the $k$-tameness of distribution $\mathcal{D}$, and the fact that $\mathbb{E}_{\mathbf{x} \sim \mathcal{D}}[(r_j(\mathbf{x}))^2] = 1$ we see that

$$\text{Var}_{\mathbf{x} \sim \mathcal{D}}(r_j(\mathbf{x})r_{j'}(\mathbf{x})) \leq \mathbb{E}_{\mathbf{x} \sim \mathcal{D}}\left[(r_j(\mathbf{x})r_{j'}(\mathbf{x}))^2\right] = \int_0^\infty \mathbb{P}\left[r_j(\mathbf{x})r_{j'}(\mathbf{x}))^2 > B^2\right]2B\, dB \leq$$

$$2e^{4k} + \int_{e^{2k}}^\infty e^{-\Omega(B^{1/(2k)})}2B\, dB = O(k^{O(k)}). \quad \text{(B.3)}$$

Thus, combining Equation B.2 and Equation B.3 and recalling that $M$ is an $m \times m$ matrix with $m \leq d^k$ we get a bound for the expected spectral norm of $M - M_i$:

$$\mathbb{E}_{S_i}\left[\left\|I - M^{-1/2}M_i M^{-1/2}\right\|_2\right] \leq \mathbb{E}_{S_i}\left[\left\|I - M^{-1/2}M_i M^{-1/2}\right\|_F\right] \leq$$

$$\sqrt{\mathbb{E}_{S_i}\left[\left\|I - M^{-1/2}M_i M^{-1/2}\right\|_F^2\right]} \leq \frac{1}{N^{1/4}}mO(k^{O(k)}) \leq \frac{1}{N^{1/4}}(dk)^{O(k)}$$

Recalling that $N \geq C\left((kd)^k \log 1/\delta\right)^C$, we see that for a sufficiently large absolute constant $C$ we can use the Chebyshev's inequality to conclude that with probability at least $0.9$ we have $\left\|I - M^{-1/2} M_i M^{-1/2}\right\|_2 \leq 10^{-3}$, which implies that for any $p$ of degree at most $k$ we have

$$p^\top M_i p = p^\top M p + p^\top (M_i - M)p = p^\top M p + p^\top M^{1/2}(M^{-1/2} M_i M^{-1/2} - I)M^{1/2}p$$
$$\in [(1 - 10^{-3})p^\top M p, (1 + 10^{-3})p^\top M p]$$

Recalling that $i$ is in $\{1, \cdots, \sqrt{N}\}$ and using the standard Hoeffding's inequality, we see that when $C$ is sufficiently large, with probability at least $1 - \delta$, the above holds for at least $0.95$ fraction of indices $i$. Call such indices good. For any pair of values $i_1, i_2$ of good indices we hence have

$$(1 - 10^{-2})p^\top M_{i_2} p \leq p^\top M_{i_1} p \leq (1 + 10^{-2})p^\top M_{i_2} p, \text{ for all } p$$

For any given indices, the above property can be checked by computing the maximum singular value of the matrix $M_{i_2}^{-1/2} M_{i_1} M_{i_2}^{-1/2}$ (where we take the Moore-Penrose pseudoinverse) and comparing the nullspaces of $M_{i_1}$ and $M_{i_2}$. Therefore, the output $i_1 = i^*$ satisfies the above property for at least $0.8$ fraction of the values of $i_2$, according to Algorithm 2. Moreover, $i^*$ satisfies the above property for at least one good index $i_2 = i_g$ (the fraction of good indices is $0.95$ and the property is satisfied for at least a $0.8$ fraction). Overall, we have that for all polynomials $p$

$$0.9p^\top M p \leq 0.99 p^\top M_{i_g} p \leq p^\top M_{i^*} p \leq 1.01 p^\top M_{i_g} p \leq 1.1 p^\top M p$$

which implies the correctness of the algorithm. $\qquad\square$

The following lemma allows one to show that as long as a distribution $\mathcal{D}'$ is filtered using a low-VC-dimension function $f$, the moments of the resulting filtered dataset approximate well the moments of the distribution one obtains by filtering the distribution $\mathcal{D}'$ using $f$.

**Lemma B.2.** *Let $\mathcal{D}$ be a probability distribution over $\mathbb{R}^d$ and let $\mathcal{F}$ be a function class over $\mathbb{R}^d$ taking values in $\{0, 1\}$ with VC dimension $V$, such that for every $f$ in $\mathcal{F}$ we have $f(\mathbf{x}) = 0$ for all $\mathbf{x}$ such that $\max\limits_{p:\ \mathbb{E}_{\mathbf{x} \sim \mathcal{D}}(p(\mathbf{x}))^2 \leq 1}(p(\mathbf{x}))^2 > B$. Let $\mathcal{D}'$ be a probability distribution over $\mathbb{R}^d$ and $S$ be collection of $N$ i.i.d. samples from $\mathcal{D}'$, then with probability at least $1 - \delta$ we have*

$$\sup_{\substack{f \in \mathcal{F},\ p\ s.t:\ \deg(p) \leq k, \\ \mathbb{E}_{\mathbf{x} \sim \mathcal{D}}(p(\mathbf{x}))^2 \leq 1}} \left| \frac{1}{N} \sum_{\mathbf{x} \sim S} \left[ f(\mathbf{x})(p(\mathbf{x}))^2 \right] - \mathbb{E}_{\mathbf{x} \sim \mathcal{D}'} \left[ f(\mathbf{x})(p(\mathbf{x}))^2 \right] \right| \leq O\left( B^{\frac{1}{2}} \left( V d^{2k} \frac{\log N}{N} \log \frac{1}{\delta} \right)^{\frac{1}{4}} \right)$$

*Proof.* Let $p$ be a polynomial of degree $k$ s.t: $\mathbb{E}_{\mathbf{x} \sim \mathcal{D}}(p(\mathbf{x}))^2 \leq 1$ and let $f$ be a function in $\mathcal{F}$. We recall that whenever $f(\mathbf{x}) \neq 0$ we have $\max\limits_{p:\ \mathbb{E}_{\mathbf{x} \sim \mathcal{D}}(p(\mathbf{x}))^2 \leq 1}(p(\mathbf{x}))^2 \leq B$ and we let $\Delta$ be a positive real number, to be chosen later. We then have via the triangle inequality

$$\left| \mathbb{E}_{\mathbf{x} \sim S} \left[ f(\mathbf{x})(p(\mathbf{x}))^2 \right] - \mathbb{E}_{\mathbf{x} \sim \mathcal{D}'} \left[ f(\mathbf{x})(p(\mathbf{x}))^2 \right] \right| \leq$$

$$\sum_{j=0}^{B/\Delta} \left| \mathbb{E}_{\mathbf{x} \sim S} \left[ f(\mathbf{x})(p(\mathbf{x}))^2 \mathbf{1}_{j\Delta \leq (p(\mathbf{x}))^2 < (j+1)\Delta} \right] - \mathbb{E}_{\mathbf{x} \sim \mathcal{D}'} \left[ f(\mathbf{x})(p(\mathbf{x}))^2 \mathbf{1}_{j\Delta \leq (p(\mathbf{x}))^2 < (j+1)\Delta} \right] \right| \leq$$

$$\sum_{j=0}^{B/\Delta} \left| \mathbb{E}_{\mathbf{x} \sim S} \left[ f(\mathbf{x})j\Delta \mathbf{1}_{j\Delta \leq (p(\mathbf{x}))^2 < (j+1)\Delta} \right] - \mathbb{E}_{\mathbf{x} \sim \mathcal{D}'} \left[ f(\mathbf{x})j\Delta \mathbf{1}_{j\Delta \leq (p(\mathbf{x}))^2 < (j+1)\Delta} \right] \right| +$$

$$+ \Delta \left( \sum_{j=0}^{B/\Delta} \mathbb{E}_{\mathbf{x} \sim S} \left[ f(\mathbf{x}) \mathbf{1}_{j\Delta \leq (p(\mathbf{x}))^2 < (j+1)\Delta} \right] + \mathbb{E}_{\mathbf{x} \sim \mathcal{D}'} \left[ f(\mathbf{x}) \mathbf{1}_{j\Delta \leq (p(\mathbf{x}))^2 < (j+1)\Delta} \right] \right) \leq$$

$$2\Delta + B \sum_{j=0}^{B/\Delta} \left| \mathbb{P}_{\mathbf{x} \sim S} \left[ (f(\mathbf{x}) = 1) \wedge \left( p(\mathbf{x})^2 \in \left( j\Delta, (j+1)\Delta \right] \right) \right] - \right.$$

$$\left. - \mathbb{P}_{\mathbf{x} \sim \mathcal{D}'} \left[ (f(\mathbf{x}) = 1) \wedge \left( p(\mathbf{x})^2 \in \left( j\Delta, (j+1)\Delta \right] \right) \right] \right| \quad \text{(B.4)}$$

The function that maps $\mathbf{x}$ to 1 if and only if $f(\mathbf{x}) = 1$ and $p(\mathbf{x})^2 \in (j\Delta, (j+1)\Delta]$ is a logical AND of a function in $\mathcal{F}$ and two polynomial threshold functions of degree at most $2k$. Thus, by Fact A.5 the VC dimension of these functions is at most $O(d^{2k} + V) \leq O(d^{2k} \cdot V)$. Therefore, we can use Fact A.6 together with the inequality above to conclude that with probability at least $1 - \delta$

$$\sup_{\substack{f \in \mathcal{F}, \ p \text{ of degree } k \text{ s.t:} \\ \mathbb{E}_{\mathbf{x} \sim \mathcal{D}}(p(\mathbf{x}))^2 \leq 1}} \left| \mathbb{E}_{\mathbf{x} \sim S}\left[ f(\mathbf{x})(p(\mathbf{x}))^2 \right] - \mathbb{E}_{\mathbf{x} \sim \mathcal{D}'}\left[ f(\mathbf{x})(p(\mathbf{x}))^2 \right] \right| \leq 2\Delta + O\left( \frac{B}{\Delta} \sqrt{\frac{V d^{2k} \log N}{N}} \log \frac{1}{\delta} \right).$$

Finally, taking $\Delta$ to minimize the expression above, we recover our proposition. $\qquad\square$

## C  Certified Learning with Distribution Shift, Omitted Details

### C.1  PQ Setting

#### C.1.1  General Halfspaces

We now prove Theorem 4.3, which is restated here for convenience.

**Theorem C.1** (PQ Learning of Halfspaces). *For any $\epsilon, \delta \in (0,1)$, there is an algorithm that PQ learns the class of general halfspaces with respect to $\mathcal{N}_d$ in the realizable setting, up to error and rejection rate $\epsilon$ and probability of failure $\delta$ that runs in time $\mathrm{poly}(d^{\log(\frac{1}{\epsilon})}, \log(1/\delta))$.*

*Proof.* The algorithm does the following for sufficiently large universal constants $C_1, C_2, C_3 \geq 1$.

1. Compute the values $\mathbb{P}_{(\mathbf{x},y) \sim S_{\mathrm{train}}}[y = 1]$ and $\mathbb{P}_{(\mathbf{x},y) \sim S_{\mathrm{train}}}[y = -1]$.

2. If either of these values is at most $\epsilon^{C_2}/C_1$, then let $g$ be the selector of Theorem 3.1 with inputs $\epsilon$, $k = C_3 \log(1/\epsilon)$, $\alpha = \epsilon/2$ and access to samples from $\mathcal{D}$ and $\mathcal{D}'$ and $h$ the constant hypothesis for the value in $\{-1, 1\}$ with which the labels are most frequently consistent.

3. Otherwise, let $\widehat{\mathbf{w}}$ and $\widehat{\tau}$ be as in Proposition 4.4 from some sufficiently large labeled sample from the training distribution. Let $h(\mathbf{x}) = \mathrm{sign}(\widehat{\mathbf{w}} \cdot \mathbf{x} + \widehat{\tau})$ and for $\mathcal{W} = \{(\mathbf{w}, \tau) : \|\mathbf{w} - \widehat{\mathbf{w}}\|_2 \leq (\epsilon/d)^{C_2}/C_1, |\tau - \widehat{\tau}| \leq (\epsilon/d)^{C_2}/C_1\}$, let $g(\mathbf{x})$ (where $g : \mathbb{R}^d \to \{0, 1\}$) be 0 if and only if there are $(\mathbf{w}_1, \tau_1), (\mathbf{w}_2, \tau_2) \in \mathcal{W}$ such that $\mathrm{sign}(\mathbf{w}_1 \cdot \mathbf{x} + \tau_1) \neq \mathrm{sign}(\mathbf{w}_2 \cdot \mathbf{x} + \tau_2)$ (which can be implemented via a linear program with quadratic constraints).

4. Return $(g, h)$.

Note that when step 3 is activated, then, with high probability, we have that the bias $\tau^*$ of the ground truth is $\tau^* = O(\sqrt{\log(1/\epsilon)})$ and, therefore, the samples required to apply Proposition 4.4 is polynomial in $1/\epsilon$. From Lemma 5.7 in [KSV24b], we then have that the selector $g$ has Gaussian rejection rate $\epsilon$, as desired. The accuracy guarantee for the case of step 3 is given by the guarantee of Proposition 4.4 combined with the fact that, with high probability, $h$ agrees with the ground truth anywhere outside the disagreement region (i.e., for all $\mathbf{x}$ such that $g(\mathbf{x}) = 1$).

When step 2 is activated, then the rejection rate is bounded by Theorem 3.1 and the accuracy guarantee is implied by the fact that $\tau^* \geq \sqrt{C_2 \log(1/\epsilon)}/C$ (for some universal constant $C \geq 1$) and the following reasoning, where we suppose, without loss of generality that $h(\mathbf{x}) = 1$.

$$\begin{aligned}
\mathbb{P}_{\mathbf{x} \sim \mathcal{D}_{\mathcal{X}}^{\mathrm{test}}}[h(\mathbf{x}) \neq \mathrm{sign}(\mathbf{w}^* \cdot \mathbf{x} + \tau^*), g(\mathbf{x}) = 1] &\leq \mathbb{P}_{\mathbf{x} \sim \mathcal{D}_{\mathcal{X}}^{\mathrm{test}}}[1 \neq \mathrm{sign}(\mathbf{w}^* \cdot \mathbf{x} + \tau^*), g(\mathbf{x}) = 1] \\
&\leq \mathbb{P}_{\mathbf{x} \sim \mathcal{D}_{\mathcal{X}}^{\mathrm{test}}}[|\mathbf{w}^* \cdot \mathbf{x}| > |\tau^*|, g(\mathbf{x}) = 1] \\
&\leq \frac{\mathbb{E}_{\mathbf{x} \sim \mathcal{D}_{\mathcal{X}}^{\mathrm{test}}}[(\mathbf{w}^* \cdot \mathbf{x})^{2k} g(\mathbf{x})]}{(\tau^*)^{2k}} \\
&\leq \frac{400}{\epsilon} \frac{(2C_3 \log(1/\epsilon))^k}{(C_2 \log(1/\epsilon)/C)^k},
\end{aligned}$$

where we used Markov's inequality and the guarantee from Theorem 3.1. Suppose that $C_2$ is sufficiently larger than $C_3$ and $C_3$ is sufficiently large. Then we have that $\frac{(2C_3 \log(1/\epsilon))^k}{(C_2 \log(1/\epsilon)/C)^k} \leq$

$(1/2)^k \leq \epsilon^{C_3}$, which gives that $\mathbb{P}_{\mathbf{x} \sim \mathcal{D}_{\mathcal{X}}^{\text{test}}}[h(\mathbf{x}) \neq \text{sign}(\mathbf{w}^* \cdot \mathbf{x} + \tau^*), g(\mathbf{x}) = 1] \leq 400\epsilon^{C_3}/\epsilon \leq \epsilon$ for sufficiently large $C_3$. $\qquad\square$

In the proof of Theorem 4.3, we use Proposition 4.4. However, the original version of Proposition 4.4 worked for constant probability of failure. We show the following general lemma which can be used to amplify the probability of success in logarithmic number of rounds.

**Lemma C.2** (Parameter Recovery Success Probability Amplification). *Let $\mathcal{X}$ be a vector space and $\|\cdot\|$ some norm. For some $\mathbf{w}^* \in \mathcal{X}$ and $\epsilon \in (0,1)$, suppose that an algorithm $\mathcal{A}$ outputs with probability at least $0.9$ some $\mathbf{w} \in \mathcal{X}$ with $\|\mathbf{w}^* - \mathbf{w}\| \leq \epsilon$. Then, for any $\delta \in (0,1)$, if we run $\mathcal{A}$ for $T = O(\log(1/\delta))$ independent rounds receiving outputs $W = \{\mathbf{w}^1, \mathbf{w}^2, \ldots, \mathbf{w}^\top\}$, and take $\widehat{\mathbf{w}} = \arg\min_{\mathbf{w} \in W} \sum_{\mathbf{w}' \in W} \|\mathbf{w} - \mathbf{w}'\|$, then we have $\|\widehat{\mathbf{w}} - \mathbf{w}^*\| \leq 5\epsilon$, with probability at least $1 - \delta$.*

*Proof.* Let $W_G$ be the subset of $W$ corresponding to $\mathbf{w}$ such that $\|\mathbf{w} - \mathbf{w}^*\| \leq \epsilon$, let $W_B = W \setminus W_G$ and note that due to a standard Hoeffding bound, $|W_G| \geq \frac{3T}{4}$, with probability at least $1 - \delta$, for $T = C \log(1/\delta)$, where $C$ is a sufficiently large universal constant.

Say that $\|\widehat{\mathbf{w}} - \mathbf{w}^*\| = \alpha$. Then, by the triangle inequality, $\|\widehat{\mathbf{w}} - \mathbf{w}'\| \geq \|\widehat{\mathbf{w}} - \mathbf{w}^*\| - \|\mathbf{w}^* - \mathbf{w}'\| \geq \alpha - \epsilon$ for any $\mathbf{w}' \in W_G$ and we have the following

$$\widehat{A} := \sum_{\mathbf{w}' \in W} \|\widehat{\mathbf{w}} - \mathbf{w}'\| = \sum_{\mathbf{w}' \in W_G} \|\widehat{\mathbf{w}} - \mathbf{w}'\| + \sum_{\mathbf{w}' \in W_B} \|\widehat{\mathbf{w}} - \mathbf{w}'\|$$
$$\geq |W_G|(\alpha - \epsilon) + \sum_{\mathbf{w}' \in W_B} \|\widehat{\mathbf{w}} - \mathbf{w}'\|$$

Let $\mathbf{w} \in W_G$. We have $\|\mathbf{w} - \mathbf{w}'\| \leq \|\mathbf{w} - \widehat{\mathbf{w}}\| + \|\widehat{\mathbf{w}} - \mathbf{w}'\| \leq \|\mathbf{w} - \mathbf{w}^*\| + \|\mathbf{w}^* - \widehat{\mathbf{w}}\| + \|\widehat{\mathbf{w}} - \mathbf{w}'\| \leq \epsilon + \alpha + \|\widehat{\mathbf{w}} - \mathbf{w}'\|$, for any $\mathbf{w} \in W$. Therefore, in total, we have

$$A := \sum_{\mathbf{w}' \in W} \|\mathbf{w} - \mathbf{w}'\| = \sum_{\mathbf{w}' \in W_G} \|\mathbf{w} - \mathbf{w}'\| + \sum_{\mathbf{w}' \in W_B} \|\mathbf{w} - \mathbf{w}'\|$$
$$\leq 2\epsilon|W_G| + (\epsilon + \alpha)|W_B| + \sum_{\mathbf{w}' \in W_B} \|\widehat{\mathbf{w}} - \mathbf{w}'\|$$

By the definition of $\widehat{\mathbf{w}}$, we have that $\widehat{A} - A \leq 0$. With probability at least $1 - \delta$, we have

$$0 \geq \widehat{A} - A \geq |W_G|(\alpha - 3\epsilon) - (\alpha + \epsilon)(T - |W_G|)$$
$$\geq \frac{3T}{4}(\alpha - 3\epsilon) - (\alpha + \epsilon)\frac{T}{4}$$
$$\geq T(\frac{\alpha}{2} - \frac{5\epsilon}{2})$$

Therefore, $\alpha \leq 5\epsilon$, which concludes the proof. $\qquad\square$

## C.2 Adversarial Setting

Our results on realizable PQ learning extend to the following related model, where the evaluation set is formed by some adversary. In the following definition, we consider each element of the considered sets to be a separate object (even if the corresponding value is the same with some other element of the set).

**Definition C.3** (Tranductive Learning [GKKM20]). Let $\mathcal{F}$ be a concept class over $\mathcal{X} \subseteq \mathbb{R}^d$ and $\mathcal{D}$ a distribution over $\mathcal{X}$. The algorithm $\mathcal{A}$ is a transductive learner for $\mathcal{F}$ with respect to $\mathcal{D}$, up to error $\gamma$, rejection rate $\eta$ and probability of failure $\delta$ if the following hold. If the algorithm $\mathcal{A}$ has access to labeled examples from the distribution $\mathcal{D}$ labeled by some concept $f^* \in \mathcal{F}$ and receives $N$ unlabeled samples $S$ that are formed by some adversary who first draws $N$ i.i.d. unlabeled samples $S_{\text{iid}}$ from $\mathcal{D}$ and then corrupts any number of elements of $S_{\text{iid}}$ and substitutes them by the same number of arbitrary points of $\mathcal{X}$, then $\mathcal{A}$ outputs w.p. at least $1 - \delta$ some set $S_{\text{filt}}$ and $h : \mathcal{X} \to \{\pm 1\}$ such that:

(a) *(accuracy)* The error after filtering is bounded as $\sum_{(\mathbf{x}, y) \in S_{\text{filt}}} \mathbb{1}\{y \neq h(\mathbf{x})\} \leq \gamma N$.

(b) (*rejection rate*) The rejection rate of i.i.d. examples is $\#\{\mathbf{x} : \mathbf{x} \in S_{\mathrm{iid}} \cap (S \setminus S_{\mathrm{filt}})\} \leq \eta N$.

In particular, we have $\#\{\mathbf{x} : \mathbf{x} \in S \setminus S_{\mathrm{filt}}\} \leq 2\eta N$.

Our proofs of Theorems 4.3 and 4.6 generalize in this setting exactly analogously, with the only difference being the use of Theorem E.1 in place of Theorem 3.1 for the outlier removal process. The learning phase is not different, since the learner has sample access to clean examples from the labeled training distribution.

For the proof of Theorem 4.3, we either run the outlier removal process to filter the evaluation dataset in order to ensure that it is concentrated in every direction (in the case when almost all the training examples have the same label) or, if the training examples are indeed informative, we reject only the examples that fall inside the disagreement region. The arguments hold analogously.

For the proof of Theorem 4.6, we filter the evaluation dataset by using degree $k$ outlier removal (see Theorem E.1) and run polynomial regression on the training distribution to find a hypothesis that has low error on the remaining points of the evaluation dataset. Once more, the analysis is analogous to the one for PQ learning.

## C.3  Tolerant TDS Learning

We now prove Theorem 5.3, which we restate for convenience.

**Theorem C.4** (Tolerant TDS Learning via Sandwiching). *For $\epsilon, \theta, \delta \in (0,1)$, let $\mathcal{X} \subseteq \mathbb{R}^d$ and $(\mathcal{D}, \mathcal{F})$ be an $(\frac{\epsilon}{C}, \frac{\delta}{C}, k, m)$-reasonable pair (Definition 4.5) for some sufficiently large universal constant $C > 0$. Then, there is an TDS learner $\mathcal{F}$ with respect to $\mathcal{D}$ up to error $O(\lambda) + 2\theta + \epsilon$, tolerance $\theta$ and probability of failure $\delta$ with sample complexity $m + \mathrm{poly}(\frac{1}{\epsilon}(kd)^k \log(1/\delta))$ and time complexity $\mathrm{poly}(\frac{m}{\epsilon}(kd)^k \log(1/\delta))$.*

Note that the notion of tolerance in property testing was introduced in [PRR06] and has been the focus of many works including [FF05, VV11, BCE$^+$19, RV20, CJKL22, CFG$^+$22, BH18, CP23]. However, over $\mathbb{R}^d$ all existing tolerant distribution testing algorithms (such as [VV11]) have run-times and sample complexities of $2^{\Omega(d)}$, which greatly exceeds our run-times.

*Proof of Theorem 5.3.* The algorithm first runs the outlier removal process of Theorem 3.1 with parameters $\alpha \leftarrow 1$, $\epsilon \leftarrow \epsilon/C$, $\delta \leftarrow \delta/C$ and $k \leftarrow k$, to receive the selector $g : \mathcal{X} \rightarrow \{0,1\}$. Using a large enough sample from the test marginal $\mathcal{D}_{\mathcal{X}}^{\mathrm{test}}$, the algorithm estimates the quantity $\mathbb{P}_{\mathbf{x} \sim \mathcal{D}_{\mathcal{X}}^{\mathrm{test}}}[g(\mathbf{x}) = 0]$ and rejects if the estimated value is larger than $2\theta + \frac{2\epsilon}{C}$. Otherwise, it runs the following box-constrained least squares problem, using at least $m$ labeled examples $S_{\mathrm{train}}$ from $\mathcal{D}^{\mathrm{train}}$, where $t = d^k$ and $B$ is the value specified in Definition 4.5.

$$\min_p \mathbb{E}_{(\mathbf{x},y) \sim S_{\mathrm{train}}}[(y - p(\mathbf{x}))^2]$$

s.t. $p$ has degree at most $k$ and coefficient bound $B$

Let $\widehat{p}$ be the minimizer of the above program. The algorithm accepts and returns classifier $h(\mathbf{x}) = \mathrm{sign}(\widehat{p}(\mathbf{x}))$.

**Soundness** follows from the observation that $\mathbb{P}_{(\mathbf{x},y) \sim \mathcal{D}^{\mathrm{test}}}[y \neq h(\mathbf{x})] \leq \mathbb{P}_{\mathbf{x} \sim \mathcal{D}_{\mathcal{X}}^{\mathrm{test}}}[g(\mathbf{x}) = 0] + \mathbb{P}_{(\mathbf{x},y) \sim \mathcal{D}^{\mathrm{test}}}[y \neq h(\mathbf{x}), g(\mathbf{x}) = 1]$ and the properties of $g$ according to Theorem 3.1, via an analysis which is analogous to the one used for Theorem 4.6, but with the difference that, since the parameter $\alpha$ of the outlier removal process was chosen to be 1, the value $\mathbb{P}_{(\mathbf{x},y) \sim \mathcal{D}^{\mathrm{test}}}[y \neq h(\mathbf{x}), g(\mathbf{x}) = 1]$ is bounded by $O(\lambda + \frac{\epsilon}{C})$ (instead of $O(\frac{\lambda}{\eta})$). The term $\mathbb{P}_{\mathbf{x} \sim \mathcal{D}_{\mathcal{X}}^{\mathrm{test}}}[g(\mathbf{x}) = 0]$, when the test has accepted, is bounded by $2\theta + \epsilon/2$.

For **completeness**, we assume that $\mathrm{d}_{\mathrm{TV}}(\mathcal{D}, \mathcal{D}_{\mathcal{X}}^{\mathrm{test}}) \leq \theta$ and observe that due to condition (b), $\mathbb{P}_{\mathbf{x} \sim \mathcal{D}_{\mathcal{X}}^{\mathrm{test}}}[g(\mathbf{x}) = 0] \leq \mathbb{P}_{\mathbf{x} \sim \mathcal{D}}[g(\mathbf{x}) = 0] + \mathrm{d}_{\mathrm{TV}}(\mathcal{D}, \mathcal{D}_{\mathcal{X}}^{\mathrm{test}}) \leq 2\mathrm{d}_{\mathrm{TV}}(\mathcal{D}, \mathcal{D}_{\mathcal{X}}^{\mathrm{test}}) + \frac{\epsilon}{C} \leq 2\theta + \frac{\epsilon}{C}$. Therefore, the tester will accept with high probability. Furthermore, via Remark 3.3 our tolerant TDS learning algorithm will also with high probability accept any distribution $\mathcal{D}'$ that is $1/2$-smooth with respect to $\mathcal{D}$. $\qquad\square$

# D  Applications to Other Learning Settings

Our techniques provide new results in other classical settings as well. In particular, we discuss applications on tolerant testable learning as well as robust learning. In total, our results for polynomial regression can be summarized in Table 2.

| | Concept class $\mathcal{F}$ | Training Marginal $\mathcal{D}$ | Run-time |
|---|---|---|---|
| 1 | Intersections of $\ell$ halfspaces | Standard Gaussian 
 Uniform on $\{\pm 1\}^d$ | $d^{\widetilde{O}(\ell^6/\sigma^2)}$ |
| 2 | Functions of $\ell$ halfspaces | Standard Gaussian 
 Uniform on $\{\pm 1\}^d$ | $d^{\widetilde{O}(4^\ell \ell^6/\sigma^2)}$ |
| 3 | Decision trees of size $s$ | Uniform on $\{\pm 1\}^d$ | $d^{O(\log(s/\sigma))}$ |
| 4 | Formulas of size $s$, depth $\ell$ | Uniform on $\{\pm 1\}^d$ | $d^{\sqrt{s} \cdot O(\log(s/\sigma))^{\frac{5\ell}{2}}}$ |

Table 2: Our learning results parameterized by $\sigma$, which captures the required precision of the $\mathcal{L}_2$-sandwiching approximators in each of the settings: (1) agnostic PQ learning with error $O(\frac{\lambda}{\eta}) + \epsilon$ and rejection rate $\eta$, where $\sigma = \epsilon\eta$, (2) agnostic $\theta$-tolerant TDS learning with error $O(\lambda) + 2\theta + \epsilon$, where $\sigma = \epsilon$, (3) $\theta$-tolerant testable learning with excess error $2\theta + \epsilon$, where $\sigma = \epsilon^2$ and (4) robust learning with nasty noise of rate $\eta$ up to error $4\eta + \epsilon$, where $\sigma = \epsilon^2$. The probability of failure in each of the cases is some considered constant and $\eta, \theta, \epsilon \in (0, 1)$.

## D.1  Classical Testable learning

The following theorem gives the first dimension-efficient algorithms for tolerant testable learning (see Definition A.16) for various important concept classes.

**Theorem D.1** (Tolerant Testable Learning via Sandwiching). *For $\epsilon, \theta, \delta \in (0, 1)$, let $\mathcal{X} \subseteq \mathbb{R}^d$ and $(\mathcal{D}, \mathcal{F})$ be an $(\frac{\epsilon^2}{C}, \frac{\delta}{C}, k, m)$-reasonable pair (Definition 4.5) for some sufficiently large universal constant $C > 0$. Then, there is a tester-learner for $\mathcal{F}$ with respect to $\mathcal{D}$ up to error $\mathsf{opt} + 2\theta + \epsilon$, tolerance $\theta$ and probability of failure $\delta$ with sample complexity $m + \mathrm{poly}(\frac{1}{\epsilon}(kd)^k \log(1/\delta))$ and time complexity $\mathrm{poly}(\frac{m}{\epsilon}(kd)^k \log(1/\delta))$, where $\mathsf{opt} = \min_{f \in \mathcal{F}} \mathrm{err}(f)$.*

Our plan is to once more make use of the outlier removal Theorem 3.1. In this case, is suffices to run tests on the marginal distribution that certify the existence of a low-degree polynomial approximator for the unknown ground truth concept (achieving optimum error), due to the following classical result from [KKMS08] (which has been used for non-tolerant testable learning in [GKK23]).

**Proposition D.2** ($\mathcal{L}_1$ regression guarantee, [KKMS08]). *Let $\mathcal{F}$ be a concept class over $\mathcal{X}$ where $\mathcal{X} \subseteq \mathbb{R}^d$ and $\mathcal{D}_{\mathcal{X}\mathcal{Y}}$ be any distribution over $\mathcal{X} \times \{\pm 1\}$ where the $\mathcal{X}$-marginal of $\mathcal{D}_{\mathcal{X}\mathcal{Y}}$ is $\mathcal{D}'$. For $\epsilon \in (0, 1)$ and $k \in \mathbb{N}$, suppose that for any $f \in \mathcal{F}$ there is some polynomial $p$ over $\mathcal{X}$ of degree at most $k$ such that $\mathbb{E}_{\mathbf{x}\sim\mathcal{D}'}[|f(\mathbf{x}) - p(\mathbf{x})|] \le \epsilon$. Then, there is an algorithm ($\mathcal{L}_1$ polynomial regression) that, upon receiving a number of i.i.d. samples from $\mathcal{D}_{\mathcal{X}\mathcal{Y}}$, outputs with probability at least $1 - \delta$ some hypothesis $h : \mathcal{X} \to \{\pm 1\}$ with $\mathbb{P}_{(\mathbf{x},y)\sim\mathcal{D}_{\mathcal{X}\mathcal{Y}}}[y \ne h(\mathbf{x})] \le \mathsf{opt} + O(\epsilon)$, where $\mathsf{opt} = \min_{f \in \mathcal{F}} \mathrm{err}(f; \mathcal{D}_{\mathcal{X}\mathcal{Y}})$. The algorithm uses $\mathrm{poly}(d^k, \frac{1}{\epsilon}) \log(1/\delta)$ time and samples.*

The tester of Theorem D.1 does the following for some sufficiently large universal constant $C \ge 1$.

1. Runs the outlier removal of Theorem 3.1 with parameters $\alpha \leftarrow 1$, $\epsilon \leftarrow \epsilon/C$, $\delta \leftarrow \delta/C$ to receive a selector $g$ with the guarantees specified in Theorem 3.1.

2. Estimates, using unlabeled samples form $\mathcal{D}_{\mathcal{X}\mathcal{Y}}$, the value of $\mathbb{P}_{\mathbf{x}\sim\mathcal{D}'}[g(\mathbf{x}) = 0]$ and rejects if the estimated value is greater than $2\theta + 2\epsilon/C$.

3. Otherwise, the tester accepts and runs the algorithm of Proposition D.2 with fresh samples from the distribution $\tilde{\mathcal{D}}_{\mathcal{X}\mathcal{Y}}$ that corresponds to the conditioning of $\mathcal{D}_{\mathcal{X}\mathcal{Y}}$ to $g(\mathbf{x}) = 1$, with parameters $\epsilon \leftarrow \epsilon/C$ and $k \leftarrow k$.

Without loss of generality, we have that $2\theta + \epsilon \le 1/2$ (otherwise, we may output a random hypothesis). This implies that the runtime does not change asymptotically by conditioning on $g(\mathbf{x}) = 1$ (which can be done through rejection sampling).

For **completeness**, we observe that, by condition (b), $\mathbb{P}_{\mathbf{x}\sim\mathcal{D}}[g(\mathbf{x}) = 0] \le \mathrm{d}_{\mathrm{TV}}(\mathcal{D}, \mathcal{D}') + \frac{\epsilon}{C}$ and hence $\mathbb{P}_{\mathbf{x}\sim\mathcal{D}'}[g(\mathbf{x}) = 0] \le 2\mathrm{d}_{\mathrm{TV}}(\mathcal{D}, \mathcal{D}') + \frac{\epsilon}{C}$. By a standard Hoeffding bound, we have that the estimated value for $\mathbb{P}_{\mathbf{x}\sim\mathcal{D}'}[g(\mathbf{x}) = 0]$ (obtained using unlabeled samples from the marginal $\mathcal{D}'$ of $\mathcal{D}_{\mathcal{X}\mathcal{Y}}$) is at most $2\mathrm{d}_{\mathrm{TV}}(\mathcal{D}, \mathcal{D}') + \frac{2\epsilon}{C}$ and the tester will, with high probability accept if $\mathrm{d}_{\mathrm{TV}}(\mathcal{D}, \mathcal{D}') \le \theta$.

For **soundness**, we want to show that, upon acceptance, for any $f \in \mathcal{F}$, there is a polynomial $p$ of degree $k$ such that $\mathbb{E}_{\mathbf{x}\sim\tilde{\mathcal{D}}}[|f(\mathbf{x}) - p(\mathbf{x})|] \le O(\frac{\epsilon}{C})$. Then, by Proposition D.2, we have that the output $h$ satisfies $\mathrm{err}(h; \tilde{\mathcal{D}}_{\mathcal{X}\mathcal{Y}}) \le \min_{f\in\mathcal{F}} \mathrm{err}(f; \tilde{\mathcal{D}}_{\mathcal{X}\mathcal{Y}}) + O(\epsilon/C)$. Moreover we would also have $\mathrm{err}(h; \mathcal{D}_{\mathcal{X}\mathcal{Y}}) \le \mathbb{P}_{\mathbf{x}\sim\mathcal{D}'}[g(\mathbf{x}) = 0] + \mathbb{P}_{\mathbf{x}\sim\mathcal{D}'}[g(\mathbf{x}) = 1]\mathrm{err}(h; \tilde{\mathcal{D}}_{\mathcal{X}\mathcal{Y}})$, by the law of total probability. The second term of the sum can be bounded as $\mathbb{P}_{\mathbf{x}\sim\mathcal{D}'}[g(\mathbf{x}) = 1]\mathrm{err}(h; \tilde{\mathcal{D}}_{\mathcal{X}\mathcal{Y}}) \le \min_{f\in\mathcal{F}} \mathbb{P}_{\mathbf{x}\sim\mathcal{D}'}[g(\mathbf{x}) = 1]\mathbb{P}_{(\mathbf{x},y)\sim\mathcal{D}_{\mathcal{X}\mathcal{Y}}}[y \ne h(\mathbf{x})|g(\mathbf{x}) = 1] + O(\epsilon/C) = \mathsf{opt} + O(\epsilon/C)$. Overall, the bound on $\mathrm{err}(h; \mathcal{D}_{\mathcal{X}\mathcal{Y}})$ would then be $\mathsf{opt} + 2\theta + \epsilon$, because after acceptance, $\mathbb{P}_{\mathbf{x}\sim\mathcal{D}'}[g(\mathbf{x}) = 0]$ is bounded, with high probability, by $2\theta + O(\epsilon/C)$.

It remains to show the polynomial approximation bound. Let $f$ be some element of $\mathcal{F}$ and $p_{\mathrm{up}}, p_{\mathrm{low}}$ the corresponding $\frac{\epsilon^2}{C}$-$\mathcal{L}_2$ sandwiching polynomials. If $\tilde{\mathcal{D}}$ is the $\mathcal{X}$-marginal of $\tilde{\mathcal{D}}_{\mathcal{X}\mathcal{Y}}$, we have the following, by applying the sandwiching property, Jensen's inequality and the definition of $\tilde{\mathcal{D}}$.

$$\begin{aligned}(\mathbb{E}_{\mathbf{x}\sim\tilde{\mathcal{D}}}[|f(\mathbf{x}) - p_{\mathrm{low}}(\mathbf{x})|])^2 &\le (\mathbb{E}_{\mathbf{x}\sim\tilde{\mathcal{D}}}[p_{\mathrm{up}}(\mathbf{x}) - p_{\mathrm{low}}(\mathbf{x})])^2 \\ &\le \mathbb{E}_{\mathbf{x}\sim\tilde{\mathcal{D}}}[(p_{\mathrm{up}}(\mathbf{x}) - p_{\mathrm{low}}(\mathbf{x}))^2] \\ &\le \frac{\mathbb{E}_{\mathbf{x}\sim\mathcal{D}'}[(p_{\mathrm{up}}(\mathbf{x}) - p_{\mathrm{low}}(\mathbf{x}))^2 g(\mathbf{x})]}{\mathbb{P}_{\mathbf{x}\sim\mathcal{D}'}[g(\mathbf{x}) = 1]}\end{aligned}$$

By applying condition (a), as well as the fact that $\mathbb{P}_{\mathbf{x}\sim\mathcal{D}'}[g(\mathbf{x}) = 1] \ge \Omega(1)$ (since $\mathbb{P}_{\mathbf{x}\sim\mathcal{D}'}[g(\mathbf{x}) = 0]$), we have

$$(\mathbb{E}_{\mathbf{x}\sim\tilde{\mathcal{D}}}[|f(\mathbf{x}) - p_{\mathrm{low}}(\mathbf{x})|])^2 \le O(1) \cdot \mathbb{E}_{\mathbf{x}\sim\mathcal{D}}[(p_{\mathrm{up}}(\mathbf{x}) - p_{\mathrm{low}}(\mathbf{x}))^2] \le O(\epsilon^2/C)$$

Therefore, indeed, $\mathbb{E}_{\mathbf{x}\sim\tilde{\mathcal{D}}}[|f(\mathbf{x}) - p_{\mathrm{low}}(\mathbf{x})|] \le \epsilon$, which concludes the proof of Theorem D.1.

### D.2 Robust Learning

We also provide the following result for learning with nasty noise (see Definition A.17). While there are algorithms for learning in the nasty noise model that are more efficient than the one we analyze here (see [DKS18]), we achieve an error bound that is close to the optimal: we only incur a multiplicative factor of 2 from the information theoretically optimal bound of $2\eta$, where $\eta$ is the noise rate (see Definition A.17). For intersections of halfspaces, for instance, to the best of our knowledge, all prior known dimension-efficient algorithms incurred an error of $\Omega(\sqrt{\eta})$ for learning under nasty noise of rate $\eta$ (see [KKM18, DKS18]).

**Theorem D.3** (Learning with Nasty Noise via Sandwiching). *For $\epsilon, \eta, \delta \in (0, 1)$, let $\mathcal{X} \subseteq \mathbb{R}^d$ and $(\mathcal{D}, \mathcal{F})$ be an $(\frac{\epsilon^2}{C}, \frac{\delta}{C}, k, m)$-reasonable pair (Definition 4.5) for some sufficiently large universal constant $C > 0$. Then, there is a robust learner for $\mathcal{F}$ under nasty noise of rate $\eta$ with respect to $\mathcal{D}$ up to error $4\eta + \epsilon$ and probability of failure $\delta$ with sample complexity $m + \mathrm{poly}(\frac{1}{\epsilon}(kd)^k \log(1/\delta))$ and time complexity $\mathrm{poly}(\frac{m}{\epsilon}(kd)^k \log(1/\delta))$.*

*Proof.* We follow a very similar approach as the one for Theorem D.1. The main differences are two. First, instead of the outlier removal of Theorem 3.1, we apply the outlier removal of Theorem E.1, which works in the adversarial setting. Second, in the nasty noise setting, we assume that the noise rate is bounded and hence we do not need to run any tests in order to obtain the desired guarantees.

Recall that in this setting, the learner receives a sample $S$ of size $N$ with $S = S_{\mathrm{iid}} \cup S_{\mathrm{adv}} \setminus S_{\mathrm{rem}}$, where $S_{\mathrm{iid}}$ is an i.i.d. labeled sample drawn from $\mathcal{D}$ and labeled by some $f^* \in \mathcal{F}$, $|S_{\mathrm{adv}}| = |S_{\mathrm{rem}}| \le \eta N$, where $S_{\mathrm{rem}}$ is an arbitrary subset of $S_{\mathrm{iid}}$ and $S_{\mathrm{adv}}$ is an arbitrary sample of size $S_{\mathrm{rem}}$ (i.e., the adversary removes the samples in $S_{\mathrm{rem}}$ and substitutes them with adversarial samples $S_{\mathrm{adv}}$).

The algorithm runs the outlier removal of Theorem E.1 on $S$ with parameters $\alpha \leftarrow 1$, $\epsilon \leftarrow \epsilon/C$, $\delta \leftarrow \delta/C$ and $k \leftarrow k$ to receive a filtered set of samples $S_{\mathrm{filt}}$ such that $|S_{\mathrm{iid}} \setminus S_{\mathrm{filt}}| \le |S_{\mathrm{adv}}| + \epsilon N/C \le$

$\eta N + \epsilon N/C$ and also $\frac{1}{N} \sum_{\mathbf{x} \in S_{\text{filt}}} (p(\mathbf{x}))^2 \leq 200 \, \mathbb{E}_{\mathbf{x} \sim \mathcal{D}}[(p(\mathbf{x}))^2]$ for any polynomial $p$ of degree at most $k$. Then, it runs polynomial regression of degree $k$ with coefficient bound $B$ (given by Definition 4.5) over the set $S_{\text{filt}}$ and outputs $h(\mathbf{x}) = \text{sign}(\widehat{p}(\mathbf{x}))$ where $\widehat{p}$ is the output of the polynomial regression routine (of Proposition D.2).

We aim to bound $\mathbb{P}_{\mathbf{x} \sim \mathcal{D}}[f^*(\mathbf{x}) \neq h(\mathbf{x})]$. By uniform convergence, we have a bound of the form $\mathbb{P}_{\mathbf{x} \sim \mathcal{D}}[f^*(\mathbf{x}) \neq h(\mathbf{x})] \leq \mathbb{P}_{(\mathbf{x},y) \sim S_{\text{iid}}}[y \neq h(\mathbf{x})] + \epsilon/C \leq \mathbb{P}_{(\mathbf{x},y) \sim S}[y \neq h(\mathbf{x})] + \eta + \epsilon/C$. We further bound the quantity $\mathbb{P}_{(\mathbf{x},y) \sim S}[y \neq h(\mathbf{x})] \leq \mathbb{P}_{(\mathbf{x},y) \sim S}[y \neq h(\mathbf{x}), (\mathbf{x},y) \in S_{\text{filt}}] + \frac{|S \backslash S_{\text{filt}}|}{N} \leq \mathbb{P}_{(\mathbf{x},y) \sim S}[y \neq h(\mathbf{x}), (\mathbf{x},y) \in S_{\text{filt}}] + 2\eta$.

We may apply Proposition D.2 to show that $\mathbb{P}_{(\mathbf{x},y) \sim S}[y \neq h(\mathbf{x}), (\mathbf{x},y) \in S_{\text{filt}}] \leq \eta + O(\epsilon/C)$, as long as the following is true for some polynomial of degree at most $k$.

$$\frac{1}{N} \sum_{(\mathbf{x},y) \in S_{\text{filt}}} |f^*(\mathbf{x}) - p(\mathbf{x})| \leq O(\epsilon/C)$$

Due to the sandwiching property, this is true for the sandwiching polynomial $p_{\text{low}}$ for $f^*$ (which exists since $f^* \in \mathcal{F}$ – see Definition 4.5). To show this, we may follow an analogous approach as the one for Theorem D.1. $\qquad \square$

# E  Outlier Removal Procedure

We now give the proofs of our outlier removal theorem in the adversarial, as well as the PQ setting.

## E.1  Outlier Removal in the Adversarial Setting

We present our outlier removal result in the adversarial setting:

**Theorem E.1.** *There exists an algorithm that satisfies the following specifications for some sufficiently large absolute constant $C$. The algorithm $\mathcal{A}$ is given parameters $\epsilon, \alpha, \delta$ in $(0,1]$, a positive integer $k$, and a pair of size-$N$ sets $S_{\mathcal{D}}$ and $S_{\mathcal{D}'}$ of points in $\mathbb{R}^d$, where $N \geq C \left( \frac{(kd)^k}{\epsilon} \log \frac{1}{\delta} \right)^C$. The algorithm $\mathcal{A}$ then accepts a subset $S_{accept} \subseteq S_{\mathcal{D}'}$, rejects a subset $S_{reject} = S_{\mathcal{D}'} \backslash S_{accept}$ and runs in time $\text{poly}(N)$.*

*Let the set $S_{\mathcal{D}}$ in $\mathbb{R}^d$ of size $N$ be sampled i.i.d. from a $k$-tame probability distribution $\mathcal{D}$, and let $S_{\mathcal{D}'}$ be generated by:*

1. *Sampling a size-$N$ i.i.d. set $S_{clean}$ from $\mathcal{D}$.*

2. *Adversary corrupting an arbitrary subset of elements in $S_{clean}$. Formally, $S_{\mathcal{D}'} = S_{uncorrupted} \cup S_{adversarial}$, where $S_{uncorrupted}$ is an adversarially chosen subset of $S_{clean}$ and $S_{adversarial}$ is a set of adversarially chosen points in $\mathbb{R}^d$ of size $N - |S_{uncorrupted}|$.*

*Then, with probability at least $1 - \delta$, the algorithm $\mathcal{A}$ given the sets $S_{\mathcal{D}}$ and $S_{\mathcal{D}'}$ will accept a set $S_{accept} \subseteq S_{\mathcal{D}'}$ satisfying the following two properties:*

- ***Degree-$k$ spectral $\frac{200}{\alpha}$-boundedness****: For every polynomial $p$ of degree at most $k$ satisfying*

$$\mathbb{E}_{\mathbf{x} \sim \mathcal{D}}[(p(\mathbf{x}))^2] \leq 1,$$

  *it is the case that*

$$\frac{1}{N} \sum_{\mathbf{x} \in S_{accept}} p(\mathbf{x})^2 \leq \frac{200}{\alpha}.$$

- ***$(\alpha, \epsilon/2)$-validity****: The set $S_{reject} \cap S_{uncorrupted}$ has a size of at most $\alpha |S_{adversarial}| + \frac{\epsilon}{2} N$.*

We describe our algorithm for Theorem E.1 (restating Algorithm 1):

1. **Input** Sets $S_{\mathcal{D}'}$ and $S_{\mathcal{D}}$ of size $N$ in $\mathbb{R}^d$, parameters $\epsilon, \delta$ in $(0,1)$.

2. $\widehat{M} \leftarrow$ **ESTIMATE-MOMENTS**$(S_{\mathcal{D}}, k, \delta/10)$. (See Lemma B.1 for further info).

3. $B_0 \leftarrow \frac{4d^{3k}}{\epsilon}$ and $\Delta_0 \leftarrow 200\sqrt{d^{2k}\frac{\log N}{N}\log\frac{1}{\delta}}$

4. $S_{\text{filtered}}^0 \leftarrow S_{\mathcal{D}'} \setminus \left\{\mathbf{x} : \max_{p:\, p^\top \widehat{M}p \leq 1}(p(\mathbf{x}))^2 > B_0\right\}$.

5. While $\max_{p:\, p^\top \widehat{M}p \leq 1} \left(\frac{1}{N}\sum_{\mathbf{x}\in S_{\text{filtered}}^i}(p(\mathbf{x}))^2\right) > \frac{50}{\alpha}(1 + \Delta_0 \cdot B_0)$.

   (a) $p_i \leftarrow \arg\max_{p:\, p^\top \widehat{M}p \leq 1}\left(\frac{1}{N}\sum_{\mathbf{x}\in S_{\text{filtered}}^i}(p(\mathbf{x}))^2\right)$

   (b) Set $\tau_i$ to be the smallest value of $\tau$ subject to:

   $$\frac{1}{N}\left|\{\mathbf{x} \in S_{\text{filtered}}^i :\ (p_i(\mathbf{x}))^2 > \tau\}\right| \geq \frac{10}{\alpha}\left(\Pr_{\mathbf{x}\sim S_{\mathcal{D}}}[B_0 \geq (p_i(\mathbf{x}))^2 > \tau] + \Delta_0\right)$$

   (c) $S_{\text{filtered}}^{i+1} \leftarrow S_{\text{filtered}}^i \setminus \{\mathbf{x} :\ (p_i(\mathbf{x}))^2 > \tau_i\}$
   (d) $i \leftarrow i + 1$

6. Output $(S_{\text{accept}}, S_{\text{reject}}) = (S_{\text{filtered}}^i, S_{\mathcal{D}'} \setminus S_{\text{filtered}}^i)$.

Note that the procedure ESTIMATE-MOMENTS produces a good spectral approximation for the degree-$k$ moment matrix of $\mathcal{D}$. Formally, Lemma B.1 says that the matrix $\widehat{M}$ is symmetric positive-semidefinite and with probability at least $1 - \delta/10$ it is the case that every degree-$k$ polynomial $p$ satisfies.

$$\frac{9}{10}\mathbb{E}_{\mathbf{x}\sim\mathcal{D}}[(p(\mathbf{x}))^2] \leq p^\top \widehat{M}p \leq \frac{11}{10}\mathbb{E}_{\mathbf{x}\sim\mathcal{D}}[(p(\mathbf{x}))^2]. \tag{E.1}$$

### E.1.1 Efficient implementation

We now explain how to execute certain steps of our algorithm in polynomial time:

- The quantity $\max_{p:\, p^\top \widehat{M}p \leq 1}(p(\mathbf{x}))^2$ equals to the largest eigenvalue of the matrix

$$(\widehat{M})^{-1/2}\left((\mathbf{x}^{\otimes d})(\mathbf{x}^{\otimes d})^\top\right)(\widehat{M})^{-1/2},$$

  which can be computed in polynomial time in the dimension $m$ of the matrix. (Note that if the matrix $\widehat{M}$ is not full-rank, then the above is still true if long as $(\widehat{M})^{-1/2}$ is replaced by the Moore–Penrose pseudo-inverse of $(\widehat{M})^{1/2}$, which again can be computed efficiently. Also note that we used the fact that the matrix $\widehat{M}$ is symmetric.)

- The quantity $\max_{p:\, p^\top \widehat{M}p \leq 1}\left(\frac{1}{N}\sum_{\mathbf{x}\in S_{\text{filtered}}^i}(p(\mathbf{x}))^2\right)$ equals to the largest eigenvalue of the matrix

$$(\widehat{M})^{-1/2}\left(\frac{1}{N}\sum_{\mathbf{x}\in S_{\text{filtered}}^i}(\mathbf{x}^{\otimes d})(\mathbf{x}^{\otimes d})^\top\right)(\widehat{M})^{-1/2},$$

  which can be computed in polynomial time in the dimension $m$ of the matrix. (Again, if the matrix $\widehat{M}$ is not full-rank, then the above is still true if long as $(\widehat{M})^{-1/2}$ is replaced by the Moore–Penrose pseudo-inverse of $(\widehat{M})^{1/2}$. Also note that we again used the fact that the matrix $\widehat{M}$ is symmetric.)

- The polynomial $p_i \leftarrow \arg\max_{p:\, p^\top \widehat{M}p \leq 1}\left(\frac{1}{N}\sum_{\mathbf{x}\in S_{\text{filtered}}^i}(p(\mathbf{x}))^2\right)$ can be computed by taking the leading eigenvector of $(\widehat{M})^{-1/2}\left(\frac{1}{N}\sum_{\mathbf{x}\in S_{\text{filtered}}^i}(\mathbf{x}^{\otimes d})(\mathbf{x}^{\otimes d})^\top\right)(\widehat{M})^{-1/2}$ and multiplying this vector by $(\widehat{M})^{-1/2}$ (again, one takes the Moore–Penrose pseudo-inverse if $\widehat{M}$ is not full-rank).

- The value of $\tau_i$ can be computed in time $\text{poly}(N)$ by considering all the candidate values $\tau$ of the form $(p_i(\mathbf{x}))^2$ for all elements $\mathbf{x}$ in $S_{\text{filtered}}^i$, and setting $\tau_i$ to be the smallest candidate that satisfies the condition in the algorithm.

Note that to prove the run-time bound of $\text{poly}(N)$ for the algorithm as a whole we will need to bound the total number of iterations in the main while loop, which is done in Section E.1.3.

### E.1.2 Correctness analysis

We now proceed to proving first the correctness of the algorithm in Section E.1.2. Then, in Section E.1.3 we show the required run-time bound.

In this section we show that with probability at least $1-\delta$ the algorithm $\mathcal{A}$ satisfies the two correctness guarantees in Theorem E.1. We begin by arguing that with probability at least $1 - \delta$ the sets $S_{\mathcal{D}}$ and $S_{\text{clean}}$ are well-behaved.

**Claim 1.** *Let the set $S$ be formed of $N$ i.i.d. samples from a $k$-tame distribution $\mathcal{D}$, where $N \geq C \left( \frac{(kd)^k}{\epsilon} \log \frac{1}{\delta} \right)^C$ and $C$ is a sufficiently large absolute constant. Also, let $B_0 = \frac{4d^{3k}}{\epsilon}$. Then with probability at least $1 - \delta/10$ the set $S$ satisfies the following properties for any polynomial $p$ over $\mathbb{R}^d$ of degree at most $k$ and any pair of values of $\tau_1, \tau_2$ in $\mathbb{R}$ :*

$$\left| \frac{|\{\mathbf{x} \in S : \tau_2 \geq (p(\mathbf{x}))^2 > \tau_1\}|}{N} - \mathbb{P}_{\mathbf{x}\sim\mathcal{D}}[\tau_2 \geq (p(\mathbf{x}))^2 > \tau_1] \right| \leq 100 \sqrt{\frac{d^{2k} \log N}{N} \log \frac{1}{\delta}} \quad (1)$$

$$\frac{1}{N} \left| \left\{ \mathbf{x} \in S : \max_{p' \text{ of degree } k:\ \mathbb{E}_{\mathbf{x}\sim\mathcal{D}}[(p'(\mathbf{x}))^2]\leq 1} (p'(\mathbf{x}))^2 > \frac{2d^{3k}}{\epsilon} \right\} \right| \leq \frac{3\epsilon}{4}, \quad (2)$$

*and if the polynomial $p$ further satisfies $\mathbb{E}_{\mathbf{x}\sim\mathcal{D}}[(p(\mathbf{x}))^2] \leq 1$ then also*

$$\mathbb{E}_{\mathbf{x}\sim S_{\mathcal{D}}} \left[ (p(\mathbf{x}))^2 \cdot \mathbf{1}_{(p(\mathbf{x}))^2 \leq B_0} \right] \leq \mathbb{E}_{\mathbf{x}\sim\mathcal{D}}[(p(\mathbf{x}))^2] + 0.01 \quad (3)$$

*Proof.* Since $(p(\mathbf{x}))^2$ is a polynomial of degree at most $2k$, the every function of the form $\{\mathbf{1}_{\tau_2 \geq (p(\mathbf{x}))^2 > \tau}\}$ is an AND of two degree-$2k$ polynomial threshold functions. Since degree-$2k$ polynomial threshold functions have a VC dimension of at most $d^{2k} + 1$, we can use Fact A.5 and Fact A.6 to conclude that property (1) holds with probability at least $1 - \delta/30$.

Now, we show that property (2) is likely to be satisfied. Then there is a collection $\{r_1, \cdots, r_{m'}\}$ of degree-$k$ polynomials that satisfy

$$\mathbb{E}_{\mathbf{x}\sim\mathcal{D}}[r_j(\mathbf{x})r_{j'}(\mathbf{x})] = \begin{cases} 1 & \text{if } j = j' \\ 0 & \text{otherwise.} \end{cases}$$

(Such collection necessarily exists via the Gram-Schmidt process.) We let $M$ denote the matrix $\mathbb{E}_{\mathbf{x}\sim\mathcal{D}}(\mathbf{x}^{\otimes d})(\mathbf{x}^{\otimes d})^\top$. Additionally, we consider a basis $\{g_1, \cdots, g_{m-m'}\}$ for the nullspace of $M$. Now, for $\mathbf{x}$ sampled from $\mathcal{D}$ we have:

- For a specific index $j$, we have $\mathbb{E}_{\mathbf{x}\sim\mathcal{D}}[(r_j(\mathbf{x})^2] = 1$ and therefore by Markov's inequality we have

$$\mathbb{P}_{\mathbf{x}\sim\mathcal{D}} \left[ (r_j(\mathbf{x}))^2 \leq \frac{2d^k}{\epsilon} \right] \geq 1 - \frac{\epsilon}{2d^k}$$

- Each $g_j$ has $\mathbb{E}_{\mathbf{x}\sim\mathcal{D}}[(g_j(\mathbf{x})^2] = 0$, and therefore $\mathbb{P}_{\mathbf{x}\sim\mathcal{D}}[g_j(\mathbf{x}) = 0] = 1$.

By union bound, all the events above take place for $\mathbf{x}$ sampled from $\mathcal{D}$ with probability at least $1 - m \frac{\epsilon}{2d^k} \geq 1 - \frac{\epsilon}{2}$. Via the standard Hoeffding bound, with probability at least $1 - \delta$ the events above take place for at least $\frac{\epsilon}{2} + \sqrt{\frac{20}{N} \log \frac{20}{\delta}}$ fraction of elements $\mathbf{x}$ in $S$. Since $\{r_j\}$ are orthonormal with respect to $M$, every degree-$k$ polynomial $p'$ satisfying $\mathbb{E}_{\mathbf{x}\sim\mathcal{D}}[(p'(\mathbf{x}))^2] \leq 1$ can be decomposed as $p' = \sum_{i=0}^{m'} \alpha_i r_i + \sum_{i=0}^{m-m'} \beta_i g_i$, where each $\alpha_i$ is in $[-1, 1]$. Therefore if the events above take

place for a point $\mathbf{x}$, then

$$\max_{\substack{p' \text{ of degree } k: \\ \mathbb{E}_{\mathbf{x}\sim\mathcal{D}}[(p'(\mathbf{x}))^2]\leq 1}} (p'(\mathbf{x}))^2 = \max_{\substack{\alpha_1,\cdots,\alpha_{m'}\in[-1,1] \\ \beta_1,\cdots\beta_{m-m'}\in\mathbb{R}}} \left( \sum_{i=0}^{m'} \alpha_i r_i(\mathbf{x}) + \overbrace{\sum_{i=0}^{m-m'} \beta_i g_i(\mathbf{x})}^{=0} \right)^2 \leq$$

$$\left( \sum_{i=0}^{m'} \overbrace{|r_i(\mathbf{x})|}^{\leq\sqrt{2d^k/\epsilon}} \right)^2 \leq \frac{2d^k(m')^2}{\epsilon} \leq \frac{2d^{3k}}{\epsilon}, \quad \text{(E.2)}$$

from which Property (2) follows.

Finally, we remark that Property (3) holds with probability at least $1 - \delta/30$ due to an argument analogous to the proof of Lemma B.2 (where in place of $f$, we have the function $\mathbf{1}_{p^2(\mathbf{x})\leq B_0}$). $\qquad\square$

We now argue that in each iteration $i$ there is a value of $\tau_i$ satisfying the condition in step (5b)

**Claim 2.** *Suppose the set $S_\mathcal{D}$ is such that it satisfies property (3) of claim 1, i.e. for every degree-$k$ polynomial $p$ satisfying $\mathbb{E}_{\mathbf{x}\sim\mathcal{D}}(p(\mathbf{x}))^2 \leq 1$ we have $\mathbb{E}_{\mathbf{x}\sim S_\mathcal{D}} \left[ (p(\mathbf{x}))^2 \cdot \mathbf{1}_{(p(\mathbf{x}))^2\leq B_0} \right] \leq \mathbb{E}_{\mathbf{x}\sim\mathcal{D}}[(p(\mathbf{x}))^2] + 0.01$ and the matrix $\widehat{M}$ satisfies Equation E.1, i.e. for every degree-$k$ polynomial $p$ we have*

$$\frac{9}{10} \mathbb{E}_{\mathbf{x}\sim\mathcal{D}}[(p(\mathbf{x}))^2] \leq p^\top\widehat{M}p \leq \frac{11}{10} \mathbb{E}_{\mathbf{x}\sim\mathcal{D}}[(p(\mathbf{x}))^2].$$

*Suppose it is the case that $\frac{1}{N}\sum_{\mathbf{x}\in S^i_{filtered}} (p_i(\mathbf{x}))^2 > \frac{50}{\alpha}(1 + \Delta_0 \cdot B_0)$ (i.e. the while loop does not terminate at step $i$). The there exists some $\tau$ for which*

$$\frac{1}{N}\left|\{\mathbf{x} \in S^i_{filtered} : (p_i(\mathbf{x}))^2 > \tau\}\right| \geq \frac{10}{\alpha}\left(\mathbb{P}_{\mathbf{x}\sim S_\mathcal{D}}[B_0 \geq (p_i(\mathbf{x}))^2 > \tau] + \Delta_0\right).$$

*Proof of Claim 2.* For the sake of contradiction, suppose that for every $\tau \geq 0$ it is the case that

$$\frac{1}{N}\left|\{\mathbf{x} \in S^i_{filtered} : (p_i(\mathbf{x}))^2 > \tau\}\right| \leq \frac{10}{\alpha}\left(\mathbb{P}_{\mathbf{x}\sim S_\mathcal{D}}[B_0 \geq (p_i(\mathbf{x}))^2 > \tau] + \Delta_0\right). \quad \text{(E.3)}$$

Since every element $\mathbf{x}$ of $S^i_{filtered}$ satisfies $(p_i(\mathbf{x}))^2 \leq B_0$ we have

$$\frac{1}{N}\sum_{\mathbf{x}\in S^i_{filtered}} (p_i(\mathbf{x}))^2 = \int_{\tau=0}^\infty \frac{1}{N}\left|\{\mathbf{x} \in S^i_{filtered} : (p_i(\mathbf{x}))^2 > \tau\}\right| d\tau =$$

$$\int_{\tau=0}^{B_0} \frac{1}{N}\left|\{\mathbf{x} \in S^i_{filtered} : (p_i(\mathbf{x}))^2 > \tau\}\right| d\tau. \quad \text{(E.4)}$$

which combined with Equation E.3 implies

$$\frac{1}{N}\sum_{\mathbf{x}\in S^i_{filtered}} (p_i(\mathbf{x}))^2 \leq \frac{10}{\alpha}\left(\Delta_0 B_0 + \int_{\tau=0}^\infty \mathbb{P}_{\mathbf{x}\sim S_\mathcal{D}}[B_0 \geq (p_i(\mathbf{x}))^2 > \tau] d\tau\right) =$$

$$\frac{10}{\alpha}\left(\Delta_0 B_0 + \mathbb{E}_{\mathbf{x}\sim S_\mathcal{D}}\left[p_i(\mathbf{x}))^2 \cdot \mathbf{1}_{(p_i(\mathbf{x}))^2\leq B_0}\right]\right) \quad \text{(E.5)}$$

Additionally, since $S_\mathcal{D}$ is assumed to satisfy property (3) in Claim 1, and $\widehat{M}$ is assumed to satisfy Equation E.1, we have

$$\mathbb{E}_{\mathbf{x}\sim S_\mathcal{D}}\left[(p_i(\mathbf{x}))^2 \cdot \mathbf{1}_{(p_i(\mathbf{x}))^2\leq B_0}\right] \leq \mathbb{E}_{\mathbf{x}\sim\mathcal{D}}[(p_i(\mathbf{x}))^2] + 0.01 \leq \frac{11}{10}(p_i)^\top\widehat{M}(p_i) + 0.01 \leq \frac{111}{100}. \quad \text{(E.6)}$$

Combining Equation E.5 and Equation E.6 we get

$$\frac{1}{N} \sum_{\mathbf{x} \in S_{\text{filtered}}^i} (p_i(\mathbf{x}))^2 \leq \frac{10}{\alpha} \left( \Delta_0 B_0 + \frac{111}{100} \right).$$

This contradicts the premise that $\frac{1}{N} \sum_{\mathbf{x} \in S_{\text{filtered}}^i} (p_i(\mathbf{x}))^2 > \frac{50}{\alpha} (1 + \Delta_0 \cdot B_0)$, finishing the proof. $\quad\square$

Now, we proceed to argue that if all the properties in Claim 1 and Equation E.1 hold then the algorithm $\mathcal{A}$ will satisfy the $(\alpha, \epsilon/2)$-validity property. In other words, we show that the set $S_{\text{accept}} \cap S_{\text{uncorrupted}}$ has a size of at most $\alpha |S_{\text{adversarial}}| + \frac{\epsilon}{2} N$. The set $S_{\text{accept}} \cap S_{\text{uncorrupted}}$ consists of two components:

- The elements in $\left\{ \mathbf{x} \in S_{\text{uncorrupted}} : \max_{\substack{p \text{ of degree } k \\ p^\top \widehat{M} p \leq 1}} (p(\mathbf{x}))^2 > B_0 \right\}$, whose number is upper-bounded by $2\epsilon N/3$ for the following reason. Equation E.1 implies that whenever $p^\top \widehat{M} p$ holds, we also have $\mathbb{E}_{\mathbf{x} \sim \mathcal{D}}[(p(\mathbf{x}))^2] \leq 10/9$ and since $S_{\text{uncorrupted}}$ is assumed to satisfy Claim 1, for at least $1 - 2\epsilon/3$ fraction of elements $\mathbf{x}$ in $S_{\text{uncorrupted}}$ we have $(p(\mathbf{x}))^2 \leq \frac{10}{9} \frac{2d^{3k}}{\epsilon}$, which is less than $B_0$.

- The elements in $\bigcup_i \left( (S_{\text{filtered}}^i \setminus S_{\text{filtered}}^{i+1}) \cap S_{\text{uncorrupted}} \right)$, the number of which is bounded by $\frac{2\alpha}{5} |S_{\text{adversarial}}|$ by the following claim.

**Claim 3.** *Suppose the sets $S_{\mathcal{D}}$ and $S_{\text{clean}}$ satisfy the properties in Claim 1. Then, for each iteration $i$ of the main loop of the algorithm, it is the case that*

$$\left| (S_{\text{filtered}}^i \setminus S_{\text{filtered}}^{i+1}) \cap S_{\text{uncorrupted}} \right| \leq \frac{2\alpha}{5} \left| (S_{\text{filtered}}^i \setminus S_{\text{filtered}}^{i+1}) \cap S_{\text{adversarial}} \right|.$$

*Proof.* Since the set $S_{\text{uncorrupted}}$ is a subset of the set $S_{\text{clean}}$, we have

$$\left| (S_{\text{filtered}}^i \setminus S_{\text{filtered}}^{i+1}) \cap S_{\text{uncorrupted}} \right| \leq \left| (S_{\text{filtered}}^i \setminus S_{\text{filtered}}^{i+1}) \cap S_{\text{clean}} \right| \qquad (\text{E.7})$$

Also, based on how the algorithm chooses the set $S_{\text{filtered}}^{i+1}$ and the parameter $\tau_i$, we have:

$$\frac{|S_{\text{filtered}}^i \setminus S_{\text{filtered}}^{i+1}|}{N} = \frac{1}{N} \left| \{ \mathbf{x} \in S_{\text{filtered}}^i : (p_i(\mathbf{x}))^2 > \tau_i \} \right| \geq$$

$$\frac{10}{\alpha} \left( \mathbb{P}_{\mathbf{x} \sim S_{\mathcal{D}}} [B_0 \geq (p_i(\mathbf{x}))^2 > \tau_i] + 200 \sqrt{d^{2k} \frac{\log N}{N} \log \frac{1}{\delta}} \right).$$

but since $S_{\mathcal{D}}$ and $S_{\text{clean}}$ satisfy Property 1 in Claim 1, we also have

$$\frac{1}{N} \left| \{ \mathbf{x} \in S_{\text{clean}} : B_0 \geq (p_i(\mathbf{x}))^2 > \tau_i \} \right| \leq \mathbb{P}_{\mathbf{x} \sim S_{\mathcal{D}}} [B_0 \geq (p_i(\mathbf{x}))^2 > \tau_i] + 200 \sqrt{d^{2k} \frac{\log N}{N} \log \frac{1}{\delta}}.$$

Combining the preceding two inequalities yields

$$\left| S_{\text{filtered}}^i \setminus S_{\text{filtered}}^{i+1} \right| \geq \frac{10}{\alpha} \left| \{ \mathbf{x} \in S_{\text{clean}} : B_0 \geq (p_i(\mathbf{x}))^2 > \tau_i \} \right|.$$

We argue that every $\mathbf{x}$ in $(S_{\text{filtered}}^i \setminus S_{\text{filtered}}^{i+1}) \cap S_{\text{clean}}$ satisfies $B_0 \geq (p_i(\mathbf{x}))^2 > \tau_i$. Indeed, if $\mathbf{x}$ belongs to $S_{\text{filtered}}^i$, it also belongs to $S_{\text{filtered}}^0$ and therefore $(p_i(\mathbf{x}))^2 \leq B_0((p_i)^\top \widehat{M}(p_i)) \leq B_0$. It also has to be that $(p_i(\mathbf{x}))^2 > \tau_i$ because of how $S_{\text{filtered}}^{i+1}$ is defined inside the algorithm. Thus, we have

$$\left| S_{\text{filtered}}^i \setminus S_{\text{filtered}}^{i+1} \right| \geq \frac{10}{\alpha} \left| (S_{\text{filtered}}^i \setminus S_{\text{filtered}}^{i+1}) \cap S_{\text{clean}} \right|.$$

Since $S_{\text{filtered}}^i \setminus S_{\text{filtered}}^{i+1}$ is the disjoint union of $(S_{\text{filtered}}^i \setminus S_{\text{filtered}}^{i+1}) \cap S_{\text{clean}}$ and $S_{\text{filtered}}^i \setminus S_{\text{filtered}}^{i+1}) \cap S_{\text{adversarial}}$ we further conclude that

$$\left| (S_{\text{filtered}}^i \setminus S_{\text{filtered}}^{i+1}) \cap S_{\text{adversarial}} \right| \geq \left( \frac{10}{\alpha} + 1 \right) \left| (S_{\text{filtered}}^i \setminus S_{\text{filtered}}^{i+1}) \cap S_{\text{clean}} \right|.$$

Finally, recalling that $S_{\text{uncorrupted}}$ is contained in $S_{\text{clean}}$, we conclude the proposition. $\quad\square$

Overall, the above claim concludes the proof of $(\alpha, \epsilon)$-validity. Now we proceed to proving the spectral $\frac{100}{\alpha}$-boundedness.

**Claim 4.** *Suppose the algorithm terminates and produces a partition $(S_{accept}, S_{reject})$ and the matrix $\widehat{M}$ satisfies Equation E.1. Also, suppose that $C$ exceeds a certain absolute constant. Then, for every polynomial $p$ of degree at most $k$ satisfying*

$$\mathbb{E}_{\mathbf{x} \sim \mathcal{D}}[(p(\mathbf{x}))^2] \leq 1,$$

*it is the case that*

$$\frac{1}{N} \sum_{\mathbf{x} \in S_{accept}} p(\mathbf{x})^2 \leq \frac{100}{\alpha}. \tag{E.8}$$

*Proof.* Since the matrix $\widehat{M}$ satisfies Equation E.1, we have $p^\top \widehat{M} p \leq 11/10$ and since the main while loop of the algorithm has terminated, for the final value $i_{\max}$ of $i$ it is the case that

$$\frac{1}{N} \sum_{\mathbf{x} \in S^i_{\text{filtered}}} (p(\mathbf{x}))^2 \leq \frac{11}{10} \cdot \frac{50}{\alpha} \cdot \left(1 + 200\sqrt{d^{2k}\frac{\log N}{N}\log\frac{1}{\delta}} \cdot B_0\right).$$

Substituting $B_0 = \frac{4d^{3k}}{\epsilon}$ and $N \geq C\left(\frac{(kd)^k}{\epsilon}\log\frac{1}{\delta}\right)^C$, we see that the inequality above yields Equation E.8 when $C$ exceeds a large enough absolute constant. $\qquad\square$

### E.1.3 Run-time analysis

We now prove the run-time bound of $\text{poly}(N)$. Since each step of the algorithm takes time $\text{poly}(Nd^k/\epsilon) = \text{poly}(N)$ (see Section E.1.1), in order to obtain a required run-time bound, it is enough to show that the number of iterations of the main while loop is at most $N$. We argue this via the following claim:

**Claim 5.** *Suppose the sets $S_{\mathcal{D}}$ and $S_{clean}$ satisfy the properties in Claim 1 and the matrix $\widehat{M}$ satisfies Equation E.1. Then, for every $i < i_{\max}$, we have $\frac{1}{N}\sum_{\mathbf{x} \in S^{i+1}_{\text{filtered}}}(p_i(\mathbf{x}))^2 \leq \frac{50}{\alpha}\left(1 + 200\sqrt{d^{2k}\frac{\log N}{N}\log\frac{1}{\delta}} \cdot \frac{4d^{3k}}{\epsilon}\right)$*

Indeed, since in $i$-th loop of the algorithm we have

$$\frac{1}{N} \sum_{\mathbf{x} \in S^i_{\text{filtered}}} (p_i(\mathbf{x}))^2 > \frac{50}{\alpha}\left(1 + 200\sqrt{d^{2k}\frac{\log N}{N}\log\frac{1}{\delta}} \cdot \frac{4d^{3k}}{\epsilon}\right),$$

the claim above implies that necessarily $S^{i+1}_{\text{filtered}} \neq S^i_{\text{filtered}}$, which means that $|S^{i+1}_{\text{filtered}}| \leq |S^i_{\text{filtered}}| - 1$. Therefore the total number of iterations $i_{\max}$ is upper-bounded by $N$. Now, we prove Claim 5.

*Proof of Claim 5.* Since every element $\mathbf{x}$ of $S^{i+1}_{\text{filtered}}$ satisfies $(p_i(\mathbf{x}))^2 \leq \tau_i$ and the set $S^{i+1}_{\text{filtered}}$ is a subset of $S^i_{\text{filtered}}$ we have

$$\frac{1}{N} \sum_{\mathbf{x} \in S^{i+1}_{\text{filtered}}} (p_i(\mathbf{x}))^2 = \int_{\tau=0}^{\infty} \frac{1}{N}\left|\{\mathbf{x} \in S^{i+1}_{\text{filtered}} : (p_i(\mathbf{x}))^2 > \tau\}\right| d\tau =$$

$$\int_{\tau=0}^{\tau_i} \frac{1}{N}\left|\{\mathbf{x} \in S^{i+1}_{\text{filtered}} : (p_i(\mathbf{x}))^2 > \tau\}\right| d\tau \leq \int_{\tau=0}^{\tau_i} \frac{1}{N}\left|\{\mathbf{x} \in S^i_{\text{filtered}} : (p_i(\mathbf{x}))^2 > \tau\}\right| d\tau. \tag{E.9}$$

Now, recall that $\tau_i$ is the smallest value of $\tau$ subject to:

$$\frac{1}{N}\left|\{\mathbf{x} \in S^i_{\text{filtered}} : (p_i(\mathbf{x}))^2 > \tau\}\right| \geq \frac{10}{\alpha}\left(\mathbb{P}_{\mathbf{x} \sim S_{\mathcal{D}}}[B_0 \geq (p_i(\mathbf{x}))^2 > \tau] + 200\sqrt{d^{2k}\frac{\log N}{N}\log\frac{1}{\delta}}\right).$$

Therefore, for all values of $\tau$ smaller than $\tau_i$ we have

$$\frac{1}{N}\left|\left\{\mathbf{x} \in S^i_{\text{filtered}} : (p_i(\mathbf{x}))^2 > \tau\right\}\right| < \frac{10}{\alpha}\left(\underset{\mathbf{x}\sim S_{\mathcal{D}}}{\mathbb{P}}[B_0 \geq (p_i(\mathbf{x}))^2 > \tau] + 200\sqrt{d^{2k}\frac{\log N}{N}\log\frac{1}{\delta}}\right),$$

which combined with Equation E.9 implies

$$\frac{1}{N}\sum_{\mathbf{x}\in S^{i+1}_{\text{filtered}}}(p_i(\mathbf{x}))^2 \leq$$

$$\frac{10}{\alpha}\left(\left(200\sqrt{d^{2k}\frac{\log N}{N}\log\frac{1}{\delta}}\right)\tau_i + \int_{\tau=0}^{\infty}\underset{\mathbf{x}\sim S_{\mathcal{D}}}{\mathbb{P}}[B_0 \geq (p_i(\mathbf{x}))^2 > \tau]\, d\tau\right) =$$

$$\frac{10}{\alpha}\left(\left(200\sqrt{d^{2k}\frac{\log N}{N}\log\frac{1}{\delta}}\right)\tau_i + \underset{\mathbf{x}\sim S_{\mathcal{D}}}{\mathbb{E}}\left[p_i(\mathbf{x}))^2\cdot\mathbf{1}_{\mathbf{x}\leq B_0}\right]\right) \quad \text{(E.10)}$$

Additionally, since $S_{\mathcal{D}}$ is assumed to satisfy property (3) in Claim 1, and $\widehat{M}$ is assumed to satisfy Equation E.1, we have

$$\underset{\mathbf{x}\sim S_{\mathcal{D}}}{\mathbb{E}}\left[(p_i(\mathbf{x}))^2\cdot\mathbf{1}_{(p(\mathbf{x}))^2\leq B_0}\right] \leq \underset{\mathbf{x}\sim\mathcal{D}}{\mathbb{E}}[(p_i(\mathbf{x}))^2] + 0.01 \leq \frac{11}{10}(p_i)^\top\widehat{M}(p_i) + 0.01 \leq \frac{111}{100}. \quad \text{(E.11)}$$

Combining Equation E.10 and Equation E.11 we get

$$\frac{1}{N}\sum_{\mathbf{x}\in S^{i+1}_{\text{filtered}}}(p_i(\mathbf{x}))^2 \leq \frac{10}{\alpha}\left(\left(200\sqrt{d^{2k}\frac{\log N}{N}\log\frac{1}{\delta}}\right)\tau_i + \frac{111}{100}\right).$$

Recall that $S^{i+1}_{\text{filtered}}$ is a subset of $S^0_{\text{filtered}}$ and therefore for every element $\mathbf{x}$ of $S^{i+1}_{\text{filtered}}$ it is the case that $(p_i(\mathbf{x}))^2 \leq B \cdot (p_i)^\top\widehat{M}(p_i) \leq B$, which combinded with the definition of $\tau_i$ implies that $\tau_i \leq B = \frac{4d^{3k}}{\epsilon}$. Substituting this above allows us to conclude the claim. $\qquad\square$

## E.2   Outlier Removal in the PQ setting

We restate Theorem 3.1:

**Theorem E.2.** *There exists an algorithm that, given sample access to an arbitrary distribution $\mathcal{D}'$ over $\mathbb{R}^d$, sample access to a $k$-tame probability distribution $\mathcal{D}$ over $\mathbb{R}^d$, parameters $\epsilon, \alpha, \delta$ in $(0,1)$, and a positive integer $k$, runs in time $\text{poly}\left(\frac{(kd)^k}{\epsilon}\log\frac{1}{\delta}\right)$ and outputs a succinct $\text{poly}\left(\frac{(kd)^k}{\epsilon}\log\frac{1}{\delta}\right)$-time-computable description of a function $g : \mathbb{R}^d \to \{0,1\}$ that satisfies the following properties with probability at least $1 - \delta$:*

- ***Degree-$k$ spectral $\frac{200}{\alpha}$-boundedness:** For every polynomial $p$ of degree at most $k$ it is the case that*

$$\underset{\mathbf{x}\sim\mathcal{D}'}{\mathbb{E}}\left[(p(\mathbf{x}))^2 g(\mathbf{x})\right] \leq \frac{200}{\alpha}\underset{\mathbf{x}\sim\mathcal{D}}{\mathbb{E}}[(p(\mathbf{x}))^2].$$

- ***$(\alpha, \epsilon)$-validity:** we have*

$$\underset{\mathbf{x}\sim\mathcal{D}}{\mathbb{P}}[g(\mathbf{x}) = 0] \leq \alpha\, dist_{TV}(\mathcal{D}', \mathcal{D}) + \frac{\epsilon}{2},$$

*which in particular implies that $\mathbb{P}_{\mathbf{x}\sim\mathcal{D}'}[g(\mathbf{x}) = 0] \leq (1 + \alpha)dist_{TV}(\mathcal{D}', \mathcal{D}) + \epsilon/2$.*

We also restate Algorithm 1 as follows:

1. Draw sets $S_{\mathcal{D}}$ and $S_{\mathcal{D}'}$ of $N = C\left(\frac{(kd)^k B}{\epsilon}\log\frac{1}{\delta}\right)^C$ samples from distributions $\mathcal{D}$ and $\mathcal{D}'$ respectively, where $C$ is a sufficiently large absolute constant.

2. Run the algorithm of Theorem E.1 on the input $S_{\mathcal{D}'}$. Set $i_{\max}$ to be the number of iterations of the main loop in the algorithm of Theorem E.1, and store the polynomials $\{p_i\}$, values $\{\tau_i\}$ computed at each iteration of the main loop, as well as the matrix $\widehat{M}$.

3. Output the function $g : \mathbb{R}^d \to \{0, 1\}$ that does the following given an input $\mathbf{x}$ in $\mathbb{R}^d$:

   (a) If $\max_{\substack{p \text{ of degree } k \\ p^\top \widehat{M} p \le 1}} (p(\mathbf{x}))^2 > B_0$, then $g(\mathbf{x})$ is defined to be 0. (See Section E.1.1 to see how to compute this quantity in time $\mathrm{poly}(N)$.)

   (b) If for some $i$ it is the case that $(p_i(\mathbf{x}))^2$ is greater than $\tau_i$, then $g(\mathbf{x})$ is defined to be 0.

   (c) Otherwise, $g(\mathbf{x})$ is defined to be 1.

It is immediate from Theorem E.1 that the algorithm above runs in time $\mathrm{poly}\left(\frac{(kd)^k B}{\epsilon} \log \frac{1}{\delta}\right)$ with probability at least $1 - \delta$. Furthermore, we also see that the function $g$ can be described using $\mathrm{poly}\left(\frac{(kd)^k B}{\epsilon} \log \frac{1}{\delta}\right)$ bits and can be computed using this description on a given input $\mathbf{x}$ in time $\mathrm{poly}\left(\frac{(kd)^k B}{\epsilon} \log \frac{1}{\delta}\right)$.

We need the following claim bounding the number of iterations in the algorithm of Theorem E.1, proof of which is deferred until the end of this section. We remark that for Theorem E.1 we bounded the total number of iterations by $N$, but in this section we will need a bound that depends only on $d$, $k$ and $\epsilon$ and not on $N$.

**Claim 6.** *If the set $S_{\mathcal{D}}$ satisfies the condition of Claim 1, then the number of iterations $i_{\max}$ of the main while loop in the algorithm of Theorem E.1 satisfies $i_{\max} = O\left(kd^k \log(B_0 d)\right)$.*

We now proceed to use Claim 6 to argue the spectral $\frac{200}{\alpha}$-boundedness and $(\alpha, \epsilon/2)$-validity. As the first step, we show the following:

**Observation E.3.** *There exists a function class $\mathcal{G}$ with a VC dimension of at most $O(i_{\max} d^{2k} \log(i_{\max}))$, such that all possible values of the function $g$ belong to $\mathcal{G}$.*

*Proof.* The function $g$ is necessarily a logical AND of at most $i_{\max} + 1$ functions, one of which is the function indicator of a ball in $\mathbb{R}^d$ and the other $i_{\max}$ are logical OR-s of pairs of degree-$2k$ polynomial threshold functions. Combining this with Fact A.5 and Fact A.4 yields the observation. $\square$

We start with arguing the $(\alpha, \epsilon/2)$-validity, as well as the stronger condition of Remark 3.3 (implied by Equation E.13):

**Claim 7.** *With probability at least $1 - \delta/2$ over the choice of the sets $S_{\mathcal{D}}$ and $S_{\mathcal{D}'}$, we have*

$$\mathbb{P}_{\mathbf{x} \sim \mathcal{D}}[g(\mathbf{x}) = 0] \le \alpha \, dist_{TV}(\mathcal{D}', \mathcal{D}) + \frac{\epsilon}{2}. \tag{E.12}$$

*Furthermore, for $\sigma > \alpha/2$ and any distribution $\mathcal{D}''$ that is $1/\sigma$-smooth w.r.t. $\mathcal{D}$, (i.e. for any measurable set $T \subset \mathbb{R}^d$ we have $\mathbb{P}_{\mathbf{x} \sim \mathcal{D}''}[\mathbf{x} \in T] \le \frac{1}{\sigma} \mathbb{P}_{\mathbf{x} \sim \mathcal{D}}[\mathbf{x} \in T]$) it is the case that*

$$\mathbb{P}_{\mathbf{x} \sim \mathcal{D}}[g(\mathbf{x}) = 0] \le \frac{\alpha}{\sigma} \, \mathrm{d_{TV}}(\mathcal{D}'', \mathcal{D}') + \frac{\epsilon}{2}, \tag{E.13}$$

*Proof.* With probability at least $1 - \delta/4$ the set $S_{\mathcal{D}}$ satisfies the condition of Claim 1. Assuming this, we have:

$$\mathbb{P}_{\mathbf{x} \sim \mathcal{D}}[g(\mathbf{x}) = 0] \le \sum_{i=0}^{i_{\max}} \mathbb{P}_{\mathbf{x} \sim \mathcal{D}}\left[(p_i(\mathbf{x}))^2 \ge \tau_i\right] \le$$

$$\sum_{i=0}^{i_{\max}} \mathbb{P}_{\mathbf{x} \sim S_{\mathcal{D}}}\left[(p_i(\mathbf{x}))^2 \ge \tau_i\right] + i_{\max}\left(200 \sqrt{d^{2k} \frac{\log N}{N} \log \frac{1}{\delta}}\right), \quad \text{(E.14)}$$

where the last step used the premise that $S_{\mathcal{D}}$ satisfies the condition of Claim 1. Recalling that by Observation E.3 the function $g$ belongs to a function class with VC dimension of $O(i_{\max} d^{O(k)} \log(i_{\max}))$, and combining this with Fact A.6, we see that with probability at least $1 - \delta/4$

$$\mathbb{P}_{\mathbf{x} \sim \mathcal{D}'}[g(\mathbf{x}) = 0] \ge \mathbb{P}_{\mathbf{x} \sim S_{\mathcal{D}'}}[g(\mathbf{x}) = 0] - O\left(\sqrt{\frac{i_{\max} d^{O(k)} \log(i_{\max}) \log N}{N}} \log \frac{1}{\delta}\right)$$

Recalling the definition of $g$, we see that for $\mathbf{x}$ in $S_{\mathcal{D}'}$, we have $g(\mathbf{x}) = 0$ when for some iteration $i$, the point $\mathbf{x}$ is in the set $\{\mathbf{x} \in S_{\text{filtered}}^i : (p_i(\mathbf{x}))^2 > \tau_i\}$, we conclude that

$$\mathbb{P}_{\mathbf{x} \sim \mathcal{D}'}[g(\mathbf{x}) = 0] \geq$$
$$\sum_{i=0}^{i_{\max}} \frac{|\{\mathbf{x} \in S_{\text{filtered}}^i : (p_i(\mathbf{x}))^2 > \tau_i\}|}{N} - O\left(\sqrt{\frac{i_{\max} d^{O(k)} \log(i_{\max}) \log N}{N}} \log \frac{1}{\delta}\right) \quad \text{(E.15)}$$

Now, we recall that for every iteration $i$ we have:

$$\frac{1}{N}\left|\{\mathbf{x} \in S_{\text{filtered}}^i : (p_i(\mathbf{x}))^2 > \tau_i\}\right| \geq \frac{10}{\alpha}\left(\mathbb{P}_{\mathbf{x} \sim S_{\mathcal{D}}}[(p_i(\mathbf{x}))^2 > \tau_i] + 200\sqrt{d^{2k}\frac{\log N}{N} \log \frac{1}{\delta}}\right),$$

and therefore:

$$\sum_{i=0}^{i_{\max}} \mathbb{P}_{\mathbf{x} \sim S_{\mathcal{D}}}[(p_i(\mathbf{x}))^2 > \tau_i] \leq$$
$$\frac{\alpha}{10}\left(\sum_{i=0}^{i_{\max}} \frac{1}{N}\left|\{\mathbf{x} \in S_{\text{filtered}}^i : (p_i(\mathbf{x}))^2 > \tau_i\}\right|\right) + 200 i_{\max}\sqrt{\frac{d^{2k}\log N}{N} \log \frac{1}{\delta}}. \quad \text{(E.16)}$$

Thus, combining Equation E.14, Equation E.15 and Equation E.16 we get:

$$\mathbb{P}_{\mathbf{x} \sim \mathcal{D}}[g(\mathbf{x}) = 0] \leq \frac{\alpha}{10} \underbrace{\left(\mathbb{P}_{\mathbf{x} \sim \mathcal{D}'}[g(\mathbf{x}) = 0]\right)}_{\leq \text{dist}_{\text{TV}}(\mathcal{D}', \mathcal{D}) + \mathbb{P}_{\mathbf{x} \sim \mathcal{D}}[g(\mathbf{x}) = 0]} + O\left(i_{\max}\sqrt{\frac{d^{O(k)}\log N}{N} \log \frac{1}{\delta}}\right). \quad \text{(E.17)}$$

We now the bound on $i_{\max}$ from Claim 6, and recall that $N = C\left(\frac{(kd)^k B}{\epsilon} \log \frac{1}{\delta}\right)^C$. Overall, we see that for sufficiently large absolute constant $C$ the error term above is upper-bounded by $\epsilon/10$, so

$$\mathbb{P}_{\mathbf{x} \sim \mathcal{D}}[g(\mathbf{x}) = 0] \leq \frac{\alpha}{10}\left(\text{dist}_{\text{TV}}(\mathcal{D}', \mathcal{D}) + \mathbb{P}_{\mathbf{x} \sim \mathcal{D}}[g(\mathbf{x}) = 0]\right) + \frac{\epsilon}{10} \quad \text{(E.18)}$$

Rearranging the inequality above and recalling that $\alpha < 1$, we conclude that Equation E.12 holds.

Finally, we prodeed to argue Equation E.13. For $\sigma > \alpha/2$ suppose that the distribution $\mathcal{D}''$ is $1/\sigma$-smooth w.r.t. $\mathcal{D}$, (i.e. for any measurable set $T \subset \mathbb{R}^d$ we have $\mathbb{P}_{\mathbf{x} \sim \mathcal{D}''}[\mathbf{x} \in T] \leq \frac{1}{\sigma} \mathbb{P}_{\mathbf{x} \sim \mathcal{D}}[\mathbf{x} \in T]$). Then, from Equation E.17 we have

$$\underbrace{\mathbb{P}_{\mathbf{x} \sim \mathcal{D}}[g(\mathbf{x}) = 0]}_{\substack{\leq \sigma \mathbb{P}_{\mathbf{x} \sim \mathcal{D}''}[g(\mathbf{x}) = 0] \\ \text{by } \sigma\text{-smoothness}}} \leq \frac{\alpha}{10} \underbrace{\left(\mathbb{P}_{\mathbf{x} \sim \mathcal{D}'}[g(\mathbf{x}) = 0]\right)}_{\leq \text{dist}_{\text{TV}}(\mathcal{D}'', \mathcal{D}) + \mathbb{P}_{\mathbf{x} \sim \mathcal{D}''}[g(\mathbf{x}) = 0]} + \underbrace{O\left(i_{\max}\sqrt{\frac{d^{O(k)}\log N}{N} \log \frac{1}{\delta}}\right)}_{\substack{\leq \epsilon/10 \\ \text{for constant } \tilde{C} \text{ sufficiently large.}}}. \quad \text{(E.19)}$$

Rearranging the inequality above and recalling that $\alpha < 1$ and $\sigma > \alpha/2$, we conclude that Equation E.13 holds.

$\square$

Now, we argue the spectral $\frac{200}{\alpha}$-boundedness. Recall that with probability at least $\delta/20$ the matrix $\widehat{M}$ satisfies Equation E.1, which we will henceforth assume. Also recall that Claim 4 says that for every polynomial $p$ of degree at most $k$ satisfying $\mathbb{E}_{\mathbf{x} \sim \mathcal{D}}[(p(\mathbf{x}))^2] \leq 1$, the set $S_{\text{accept}}$ given by the algorithm in Theorem E.1 satisfies $\frac{1}{N}\sum_{\mathbf{x} \in S_{\text{accept}}} p(\mathbf{x})^2 \leq \frac{100}{\alpha}$. By inspecting the definition of the function $g$, we see that for $\mathbf{x}$ in $S_{\mathcal{D}'}$ we have $g(\mathbf{x}) = 1$ if an only if $\mathbf{x}$ is in $S_{\text{accept}}$. Therefore,

$$\max_{\substack{p \text{ of degree } k \text{ s.t:} \\ \mathbb{E}_{\mathbf{x} \sim \mathcal{D}}[(p(\mathbf{x}))^2] \leq 1}} \left[\mathbb{E}_{\mathbf{x} \sim S_{\mathcal{D}'}}[g(\mathbf{x})p(\mathbf{x})^2]\right] \leq \frac{100}{\alpha} \quad \text{(E.20)}$$

In order to conclude the spectral $\frac{200}{\alpha}$-boundedness condition we need to be able to conclude that the equation above is likely to generalize, i.e. it approximately holds when one replaces the expectation w.r.t. $S_{\mathcal{D}'}$ with the expectation w.r.t. the distribution $\mathcal{D}'$. To show this, we first recall that via Observation E.3 the function $g$ belongs to a function class $\mathcal{G}$ with a VC dimension of at most $O(i_{\max} d^{2k} \log(i_{\max}))$. We also see that $g(\mathbf{x}) = 0$ for all $\mathbf{x}$ satisfying $\max\limits_{p:\ \mathbb{E}_{\mathbf{x}\sim\mathcal{D}}[(p(\mathbf{x}))^2]\leq 1} (p(\mathbf{x}))^2 > 10B_0$, because the matrix $\widehat{M}$ satisfies Equation E.1 and therefore if $\mathbb{E}_{\mathbf{x}\sim\mathcal{D}}[(p(\mathbf{x}))^2] \leq 1$ and $(p(\mathbf{x}))^2 > 10B_0$, then also $(\sqrt{0.9}p)^\top M (\sqrt{0.9}p) \leq 1$ and $\sqrt{0.9}p(\mathbf{x}) > B_0$, which implies that $g(\mathbf{x}) = 0$ by the definition of $g$. We show in Lemma B.2 that with probability at least $1 - \delta/20$ such function classes satisfy

$$
\sup_{\substack{g\in\mathcal{G} \\ p \text{ of degree } k \text{ s.t:\ } \mathbb{E}_{\mathbf{x}\sim\mathcal{D}}(p(\mathbf{x}))^2 \leq 1}} \left| \frac{1}{N} \sum_{\mathbf{x}\sim S_{\mathcal{D}'}} \left[ f(\mathbf{x})(p(\mathbf{x}))^2 \right] - \mathbb{E}_{\mathbf{x}\sim\mathcal{D}'} \left[ f(\mathbf{x})(p(\mathbf{x}))^2 \right] \right| \leq
$$

$$
O\left( \sqrt{B_0 \left( i_{\max} d^{2k} \log(i_{\max}) \, d^{2k} \frac{\log N}{N} \log \frac{1}{\delta} \right)^{1/4}} \right) \leq 1 \leq \frac{1}{\alpha}, \quad \text{(E.21)}
$$

where the penultimate inequality above is achieved by substituting the bound $i_{\max} = O\left(kd^k \log B_0 d\right)$ from Claim 6 into the expression above, substituting $B_0 = \frac{4d^{3k}}{\epsilon}$, recalling that $N = C\left(\frac{(kd)^k}{\epsilon} \log \frac{1}{\delta}\right)^C$ and taking $C$ to be a sufficiently large absolute constant. Combining Equation E.21 with Equation E.20 we conclude that with probability at least $1 - \delta/20$ it is the case that

$$
\max_{\substack{p \text{ of degree } k \text{ s.t.} \\ \mathbb{E}_{\mathbf{x}\sim\mathcal{D}}[(p(\mathbf{x}))^2] \leq 1}} \mathbb{E}_{\mathbf{x}\sim\mathcal{D}'} \left[ (p(\mathbf{x}))^2 g(\mathbf{x}) \right] \leq \frac{101}{\alpha},
$$

Overall, with probability at least $1 - \delta$ the function $g$ satisfies spectral $\frac{200}{\alpha}$-boundedness, $(\alpha, \epsilon)$-validity, as well as the required run-time bound.

Finally, we come back to Claim 6, proving which concludes this section.

*Proof of Claim 6.* Let $i$ be an iteration such that $i < i_{\max}$. Since the while loop did not terminate on step $i$, we have

$$
\sum_{\mathbf{x}\in S^i_{\text{filtered}}} (p_i(\mathbf{x}))^2 > \frac{100}{\alpha}. \quad \text{(E.22)}
$$

At the same time, Claim 5 implies that

$$
\frac{1}{N} \sum_{\mathbf{x}\in S^{i+1}_{\text{filtered}}} (p_i(\mathbf{x}))^2 \leq \frac{20}{\alpha} \quad \text{(E.23)}
$$

Let $m \leq d^k$ denote the dimension of the vector space of degree-$k$ polynomials. For values of $i$ between 0 and $i_{\max}$ and for values of $j$ between 1 and $m$, let the collection of polynomials $\{R^i_j\}$ and non-negative real values $\{\lambda^i_j\}$ be defined as

$$
R^i_j = \underset{\substack{R \text{ of degree } k \text{ s.t:} \\ \forall j' < j:\ (R^i_{j'})^\top \widehat{M} R = 0 \\ R^\top \widehat{M} R \leq 1}}{\arg\max} \frac{1}{N} \sum_{\mathbf{x}\in S^i_{\text{filtered}}} \left[ (R(\mathbf{x}))^2 \right] \qquad \lambda^i_j = \frac{1}{N} \sum_{\mathbf{x}\in S^i_{\text{filtered}}} \left[ (R^i_j(\mathbf{x}))^2 \right] \quad \text{(E.24)}
$$

In particular[7], we have $p_i = R_1^i$. We will use the quantity $\varphi_i := \sum_{j=1}^m \lambda_j^i$ as a potential function, for which we have:

$$\varphi_i - \varphi_{i+1} =$$

$$\frac{1}{N} \sum_{j=1}^m \sum_{\mathbf{x} \in S_{\text{filtered}}^i \setminus S_{\text{filtered}}^{i+1}} (R_j^i(\mathbf{x}))^2 \geq \frac{1}{N} \sum_{\mathbf{x} \in S_{\text{filtered}}^i \setminus S_{\text{filtered}}^{i+1}} (R_1^i(\mathbf{x}))^2 = \frac{1}{N} \sum_{\mathbf{x} \in S_{\text{filtered}}^i \setminus S_{\text{filtered}}^{i+1}} (p_i(\mathbf{x}))^2 =$$

$$= \sum_{\mathbf{x} \in S_{\text{filtered}}^i} (p_i(\mathbf{x}))^2 - \sum_{S_{\text{filtered}}^{i+1}} (p_i(\mathbf{x}))^2 \geq \lambda_1^i - \frac{20}{\alpha} \quad \text{(E.25)}$$

Where in the end we substituted Equation E.23. Since $\lambda_1^i$ equals to $\frac{1}{N} \sum_{\mathbf{x} \in S_{\text{filtered}}^i} \left[ (p_i(\mathbf{x}))^2 \right]$ and has a value of at least $100/\alpha$ by Equation E.22, the inequality above allows us to conclude

$$\varphi_{i+1} \leq \sum_{j=1}^m \lambda_j^i - 0.8\lambda_1^i \leq \sum_{j=1}^m \lambda_j^i - \frac{0.8}{m} \sum_j \lambda_j^i \leq \left(1 - \frac{0.9}{d^k}\right) \varphi_i \quad \text{(E.26)}$$

We now combine the inequality above with the following two observations:

- We have
$$\varphi_{i_{\max}-1} > \frac{100}{\alpha} > 1,$$
because the algorithm did not terminate in the $(i_{\max}-1)$-th iteration, and therefore Equation E.22 holds.

- We have
$$\varphi_0 = \frac{1}{N} \sum_{j=1}^m \sum_{\mathbf{x} \in S_{\text{filtered}}^0} (R_j^i(\mathbf{x}))^2 \leq B_0 m,$$
where the last inequality follows from the fact that every element $\mathbf{x}$ in $S_{\text{filtered}}^0$ satisfies
$$\max_{p \text{ of degree } k: \; p^\top \widehat{M} p \leq 1} (p(\mathbf{x}))^2 \leq B_0.$$

Overall, the two bounds above, together with Equation E.26 allow us to conclude that:

$$i_{\max} \leq O\left(d^k \log(B_0 m)\right) = O\left(k d^k \log(B_0 d)\right),$$

where the last step follows by substituting the definitions of $m$ and $\epsilon$. $\qquad\square$

---

[7]Speaking precisely, there might be multiple choices for the collection of the polynomials $\{R_j^\top\}$. In this case, still, we can choose these polynomials without loss of generality in such a way that $p_i = R_1^i$.

