# OpenReview forum: "Tolerant Algorithms for Learning with Arbitrary Covariate Shift"
_NeurIPS.cc/2024/Conference — NeurIPS 2024 spotlight_

### Official Review · Reviewer_QNbu · 2024-07-08

**Soundness:** 3
**Presentation:** 2
**Contribution:** 2
**Rating:** 6
**Confidence:** 3

**Summary:**

This paper studies the problem of PAC learning with covariate shift. In particular, it examines two specific learning frameworks: one is PQ learning, where a learner is allowed to "absent" from some testing samples but is required to have good accuracy for the retained samples; the other is TDS learning, where one is allowed to completely absent if the testing distribution is detected to be far from the training distribution. From a technical point of view, the paper is restricted to cases when the covariate distribution is Gaussian or uniform and when the concept class is nicely approximated by polynomials of low degree (i.e., the so-called "sandwiching" functions). The main contribution appears to extend prior results to handle arbitrary fractions of outliers and provide tighter bounds.

**Strengths:**

While I'm not an expert in PAC learning using polynomial regression, this paper appears to provide some interesting technical results. The outlier removal algorithm introduced in Theorem 3.1 seems to be an interesting technical primitive for handling outliers in low-degree polynomial regressions. Although the settings considered in this paper are fairly restrictive, the results seem to provide a valuable step toward understanding learning with abstention in covariate shift.

**Weaknesses:**

The main weakness of the paper, in my opinion, is the presentation. The paper is very hard to read, especially for those who are not immediately familiar with the relevant literature.

I outline the following specific comments:

1. The main text as written does not provide much technical information. For example, Theorem 3.1 is supposed to be the main technical ingredient of the paper, but the proof overview provides nearly no information. Due to the limited time period of the NeurIPS review, I don't have time to delve into the technical details provided in the appendix. I suggest the authors provide a much more detailed overview in the main text. It does not necessarily need to include all the technical details, but it should provide enough information on the logical flow. Perhaps the "Comparison with [DKS18]" section can be omitted to save space?

2. The theorem statements are sometimes framed very sloppily. For example, the $\lambda$ in Theorem 4.1 was never defined (though it appears in the proof); in Lemma B.2., the sentence "we have f(x) = 0 for all x such that..." does not make sense to me, as the subsequent quantifier has nothing to do with x, referring to f(x)=0 always. Since there are too many sloppy statements like this, I lack the energy to check the details carefully.

3. The authors claim to provide "efficient learning algorithms," but all the computational complexities scale exponentially with respect to the error parameters $\epsilon$. Did I miss something? Can this parameter be selected as a constant and be boosted in some way?

4. Is there a specific reason why Theorem 3.1 is stated for the second moment? Is it due to Lemma B.1?

Given these reasons, although I believe this paper has some interesting technical contributions, it would require substantial revision to be suitable for publication at NeurIPS.

**Questions:**

See above.

---

> ### Author Rebuttal · Authors · 2024-08-05
>
> We thank the reviewer for their time and effort.
>
> **Question 1.** We note that, while we provide most of the technical details of our proofs in the appendix, we do provide a high-level explanation of our results in the main part, including Theorem 3.1 (see lines 189–223). In fact, the comparison with [DKS18] partly explains the technical differences between Theorem 3.1 and results on outlier removal from prior work. Given the reviewer’s feedback, however, we will add some further technical details in the main paper.
>
> **Question 2.** We respectfully disagree with the reviewer’s comment that our statements lack precision. Note that the parameter $\lambda$ is defined in the preliminaries (lines 146–148). Moreover, in Lemma B.2, the premise regarding $f$ is that $f(x) = 0$ for any $x$ that satisfies some property (the one described in line 693, i.e., that $p(x)^2$ is more than $B$ for some polynomial $p$ with low second moment) and not all $x$. We will clarify these points in the main paper and we are happy to address any further specific concerns that the reviewer may have.
>
> **Question 3.** The algorithms we propose are efficient in the dimension of the input, which is a standard goal in learning theory (see, for example, the classical work of [KKMS08]). In the presence of label noise, the exponential dependence on the parameter $\epsilon$ cannot be removed, even if one assumes that there is no distribution shift (see [DKPZ21]). Additionally, when there is no label noise, the algorithm of Theorem 4.3 for PQ learning of halfspaces is quasi-polynomial in all parameters and is optimal in the SQ framework (see [KSV24a]).
>
> **Question 4.** Theorem 3.1 is stated for the second moment for two reasons. First, using the second moment enables the use of spectral methods for the outlier removal procedure and is a common approach for prior results on outlier removal. Second, it is sufficient for our purposes, since the difference of the upper and lower $\mathcal{L}_2$ sandwiching approximators is itself a polynomial with low second moment under the target distribution. Subsequently, we can use the outlier removal procedure to find a subset of the input set over which the bound on the squared difference of the sandwiching approximators is preserved.
>
> *[KKMS08] Adam Tauman Kalai, Adam R Klivans, Yishay Mansour, and Rocco A Servedio. Agnostically learning halfspaces. SIAM Journal on Computing, 37(6):1777–1805, 2008.*
>
> *[DKPZ21] Diakonikolas, Ilias, Daniel M. Kane, Thanasis Pittas, and Nikos Zarifis. "The optimality of polynomial regression for agnostic learning under gaussian marginals in the SQ model." In Conference on Learning Theory, pp. 1552-1584. PMLR, 2021.*
>
> *[KSV24a] Adam R Klivans, Konstantinos Stavropoulos, and Arsen Vasilyan. Learning intersections of halfspaces with distribution shift: Improved algorithms and sq lower bounds. 37th Annual Conference on Learning Theory, COLT, 2024.*

---

> > ### Comment · Reviewer_QNbu · 2024-08-07
> >
> > I thank the authors for the clarifications. I raise my rating to 6. Since I'm not an expert in this particular field, I will have no objection if the other reviewers lean towards acceptance.

---

### Official Review · Reviewer_wPZr · 2024-07-11

**Soundness:** 4
**Presentation:** 4
**Contribution:** 3
**Rating:** 8
**Confidence:** 4

**Summary:**

The authors consider the problem of efficiently learning  a concept class under distribution shift, i.e. in the setting where the training data and testing data are drawn from different distributions. They study two frameworks: PQ learning and TDS learning. In the former, the learner is allowed to abstain from classifying a part of the test samples. In the latter, the learner is allowed to abstain entirely if the test distribution is 'visibly' different from the training distribution.

The paper has two main contributions. First, in the PQ setting, the authors obtain the first dimensionally-efficient learning algorithms, under the assumption that the training data is nicely distributed (Gaussian or uniform on the cube) and the concept class is simple (intersection of halfspaces or low-complexity formulas).
Second, in the TDS setting, under the same assumptions, they provide the first efficient algorithms that tolerate small distribution shift in TV-distance. This generalizes earlier results which only tolerate shift 'in the moments' of the distribution (which is a weaker notion).

The proof of these results consists roughly of two steps. First, the authors adapt/improve an existing spectral technique for outlier-removal to show that the test data can be pruned so that it satisfies a 'low-degree spectral boundedness property', without removing too many samples. Second, they show that this spectral boundedness property suffices to apply low-degree polynomial regression to the PQ/TDS-learning problems (in certain settings). In order to do so, they rely on the notion of 'L2-sandwiching polynomials' of [KSV24b]. The important distinction w.r.t. [KSV24b] is that, there, the authors rely on a 'moment-matching property' (which is stronger than spectral boundedness, and in particular, not always satisfied even if the training and test data are close in TV-distance).

**Strengths:**

- The paper achieves strong and novel results in an area which I believe to be of interest to the Neurips community. It falls in a line of research focussing on relaxing (or testing) distributional assumptions in algorithmic learning theory, which has recently received a lot of interest. In particular, I like that the paper goes beyond the 'moment-matching' approach of earlier works.

- I find the combination of ideas from [KSV24b] and outlier removal to obtain the main results very insightful.

- The paper is very well written: the key technical ideas are exposed clearly, and it is easy even for the non-expert to graps the main concepts. Furthermore, the results and some of their consequences are positioned clearly in the existing literature. Lastly, the technical background (on learning models, previous algorithms etc.) is well presented.

- The work leaves open some directions for future research, and I think it is likely that its ideas will be applied to obtain future results in this area.

**Weaknesses:**

- Arguably, much of the technical 'heavy-lifting' to obtain the results of this paper is done by the 'L2-sandwiching' of [KSV24b], and to a lesser extend the outlier removal technique of [DKS18] (which the authors do significantly improve on).

- The results apply only to very simple concept classes. It would have been nice to have guarantees in more complicated settings as well, e.g. for (low-degree) polynomial threshold functions. Similarly, the distributional assumptions on the training data are quite restrictive (although it should be noted that this is not uncommon in the literature).

**Questions:**

- l62-63: In testable learning, one does not test whether the underlying distribution is actually Gaussian (which is not possible), but rather whether it shares some important characterics with a Gaussian distributions (e.g. its low-degree moments match). I guess the word 'indirectly' is meant to allude to this distinction, but I think this phrasing could be confusing.

- l111-117: It is not clear to me from this paragraph if any work on *distribution-specific* PQ-learning was done before this paper. In particular, it think it would be good to clarify if  1) this work gives the first *efficient* distribution-specific PQ algorithm (but inefficient *distribution-specific* algorithms had been considered, or 2) this work is the first to consider distribution-specific PQ-learning at all.

**Limitations:**

Adequately addressed

---

> ### Author Rebuttal · Authors · 2024-08-05
>
> We thank the reviewer for their time and appreciation of our work.
>
> - Lines 62-63: We will clarify this point, thank you for pointing out.
>
> - Lines 111-117: Our work is indeed the first to consider distribution-specific PQ-learning at all. Prior work involved reductions to other learning primitives like agnostic or reliable agnostic learning, but the reductions did not preserve the marginal distributions and, hence, the resulting algorithms were inherently distribution-free.

---

### Official Review · Reviewer_tKBR · 2024-07-12

**Soundness:** 2
**Presentation:** 2
**Contribution:** 3
**Rating:** 6
**Confidence:** 4

**Summary:**

This work proposes methods for learning under covariance shifts. In particular, it studies PQ learning and TDS learning models. It provides an algorithm based on filtering technique. Given sample access to two distributions, the algorithm produces a filter that rejects points from the second distribution which attains large values on some degree-k polynomial, and does not reject many values of the first distribution.

This algorithm is then applied to learning some function classes (halfspaces and classes with low sandwiching degree) in PQ learning model and to tolerant TDS learning model. In the former model authors obtain a computationally-efficient algorithm, and in the latter an algorithm which does not reject a distribution even if there exists a small but non-negligible shift.

**Strengths:**

1. Paper provides novel results in the PQ learning and TDS learning. Solving these learning models is important, since in practice algorithms need to perform well under covariate shifts.
2. Core algorithm of the paper is versatile, as the authors show how to apply it to multiple other learning settings (e.g. learning with nasty noise).
3. For TDS learning, prior works rejected distribution with a very small distributional shift. This work provides an algorithm which gives non-trivial results even if the distance between distributions is larger.
4. This work suggests a novel technique to bound number of iterations of their algorithm through the use of potential function.
5. The main text of the paper is well-written.

**Weaknesses:**

I find the main weakness of the paper is the quality of the proofs in the appendix. They are often inaccurately written and require a lot of time to understand the arguments because of the typos. But it seems that there are some inaccuracies in the proofs of the core results beyond typos, which I highlight in the 'Questions' section.

**Questions:**

My main question is about Appendix B, which I believe is crucial for the proof of all technical theorems.
I find this section not carefully written with inconsistent notation and typos, which makes it hard to follow the arguments.

Next, I focus on the parts which were not merely typos:
1. I do not follow equation (B.2). First of all, it should be square of the Frobenius norm. But still second and third step is unclear. Could the authors explain these steps in more detail?
2. If I understand correctly, equation (B.3) requires $2k$-tameness, instead of $k$-tameness. This is because $r_j(x)r_{j’}(x)$ are of degree $2k$.
3. Equation (B.4): I do not follow first inequality, where does first $\Delta$ come from? Since $f(x)(p(x))^2 = \sum_{j = 0}^{B / \Delta} f(x)(p(x))^2 \mathbb{I}(j\Delta \leq (p(x))^2 < (j+1)\Delta)$, we obtain that $\lvert E_{x\sim S} f(x) (p(x))^2 - E_{x\sim D'} f(x)(p(x))^2\rvert \leq \sum_j \lvert E_{x\sim S} f(x) (p(x))^2 \mathbb{I}_j  - E_D f(x) (p(x))^2 \mathbb{I}_j \rvert$, and there is no additive $\Delta$? $(\mathbb{I}_j := \mathbb{I}(j\Delta \leq p(x)^2 < (j+1)\Delta)$
4. Line 922: Can authors clarify why there exists such $\tau_i$? it seems that if I increase $\tau$, then the value on the left decreases to 0, while the value on the right stays positive?

I will consider changing my score when authors address these questions.

**Limitations:**

There are no ethical limitations in this work.

---

> ### Author Rebuttal · Authors · 2024-08-05
>
> We thank the reviewer for their constructive feedback. We will carefully go through our proofs in the appendix to fix all typos for future revisions. Regarding the specific questions of the reviewer:
>
> **Question 1.** You are correct that Equation (B.2) should have the square of Frobenius norm on the left side. Thank you for pointing out that typo. This will in turn slightly change the bound of the equation between lines 677 and 678, where we will instead use the fact that $\mathbb{E}_{S_i}[||I-M^{-1/2}M_iM^{-1/2}||_2] \le \sqrt{ \mathbb{E} _{S_i}[||I-M^{-1/2}M_iM^{-1/2}||_F^2]} \le \frac{(dk)^{O(k)}}{N^{1/4}}$. The statement of line 678 is still true for some sufficiently large constant $C$.
>
> We now give the omitted details for Equation (B.2). We start with the second equality. We have $\mathbb{E}_ {x\sim D}[ r_{j}(x) r_{j'}(x) ] = e_{j}^{T} e_{j'}$, which is zero if $j=j’$ and 1 otherwise (by the equation starting directly after line 669). We also have $e_j^T M^{-1/2}=r_j^T$ and $M^{-1/2} e_{j’}= r_{j’}$ and therefore $e_j^T M^{-1/2} M_i M^{-1/2} e_{j’}= r_j^T M_i  r_{j’}$ which equals to $\mathbb{E}_ {x \sim S_i}[r_{j}(x) r_{j'}(x)]$.
>
> Now we explain the third equality, after adding $\mathbb{E}_ {S_i}$ (the expectation over the random selection of the elements of $S_i$) in the beginning of the second line, which is missing due to a typo. Since the set $S_i$ is composed from i.i.d. samples from $D$, the quantity $\mathbb{E}_ {S_i}(\mathbb{E}_ {x \sim D}[r_{j}(x) r_{j'}(x)]-\mathbb{E}_ {x \sim S_i}[r_{j}(x) r_{j'}(x)])^2$ equals to the variance of the random variable $\mathbb{E}_ {x \sim S_i}[r_{j}(x) r_{j'}(x)] = \frac{1}{|S_i|}\sum_{x\in S_i}r_{j}(x) r_{j'}(x)$ (the randomness is over the choice of $S_i$). Since the choice of elements of $S_i$ is i.i.d. from $D$, we can use the fact that the variance of a sum of i.i.d. variables  equals to the sum of their variances. Overall, we have that the variance of $\mathbb{E}_ {x \sim S_i}[r_{j}(x) r_{j'}(x)]$ equals to $\frac{1}{|S_i|}\mathrm{Var}_ {x \sim D}[r_{j}(x) r_{j'}(x)]$. Finally, we substitute $|S_i|=\sqrt{N}$.
>
> **Question 2.** Even though the polynomial $r_j r_{j’}$ is of degree $2k$, it has the extra property of being a product of two degree-k polynomials. If $(r_j(x) r_{j’}(x))^2>B^2$ it has to either be the case that $(r_j(x))^2>B$ or $(r_{j’}(x))^2>B$. The probability of either of those events can be bounded by referring to the definition of $k$-tameness.
>
> **Question 3.** We confirm that the additive term $\Delta$ is indeed not necessary in the second line of Equation (B.4). We will correct this typo in the revision. We however note that the third line of Equation (B.4) should still have an additive term of $2\Delta$ (replacing the current value $\Delta$ with the value $2\Delta$ entails changing $\Delta$ to $\Delta/2$ in the line after line 701 and does not change the conclusion on line 702). The third line of Equation (B.4) follows from the second line of Equation (B.4) (with $\Delta$ removed) as follows. By triangle inequality, the second line of Equation (B.4) can be upper-bounded by a sum of terms if we let $u_j$ denote $\mathbb{E}_ {x \sim S} [f(x) \mathbf{1}_ {j\Delta \leq (p(x))^2 < (j+1)\Delta } ]$ and $v_j$ denote $\mathbb{E}_ {x \sim D’} [f(x) \mathbf{1}_ {j\Delta \leq (p(x))^2 < (j+1)\Delta } ]$, then by the triangle inequality we see that the second line of Equation (B.4) is at most $\sum_{j=0}^{B/\Delta}( (j \Delta) |u_j - v_j| + \Delta u_j + \Delta v_j)$. We also observe that $j \Delta \leq B$, and $\sum_{j=0}^{B/\Delta} u_j \leq 1$ and $\sum_{j=0}^{B/\Delta} v_j \leq 1$ which bounds the whole expression by $2\Delta+B\sum_{j=0}^{B/\Delta}( |u_j - v_j|).$
>
> **Question 4.** Thank you for pointing this out, we added a short claim that proves that there indeed exists a value of $\tau_i$ satisfying the condition, assuming certain high-probability events take place. The proof is based on a simple averaging argument. In particular, $\tau_i$ is defined as the threshold such that the current set $S_i ^{\text{filtered}}$ contains unreasonably many points $x$ such that $(p_i(x))^2 >\tau_i$. If such a threshold $\tau_i$ did not exist, then $S_i ^{\text{filtered}}$ would not satisfy line 919 (which is a contradiction).
>
> More formally, for the sake of contradiction, suppose that for every $\tau \ge 0$ it is the case that
> $$
> \frac{1}{N} | \{ x \in S_i^ {\text{filtered}} : (p_ i(x))^2 > \tau \} | \le \frac{10}{\alpha} ( \mathbb{P}_ {x \sim S_D} [B_0 \ge (p_ i(x))^2 > \tau] + \Delta_0 ). \tag{1}
> $$
> Since every element $x$ of $S_i^ {\text{filtered}}$ satisfies $(p_i(x))^2 \le B_0$ we have
> $$
> \frac{1}{N} \sum_{x \in S_i^{\text{filtered}}} (p_i(x))^2 = \int_{\tau = 0}^{B_0} \frac{1}{N} | \{ x \in S_i^{\text{filtered}} : (p_i(x))^2 > \tau \} | d\tau. \tag{2}
> $$
> Combining Equations (1) and (2) gives
> $$
> \frac{1}{N} \sum_{x \in S_i^ {\text{filtered}}} (p_i(x))^2 \le \frac{10}{\alpha} \left( \Delta_0 B_0 + \int_{0}^{B_0} \mathbb{P}_ {x \sim S_D} [B_0 \ge (p_i(x))^2 > \tau] d\tau \right),
> $$
> which in turn is bounded by $\frac{10}{\alpha} ( \Delta_0 B_0 + \mathbb{E}_ {x \sim S_D} [ (p_i(x))^2 \cdot 1_{x \le B_0} ] )$. Additionally, since $S_D$ is assumed to satisfy property (3) in Claim 1, and $\hat{M}$ is assumed to satisfy Equation (E.1) (see line 929), we have
> $\mathbb{E}_ {x \sim S_D} \left[ (p_i(x))^2 \cdot 1_ {(p_i(x))^2 \le B_0} \right] \le 2\mathbb{E}_ {x \sim D}[(p_i(x))^2] \le \frac{11}{5} (p_i)^T \hat{M} (p_i)  \le \frac{11}{5} \le 5$. Overall, we have bounded $\frac{1}{N} \sum_{x \in S_i^ {\text{filtered}}} (p_i(x))^2$ by $\frac{10}{\alpha} \left( \Delta_0 B_0 + 5 \right)$, which contradicts the premise that $\frac{1}{N} \sum_{x \in S_i^ {\text{filtered}}} (p_i(x))^2 > \frac{50}{\alpha} (1 + \Delta_0 B_0)$, finishing the proof.

---

> > ### Comment · Reviewer_tKBR · 2024-08-12
> >
> > I thank the authors for the detailed answers to my and other reviewers questions, and I will increase my score.

---

### Official Review · Reviewer_JMSW · 2024-07-13

**Soundness:** 3
**Presentation:** 3
**Contribution:** 4
**Rating:** 8
**Confidence:** 4

**Summary:**

This paper studies two fundamental learning setups, namely PQ learning and TDS learning (Testable Learning with distribution shift), both motivated by covariate shift.

In PQ learning, the algorithm is given labeled samples from $\mathcal D^{\text{train}}$ over $\mathbb R^d \times \{0, 1\}$ with marginal distribution $D$, and unlabeled samples from some test distribution $\mathcal D^{\text{test}}$ (also over $\mathbb R^d \times \{0, 1\}$). Assume that there is some common hypothesis $h$ that achieves a small error $\lambda$ under both $\mathcal D^{\text{train}}$ and $\mathcal D^{\text{test}}$, the goal of the learning algorithm is to output a selector $g: \mathbb R^d \mapsto \{0, 1\}$ and a classifier $\hat h$ such that (i) $\hat h$ achieves a small error with respect to $\mathcal D^{\text{test}}$ restricted to the acceptance region of $g$, and (ii) most samples from $D$ will not be rejected by $g$. The main contribution is an algorithm that works when (i) the marginal $D$ of $\mathcal D^{\text{train}}$ satisfies some polynomial concentration properties (including isotropic log-concave) and (ii) the hypothesis class admits $L_2$ sandwiching polynomial under $D$. In particular, it achieves a rejection rate of $\eta$ and an accuracy guarantee of $O( \lambda / \eta )$. The bounds are optimal when one sets $\eta = \sqrt{\lambda}$ to balance the rejection rate and accuracy.

In tolerant TDS learning, the algorithm similarly has labeled sample access to $D^{\text{train}}$ and unlabeled sample access to $D^{\text{test}}$. In addition, the algorithm is allowed to reject $\mathcal D^{\text{test}}$ when it finds that the marginals of $D^{\text{train}}$ and $\mathcal D^{\text{test}}$ are at least $\theta$-far from each other in TV distance.  The name ``tolerance'' follows exactly from the constraint that the algorithm is not allowed to reject when $D^{\text{train}}$ and $\mathcal D^{\text{test}}$ are close but not identical to each other. Their algorithm works in a setup similar to the PQ learning case and achieves a final error of $O(\lambda) + 2 \theta + \epsilon$, where $\lambda$ is the optimal error of some common hypothesis under $\mathcal D^{\text{train}}$ and $\mathcal D^{\text{test}}$ (the sum of the two errors), $\theta$ is the tolerance in TV distance, and $\epsilon$ is the error parameter.

The core technique is a spectral based outlier removal procedure that resembles that from [DKS18] that performs filtering on the dataset until the $L_2$ norm of any polynomial under the empirical distribution becomes at most a constant factor of its $L_2$ norm under the reference distribution (the marginal of $\mathcal D^{\text{train}}$). However, the analysis in the current work is tighter, and in particular exhibits non-trivial guarantees even if $\mathcal D^{\text{test}}$ is very far from $\mathcal D^{\text{train}}$ in total variation distance, which is crucial in the technique's application in PQ learning. After ensuring that the spectral of the dataset is appropriately bounded, the algorithm runs polynomial regression, and its guarantees mainly follow from the existence of $L_2$ sandwiching polynomials.

**Strengths:**

The application of spectral outlier removal in the context of distribution shift is natural and turns out to b quite powerful. The most surprising part is that the procedure works even when the TV distance between the test and the training distributions is more than $1/2$. A naive application of spectral outlier removal will not work in this case as the filter may end up removing all points from the dataset (since the standard guarantee only says that the filter remove more "bad" points than "good" points). The analysis of this work circumvents the issue, and characterizes a tradeoff between the rejection rate $\alpha$ and the final bound on the $L_2$ norm of polynomials. This makes PQ learning possible even when the TV distance between $\mathcal D^{\text{train}}$ and $\mathcal D^{\text{test}}$ approaches $1$ (albeit with a cost of having larger learning error in the end).

**Weaknesses:**

A recent work [CKKSV24] shows that non-tolerant TDS learning can also be accomplished when the hypothesis classes has $L_1$ sandwiching polynomials. This makes TDS learning of some important hypothesis classes such as $AC_0$ circuits and quadratic threshold functions possible. However, the technique of spectral outlier removal in this work seems only applicable when one has $L_2$ sandwiching polynomial as otherwise one cannot easily replace moment-matching with spectral boundedness.

**Questions:**

In Definition 4.5, should it be "from some distribution $\mathcal D^{\text{train}}$" instead of "from some distribution $\mathcal D$"?

**Limitations:**

Yes, they have.

---

> ### Author Rebuttal · Authors · 2024-08-05
>
> Thank you for your time and for appreciating our work. In Definition 4.5, the distribution $\mathcal{D}$ represents some distribution over the features, which is unlabeled. We instead typically use $\mathcal{D}^{\mathrm{train}}$ to denote the labeled training distribution (see lines 136–137 and also Definition 4.1).

---

### Official Review · Reviewer_wKf2 · 2024-07-13

**Soundness:** 4
**Presentation:** 2
**Contribution:** 3
**Rating:** 7
**Confidence:** 4

**Summary:**

This work considers learning under distribution shift. Here, a learner receives i.i.d labeled data from $D_{train}$, along with i.i.d unlabled data from $D_{test}$. It then builds a classifier with the goal of achieving high accuracy over $D_{test}$. Under arbitrary conditions, this task is impossible, so in the PQ-learning framework the classifier is allowed to abstain from prediction for points with a frequency close to the total-variation distance between $D_{train}$ and $D_{test}$. This work also considers the TDS setting in which the learner is allowed to completely abstain from prediction if it believes its inputs are drawn from a different data distribution.

The main technical idea of this work is an outlier detection scheme, which (efficiently) computes an outlier detection function $g$ meant to distinguish the points in the training sample from $D_{test}$ that are "far" from the support of $D_{train}$. Their outlier procedure works by iteratively utilizing a quadratic program to find a polynomial of degree k that strongly distinguishes an outlier set. They then remove points until the outlier set only consists of points where the polynomial solution evaluates to very large amounts. The overall idea here is that similar distributions (i.e. if $D_{train} = D_{test}$) will lead to similar polynomial evaluations over them, and consequently the existence of a "separating" polynomial serves as a means to detecting where the distributions differ.

This idea is reflected in their first result, which shows that the output of this scheme results in a funtion $g$ such that upon filtering with $g$, polynomial evaluations over the test and training distribution must be bounded within a factor of each other (based on tolerance parameters). They additionally bound the runtime of their algorithm. The only requirement for this theorem to hold is that the training distribution must satisfy regularity conditions based on the degree of polynomials being used, and they subsequently show that these conditions are met by isotropic log-concave distributions.

Utilizing this technical idea, this work proceeds by giving PQ-learning algorithm for half-spaces over Gaussian training distributions. Despite the limited scope of this case, this work neverthless provides the first dimension-efficient solution to this problem. Then, this work continues with a more general result for PQ-learning, which, under a condition of "reasonableness" for a pair $(D, F)$, gives an efficient learning algorithm. Under the same condition, this work concludes by giving results for success in the related TDS setting.

**Strengths:**

This paper offers a solution to well-known theoretical problem that achieves the best known bounds. I particularly like that their algorithms are computationally efficient (in addition to enjoying the typical performance guarantees). Their technical idea for outlier detection (which forms the core of the paper) is relatively well explained and seems to me an innovative way to approach outlier detection.

**Weaknesses:**

The latter half of the paper is fairly burdened with algebra and consequently a bit difficult to follow. I would have appreciated a greater focus on intuition with more of the technical details being left to the appendix.

**Questions:**

1. Could you give some intuition for why you began your PQ-results for Gaussian distributions? The outlier-detection procedure works for a more general set of distributions, so it would be interesting if you could spell out the additional properties of Gaussians that make your first result possible.

2. Could you similarly give more intuition about the $(F, D)$-reasonableness condition? The definition feels rather extensive (given 3 conditions in it) and i consequently feel it could merit a longer discussion. As it is currently written, it feels a bit convoluted to me.

**Limitations:**

Yes.

---

> ### Author Rebuttal · Authors · 2024-08-05
>
> We wish to thank the anonymous reviewer for their constructive feedback and suggestions.
>
> **Gaussian assumption:** The reviewer is correct that the outlier removal procedure works under weaker assumptions than Gaussianity. In particular, it will work for any tame distribution (see lines 155–158 for a definition). However,  in order to obtain our PQ learning results, we also make use of the existence of low degree $\mathcal{L}_2$-sandwiching approximators. The existence of such approximators has been proven in prior work for the Gaussian distribution as well as for the uniform distribution over the hypercube. Nonetheless, we believe that simple classes like halfspaces and halfspace intersections admit low-degree sandwiching approximators with respect to other distributions as well and establishing appropriate bounds is an interesting direction for future work. The important relevant properties are, in general, both concentration and the anti-concentration of the Gaussian distribution, which are, for example, also satisfied by strongly log-concave distributions.
>
> **Reasonable pairs:** Definition 4.5 indeed consists of 3 properties. The first one is the existence of sandwiching approximators, which is known to be important for learning in the presence of distribution shift by prior work on TDS learning [KSV24b]. The second one is the tameness condition, which is important for the outlier removal procedure and was introduced in the work of [DKS18] for similar purposes. The third one ensures generalization for polynomial regression. We will add an appropriate discussion in future revisions.
>
> *[DKS18] Ilias Diakonikolas, Daniel M Kane, and Alistair Stewart. Learning geometric concepts with nasty noise. In Proceedings of the 50th Annual ACM SIGACT Symposium on Theory of Computing, pages 1061–1073, 2018.*
>
> *[KSV24b] Adam R Klivans, Konstantinos Stavropoulos, and Arsen Vasilyan. Testable learning with distribution shift. 37th Annual Conference on Learning Theory, 2024.*

---

> > ### Comment · Reviewer_wKf2 · 2024-08-12
> > **Response to Rebuttal**
> >
> > Thank you for your response. Overall, I'll maintain my current score but increase my confidence.

---

### Decision · Program_Chairs · 2024-09-25

**Decision:**

Accept (spotlight)

**Comment:**

This is a theory submission about the important problem of learning under covariate shift. The authors consider the two recent frameworks of testable learning [KSV24b] and PQ learning [GKKM20], where both frameworks allow abstentions (but in different ways). The goal is to have a computationally efficient learner than does not make a lot of unnecessary abstentions. The authors achieve this for some classes of hypotheses (e.g., intersection of half spaces) and some distributions (e.g., Gaussians) through the use of a spectral outlier removal. The authors' significant contributions were appreciated by the reviewers. There is room for improvement in terms of the clarity of the proofs, especially in the appendices.